# FoxM1 repression during human aging leads to mitotic decline and aneuploidy-driven full senescence

Joana Catarina Macedo[1,2], Sara Vaz[1,2], Bjorn Bakker[3], Rui Ribeiro [1,2], Petra Lammigje Bakker[3], Jose Miguel Escandell[4], Miguel Godinho Ferreira [4,5], René Medema[6], Floris Foijer [3] & Elsa Logarinho [1,2,7]

Aneuploidy, an abnormal chromosome number, has been linked to aging and age-associated diseases, but the underlying molecular mechanisms remain unknown. Here we show, through direct live-cell imaging of young, middle-aged, and old-aged primary human dermal fibroblasts, that aneuploidy increases with aging due to general dysfunction of the mitotic machinery. Increased chromosome mis-segregation in elderly mitotic cells correlates with an early senescence-associated secretory phenotype (SASP) and repression of Forkhead box M1 (FoxM1), the transcription factor that drives G2/M gene expression. FoxM1 induction in elderly and Hutchison–Gilford progeria syndrome fibroblasts prevents aneuploidy and, importantly, ameliorates cellular aging phenotypes. Moreover, we show that senescent fibroblasts isolated from elderly donors' cultures are often aneuploid, and that aneuploidy is a key trigger into full senescence phenotypes. Based on this feedback loop between cellular aging and aneuploidy, we propose modulation of mitotic efficiency through FoxM1 as a potential strategy against aging and progeria syndromes.

[1] Aging and Aneuploidy Laboratory, IBMC, Instituto de Biologia Molecular e Celular, Universidade do Porto, 4200-135 Porto, Portugal. [2] i3S, Instituto de Investigação e Inovação em Saúde, Universidade do Porto, 4200-135 Porto, Portugal. [3] European Research Institute for the Biology of Aging, University of Groningen, University Medical Center Groningen, NL-9713 AV Groningen, The Netherlands. [4] Telomere and Genome Stability Laboratory, Instituto Gulbenkian de Ciência, 2781-901 Oeiras, Portugal. [5] Telomere Shortening and Cancer Laboratory, Institute for Research on Cancer and Aging (IRCAN), UMR7284, U1081, UNS, 06107 Nice, France. [6] Division of Cell Biology and Cancer Genomics Center, The Netherlands Cancer Institute, Plesmanlaan 121, 1066 CX Amsterdam, The Netherlands. [7] Cell Division Unit, Faculty of Medicine, Department of Experimental Biology, Universidade do Porto, 4200-319 Porto, Portugal. Correspondence and requests for materials should be addressed to E.L. (email: elsa.logarinho@ibmc.up.pt)

Numerous studies over the past decades have supported a link between aging and aneuploidy[1]. This association has been well documented for oocytes and is considered to be the main cause of miscarriage and birth defects in humans[2]. However, aneuploidy can also arise in somatic cells, and a number of studies have reported age-dependent increases in aneuploidy[3–7]. These studies have shown that aging is positively correlated with the incidence of chromosome mis-segregation, raising the question whether there is a general dysfunction of the mitotic apparatus in aged cells[8]. Transcriptome analyses of a panel of fibroblast and lymphocyte cultures from young and old individuals revealed changes in the expression of genes controlling the mitotic machinery[9,10]. However, the use of mixed cell populations in different stages of the cell cycle and the lower mitotic indexes (MIs) of elderly cell cultures has limited these studies from clearly demonstrating whether mitotic genes are repressed intrinsically in old dividing cells. Moreover, analysis of the mitotic process in aging models and diseases has been scarce and a comprehensive analysis of cell division efficiency in naturally aged cells is largely missing.

More recently, aneuploidy has been also linked to aging. Mutant mice with low levels of the spindle assembly checkpoint (SAC) protein BubR1 were found to develop progressive aneuploidy along with a variety of progeroid features, including short lifespan, growth retardation, sarcopenia, cataracts, loss of subcutaneous fat, and impaired wound healing[11–14]. In addition, systematic analyses of disomic yeast, trisomic mouse, and human cells, all cells with an extra chromosome, have elucidated the impact of aneuploidy in cellular fitness[15]. The cumulative effect of copy number changes of many genes induces the so called pan-aneuploidy phenotypes, namely proliferation defects[16,17], gene signature of environmental stress response[18,19], multiple forms of genomic instability[20–22], and proteotoxicity[23–25], which, interestingly, are hallmarks of cellular aging[26].

Together, these observations suggest that there is a positive correlation between aging and aneuploidy, but evidence for mitotic decline in elderly dividing cells and for aneuploidy-driven permanent loss of proliferation capacity (full senescence) is limited. Here, we used live-cell time-lapse imaging to investigate the mitotic behavior of human dermal fibroblasts (HDFs) collected from healthy Caucasian males with ages ranging from neonatal to octogenarian and cultured under low passage number. We show that mitotic duration increases with advancing age, concurrent with a higher rate of mitotic defects. We demonstrate this mitotic decline to arise from a transcriptional repression of mitotic genes in pre-senescent dividing cells exhibiting senescence-associated secretory phenotype (SASP). We show short-term induction of Forkhead box M1 (FoxM1) transcriptional activity to improve mitotic fitness and ameliorate senescence phenotypes in elderly and progeroid cells. Finally, we show aneuploidy to induce full senescence in naturally aged cells, which suggests that mitotic fitness enhancement may be a potential anti-aging strategy.

## Results

**Aneuploidy increases during natural aging.** In our study, we used HDFs collected from healthy Caucasian males with ages ranging from neonatal to octogenarian (Supplementary Table 1), to test whether aneuploidy increases during natural aging and if this is a consequence of a general dysfunction of the mitotic machinery in aged cells. As inter-individual differences exist in the rate at which a person ages (biological age), we included two donors of each age in a total of five chronological ages to increase the robustness of any correlation found with aging. In addition, we used dermal fibroblasts from an 8-year-old child with the Hutchison–Gilford progeria syndrome (HGPS) as a model of

premature cellular aging[27]. Considering that during a normal post-natal lifespan, cells in vivo will hardly reach the exhaustion number of replications observed in culture[28], and to limit any artifacts and clonal selection of in vitro culturing, only early cell culture passages (≤5) way below replicative lifespan exhaustion were used (population doubling level (PDL) < 24; Supplementary Table 1 and Methods). Chronic accumulation of macromolecular damage during natural aging induces a cellular stress response known as senescence[29], and accumulation of senescent cells has been widely reported for chronological aging and age-related disorders[30,31]. Accordingly, we found higher levels of senescence-associated (SA) biomarkers[32], measured under strict quantitative parameters by microscopy analysis, in the proliferating fibroblast cultures from older individuals and the HGPS patient (Supplementary Fig.1 a–e), thus validating their suitability as models of advanced and premature aging, respectively.

To determine whether aneuploidy increases with aging, we performed fluorescence in situ hybridization (FISH) for three chromosome pairs in asynchronous cell populations from different age donors and scored all samples blindly. We found significantly higher (two-tailed $\chi^2$ test) aneusomy indices (ratio of aneusomic cells for chromosomes 7, 12, and 18 to the total cell count for a sample) in the middle-aged, old-aged, and progeria samples (Fig. 1a). Interestingly, in these samples, not all single-chromosome aneusomy indices were significantly higher, suggesting that a combination of multi-chromosome aneusomy indices is a stronger predictor of mild aneuploidy levels if there are chromosomes with unequal probability of contributing to aneuploid progeny or if different aneuploidies are distinctly selected in the cell population (Supplementary Table 2). Moreover, as aneuploid cells are most likely outcompeted in culture by diploid cells[16], thereby diluting the aneuploidy index, we additionally measured the rate of chromosome mis-segregation (number of events in which two sister chromatids co-segregate to the same daughter cell) by combining a cytokinesis-block assay with FISH staining. In this approach, we scored all binucleated (BN) cells generated during a 24 h treatment with cytochalasin D (cytokinesis inhibitor; see Methods). We found the percentage of BN cells with chromosome mis-segregation over the total BN cell count for a sample to be significantly higher (two-tailed $\chi^2$ test) in the middle-aged, old-aged, and progeria cultures (Fig. 1b). Taken together, these FISH experiments show that increased mis-segregation rates are associated with mild aneuploidy levels in elder cultures, supporting the idea of an age-associated loss of mitotic fidelity.

**Elder cells divide slower with increased rate of mitotic defects.** To gain insight into how old cells divide, we followed individual mitotic cells by long-term phase-contrast time-lapse imaging. Interestingly, we found the interval between nuclear envelope breakdown (NEB) and anaphase onset to increase steadily with advancing age (Fig. 1c, d). To exclude any effects due to genetic heterogeneity between the Caucasian donors and/or biobank discrepancies in culture set up, we used mouse adult fibroblasts recurrently sampled from female mice over a period of 2 years. In this model, culture conditions were highly homogenous, thereby providing a solid "ex vivo" model of chronological aging. Again, both mitotic duration (Supplementary Fig. 2a–c) and SA biomarkers (Supplementary Fig. 2d, e) progressively increased with aging.

We then asked what could be leading to the age-associated mitotic delay. Bypass of short telomere-triggered senescence by disruption of tumor-suppressive pathways, shown to elicit telomere fusion-driven prolonged mitosis[33], was ruled out as potential cause, as we found no evidence for chromosome fusions

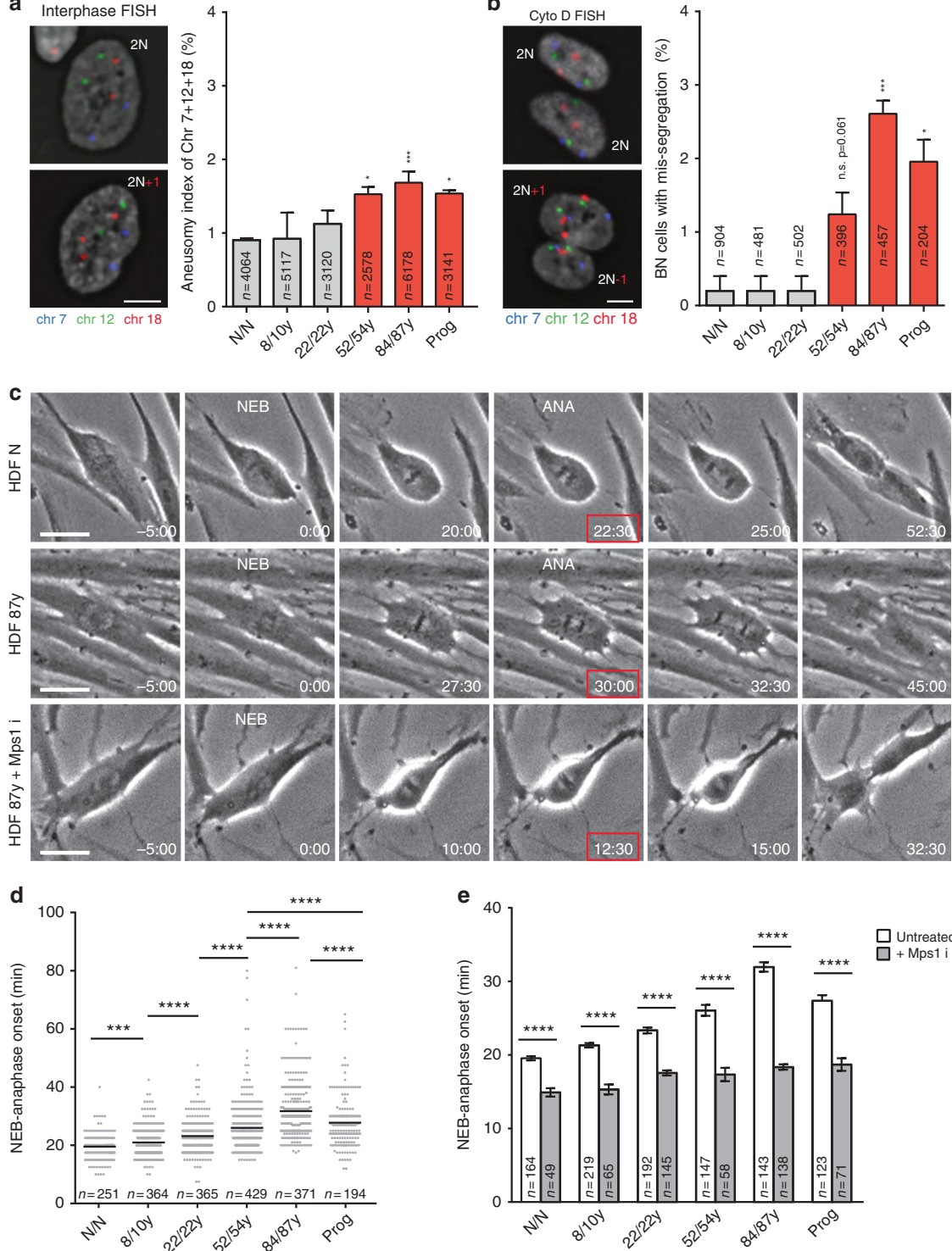

**Fig. 1** Aneuploidy and mitotic duration increase during normative aging. **a** Aneusomy index of three chromosome pairs (7, 12, and 18) in interphase cells from different age donors. **b** Percentage of cytochalasin D-induced binucleated cells with chromosome 7, 12, and 18 mis-segregation. Representative images of euploid/2N and aneuploid/2N ± l cells are shown in **a**, **b**, with chromosome-specific centromeric probes as indicated. Scale bars, 5 μm. **c** Frame series of time-lapse phase-contrast movies of mitotic neonatal and elderly cells ± Mps1 inhibitor. Time, min:s. Scale bar, 20 μm. **d** Mitotic duration of individual fibroblasts from different donors, measured from nuclear envelope breakdown (NEB) to anaphase onset (ANA). **e** Mitotic duration under standard conditions as in **d** (white bars) and following treatment with Mps1 inhibitor (gray bars). Values are mean ± SD from at least three independent experiments, using two biological samples of similar age (except for progeria). Sample size (n) is indicated in each graph. NS, $p > 0.05$, *$p \leq 0.05$, **$p \leq 0.01$, ***$p \leq 0.001$, and ****$p \leq 0.0001$ in comparison with neonatal (N/N) by two-tailed $\chi^2$ (**a**, **b**) and Mann–Whitney (**d**, **e**) statistical tests

in metaphase spreads of the primary dermal fibroblasts used in our study (Supplementary Fig. 3a). Activation of the SAC by defective kinetochore–microtubule attachments and/or reduced efficiency of the ubiquitin-proteasome system induced by proteotoxic stress could alternatively explain the delay between NEB and anaphase onset in older cells. We found the mitotic delay to depend on SAC activity as treatment with a small molecule inhibitor of the Mps1 kinase rescued mitotic duration to similar levels in all cell cultures (Fig. 1c, e). Enhancement of proteasome activity with a small molecule inhibitor of the Usp14 deubiquitinase did not change mitotic duration considerably (Supplementary Fig. 3b). As a correlation between cell size, mitotic duration, and SAC strength was recently described[34], we further tested whether aging-associated mitotic delay was due to increased cell size. We found no correlation between mitotic duration and cell size in fibroblast cultures (Supplementary Fig. 3 c-f). In addition, when treated with taxol and the kinesin-5 inhibitor (S-Trityl-l-cysteine, STLC) to induce chronic activation of the SAC, elderly cells arrested in mitosis as long as young cells before they slipped out, suggesting that SAC strength is similar (Supplementary Fig. 3g, h).

To investigate chromosome and/or spindle defects contributing to increased mitotic duration in older cells, we performed high-resolution spinning-disk confocal microscopy in cells expressing H2B–GFP and α-Tubulin–mCherry (Supplementary Movies 1–4). We found several mitotic defects to be increased in middle-aged and old-aged samples (Fig. 2a–f), namely chromosome congression delay (Fig. 2b, e, f; Supplementary Movie 2), anaphase lagging chromosomes (Fig. 2c, e, f; Supplementary Movie 3), and spindle mispositioning in relation to the growth surface (Fig. 2d–f; Supplementary Movie 4). In agreement with proper SAC functioning (Supplementary Fig. 3g, h), aged cells entering anaphase with unaligned chromosomes were never observed. Moreover, we found middle-aged, old-aged, and HGPS cells to more often exhibit micronuclei (an outcome of anaphase lagging chromosomes[35]; Fig. 2g) and fail cytokinesis (Fig. 2h), based on complementary quantitative cell-based assays (fixed-cell analysis and phase-contrast live-cell imaging, respectively; Methods). Overall, the data indicated that aging triggers abnormalities at several mitotic stages, in agreement with the increased levels of aneuploidy found.

**Mitotic gene transcriptional shutdown and SASP in elderly mitotic cells.** To identify the molecular mechanisms behind this complex aging-associated mitotic phenotype, we performed RNA-sequencing (RNA-seq) gene expression profiling of cells captured in mitosis, accordingly to the experimental layout shown in Fig. 3a. Fibroblast cultures from neonatal and octogenarian donors were treated with STLC to enrich for MI, and mitotic cells collected by mechanical detachment. As cell synchronization (mitotic enrichment) was limited in elderly vs. neonatal cultures, we used a higher number of cell culture flasks with octogenarian fibroblasts treated with STLC to collect equivalent numbers of mitotic elderly and neonatal cells for RNA-seq. The procedure yielded purified cell populations with mitotic indices >95% in both neonatal and 87 y donor cultures (Fig. 3b, c), and importantly, it uncoupled any changes found in mitotic gene expression from the samples' differences in MI (Supplementary Fig. 4a) and overall proliferation capacity (Supplementary Table 1). Moreover, apoptosis was excluded as potential cause of reduced proliferation in elderly cell samples as similar levels of apoptotic cells were found in neonatal and octogenarian samples by flow cytometry analysis for annexin V staining (Supplementary Fig. 4b, c).

RNA-seq revealed that the abundance of 3309 gene transcripts was altered in octogenarian mitotic fibroblasts compared to neonatal mitotic fibroblasts (Fig. 3d; Supplementary Data 1). Principal component analysis showed that two independent experiments from the same donor were consistent for the main component (Fig. 3e). In agreement with the elderly cell defects in mitosis, the top 10 most altered Gene Ontology (GO) terms included six cell cycle-related and mitosis-related gene ontologies (Fig. 3f). Seventy-one "mitosis" GO genes were altered in octogenarian mitotic fibroblasts ($p$-value < 0.05; likelihood ratio test; Supplementary Data 2), representing a 1.97-fold enrichment of alterations in this gene set ($p$-value = 9.73E−9, Fisher exact test) (Fig. 3g). Moreover, a comprehensive list of 51 SA genes, including CDKN1A, CDKN2A, GLB1, LMNB1, and SASP genes (i.e., chemokines, cytokines, metalloproteinases, and others)[36,37], was interrogated in the RNA-seq data set (Supplementary Data 3). We found altered expression of 26 genes in the elderly vs. neonatal mitotic cells, with 19 genes behaving as previously reported[36,37] (Fig. 3h; Supplementary Data 4). As the transcriptome of senescent cells is highly heterogeneous and depends on the cell type and senescence stimulus, we additionally extended our analysis into 55 genes recently identified as comprising a "senescence core signature[37]." Indeed, we found 19 genes of this signature to be differentially regulated in the RNA-seq data set of mitotic octogenarian dermal fibroblasts, with 16 genes behaving as reported[37] (Fig. 3i; Supplementary Data 5). Alterations in the expression levels of CDKN1A, MMP1, CXCL8, TSPAN13, and FAM214B, were further confirmed by real-time PCR analysis (Fig. 3j). Thus, an unforeseen SASP phenotype evolves in elderly dividing cells, alongside a global transcriptional shutdown of mitotic genes, likely accounting for aging-associated mild aneuploidy levels.

**FoxM1 repression dictates mitotic decline during natural aging.** RNA-seq analysis also disclosed FOXM1, the transcription factor that primarily drives the expression of G2/M cell cycle genes[38,39] (Fig. 4a), to be downregulated in elderly mitotic cells (2logFC: −0.85; false discovery rate (FDR) < 5%; likelihood ratio test; Supplementary Data 1). Indeed, 36 out of the 71 mitotic genes altered in octogenarian mitotic cells have been reported as targets of the Myb-MuvB (MMB)-FOXM1 transcription complex containing the cell cycle genes homology region (CHR) motif in their promoters[38–40] (Fig. 3b; Supplementary Data 6). In addition, loss of FOXM1 has previously been shown to cause pleiotropic cell cycle defects leading to embryonic lethality[41]. We therefore asked whether loss of mitotic proficiency during normative aging was due to FOXM1 downregulation. Indeed, increasing age correlated with decreased FoxM1 transcript and protein levels in our human (Fig. 4c–e) and mouse (Supplementary Fig. 2f–h) mitotic cell samples. Importantly, unlike previous studies linking FoxM1 repression and aging[42,43], by analyzing mitotic cell samples, we uncoupled FoxM1 downregulation from decreased cell proliferative capacity, demonstrating that elderly cells divide with intrinsically low levels of FoxM1. To get evidence that FoxM1 repression actually accounts for an elderly cell mitosis with low mitotic protein levels, we measured the protein levels of FoxM1 and some of its known mitotic targets (Cyclin B1, Plk1, Ndc80, and Cdc20)[44] by western blotting analysis of mitotic cell extracts (Fig. 4f, g), as well as by immunofluorescence analysis of single mitotic cells (Fig. 4h–j; Supplementary Fig. 5). These quantitative analyses demonstrated that in elderly mitotic cells, important protein players acting from mitotic entry (e.g., Cyclin B1) to anaphase onset (e.g., Cdc20) are in fact expressed at low levels. We thus conclude that an aging-associated FoxM1 repression likely accounts for an extensive transcriptional downregulation of mitotic genes and the phenotypes observed in early senescent dividing cells.

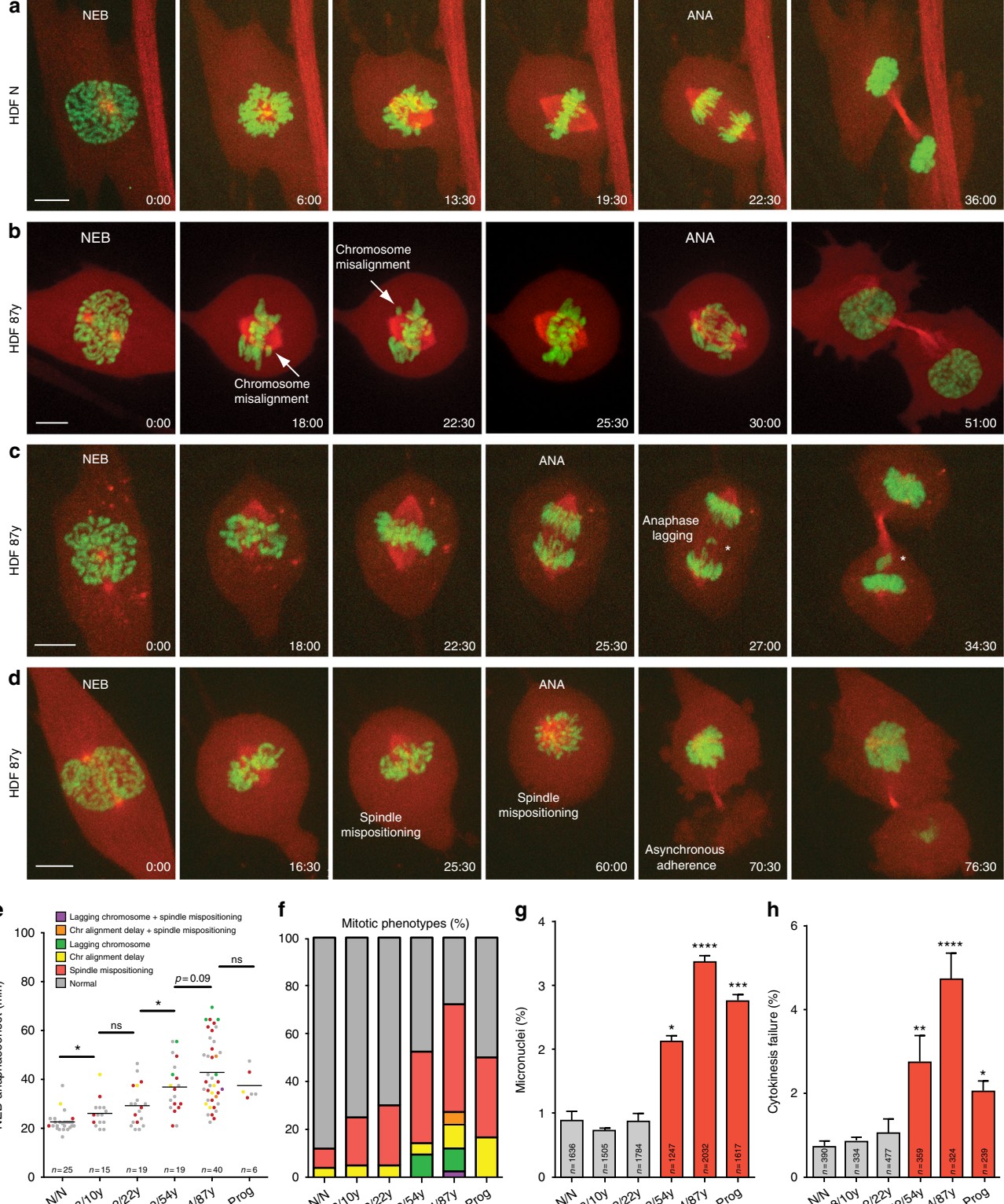

**Fig. 2** Mitotic defects associated with advanced age. **a–d** Frame series of spinning-disk confocal movies of neonatal (HDF N) and elderly (HDF 87 y) dividing cells expressing H2B–GFP/α-Tubulin–mCherry. **a** Neonatal cell division. **b–d** Elderly cell with chromosome alignment delay (**b**) with anaphase lagging chromosome and micronucleus formation (*) (**c**) or with spindle mispositioning in relation to the growth surface and asynchronous adherence of the daughter cells (**d**) (Supplementary Movies 1–4). Time, min:s. Scale bar, 5 μm. **e** Mitotic duration (NEB to anaphase onset) of HDFs from different age donors imaged under spinning-disk confocal microscopy. Mitotic phenotypes are indicated by distinct colors. **f** Quantification of the observed mitotic phenotypes. **g** Percentage of cells with micronuclei in fixed-cell analysis. **h** Rate of cytokinesis failure under phase-contrast microscopy. Values are mean ± SD from at least two independent experiments, using two biological samples of similar age (except for progeria). Sample size (*n*) is indicated in each graph. *$p \leq 0.05$, **$p \leq 0.01$, ***$p \leq 0.001$, and ****$p \leq 0.0001$ in comparison with N/N by two-tailed Mann–Whitney (**e**) and $\chi^2$ (**g**, **h**) statistical tests

To test the correlation between FoxM1 repression and age-associated phenotypes further, we RNA interference (RNAi)-depleted FoxM1 in fibroblasts from the 10-year-old donor (Supplementary Fig. 6a). Again, to uncouple gene expression analysis from the reduction in MI following FoxM1 RNAi (Supplementary Fig. 6b), we performed RNA-seq profiling of cells captured in mitosis accordingly to the experimental layout in Fig. 3a (Supplementary Fig. 6c). RNA-seq revealed that FoxM1 repression altered the abundance of 5841 gene transcripts (Supplementary Fig. 8a,b; Supplementary Data 7), which were

enriched for mitosis genes (143 genes, GO analysis; FDR < 5%, likelihood ratio test) (Fig. 5a; Supplementary Data 8). This constituted a 2.41-fold enrichment of alterations in this gene set (p-value = 7.43E−32, Fisher exact test). The top ten most altered GO terms included six related to mitosis (Supplementary Fig. 8d), and 224 out of 249 G2/M cell cycle genes previously reported as targets of the DREAM (DP, RB-like, E2F4, and MuvB) and MMB-FOXM1 transcriptional complexes[39], were downregulated following FoxM1 depletion (Supplementary Fig. 8d). Moreover, we found 1937 out of the 3309 genes altered in old-aged

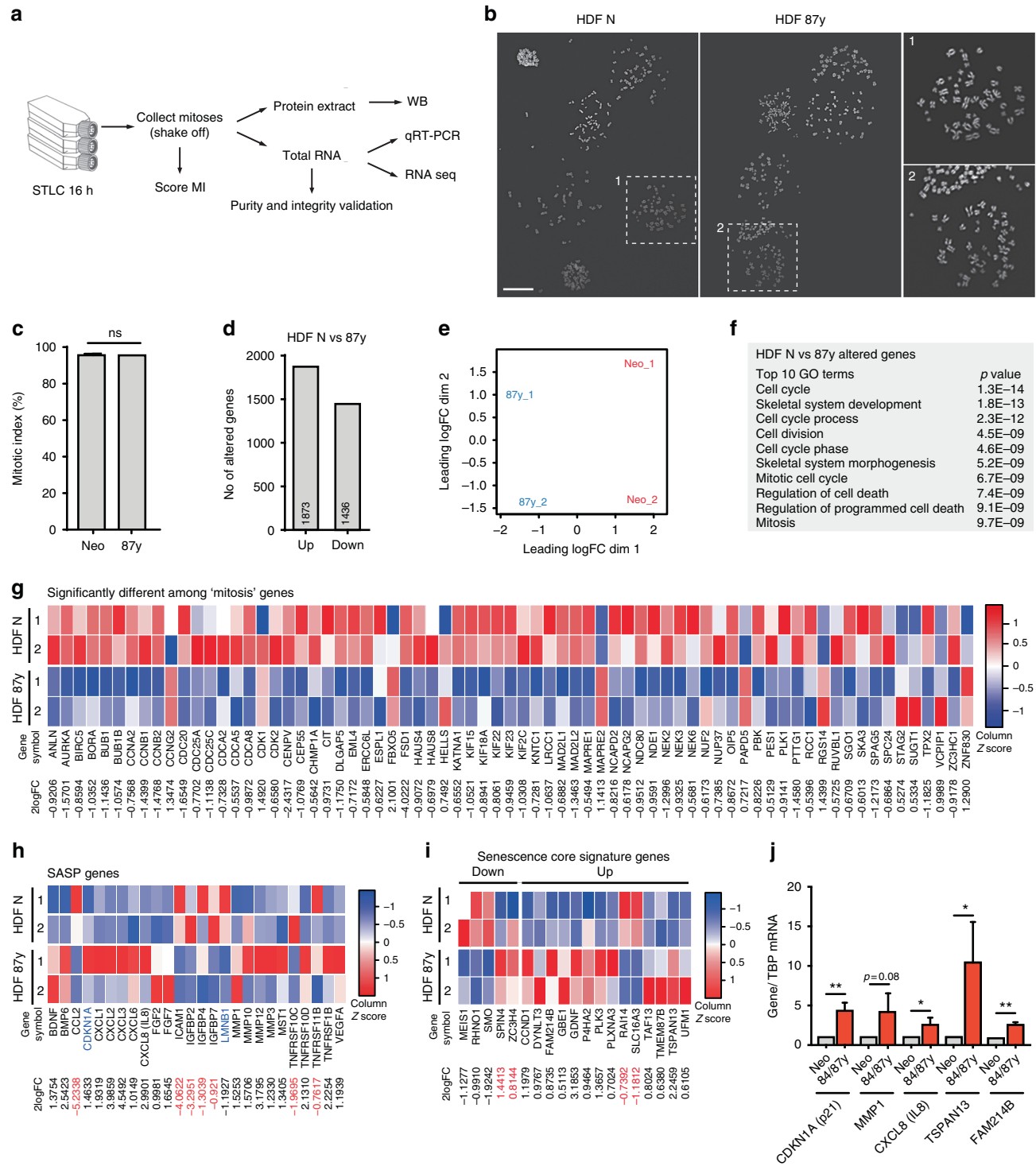

fibroblasts, to be also altered in FoxM1-depleted cells (Fig. 5b; Supplementary Data 9). Thus, 58.5% of the age-dependent transcriptional changes are FoxM1-dependent and enriched for cell cycle/mitosis GO terms (Fig. 5b). In agreement, FoxM1-depleted young fibroblasts displayed decreased protein levels of FoxM1 targets (Supplementary Fig. 6d, e), a mitotic delay (Fig. 5c), mitotic defects (Supplementary Fig. 6f–h), and increased aneuploidy (Fig. 5d, e), similar to old-aged fibroblasts. Interestingly, the percentage of cells with SA biomarkers also increased following FoxM1 repression (Fig. 5f, g), further supported by changes in the expression levels of *SASP* genes and genes of the senescence core signature (Fig. 5h, i; Supplementary Fig. 6i; Supplementary Data 10; Supplementary Data 11). Altogether, these results show that young cells with low levels of FoxM1, equivalent to those in octogenarian cells (Supplementary Fig. 7j,k), recapitulate the aging-associated mitotic defects, aneuploidy, and senescence phenotypes.

**FoxM1 induction rescues aging phenotypes in elderly and HGPS cells**. As FoxM1 depletion reduced mitotic fidelity and induced senescence in young cells, we hypothesized that FoxM1 overexpression in elderly cells should counteract aging phenotypes. Expression of a constitutively active truncated form of FoxM1 (FoxM1-dNdK) in old-aged fibroblasts[45,46] (Supplementary Fig. 7a) increased MI (Supplementary Fig. 7b) and resulted in altered expression of 2966 gene transcripts as determined by RNA-seq profiling of cells captured in mitosis accordingly to experimental layout in Fig. 3a (Supplementary Fig. 7c; Supplementary Data 12; Supplementary Fig. 8a, c). Altered gene transcripts were enriched for "mitosis" GO genes (79 genes; Fig. 6b; Supplementary Data 13). To gain insight into the mitosis genes found altered in elderly cells that are dependent on FoxM1 transcriptional activity, we performed comparative analysis with mitosis genes altered in 87 y FoxM1-dNdK and 10 y siFoxM1 samples (Fig. 6c; Supplementary Data 14). Fifty genes out of the 71 mitosis genes altered in 87 y cells were also altered in FoxM1-dNdK and siFoxM1 experimental conditions, suggesting that FoxM1 primarily accounts for age-related mitotic phenotypes. Concordantly, mitotic protein levels (Supplementary Fig. 7d, e) and age-associated mitotic defects were ameliorated in elderly fibroblasts expressing FoxM1-dNdK (Fig. 6a, d; Supplementary Movies 5, 6; Supplementary Fig. 7f–h), resulting in decreased aneuploidy levels (Fig. 6e, f). Furthermore, the percentage of cells with SA biomarkers was decreased (Fig. 6g, h), consistently to amelioration of SASP and senescence core transcriptional signatures (Fig. 6i, j; Supplementary Fig. 7i; Supplementary Data 15; Supplementary Data 16). Interestingly, the aging phenotypes were rescued by restoring FoxM1 levels in elderly cells to the levels as present in young cells, indicating that

only mild overexpression of FoxM1-dKdK is sufficient to overcome the aging phenotypes (Supplementary Fig. 7j, k). To gain insight into the SA genes found altered in elderly cells that rely on FoxM1 expression levels, we compared the senescence response between 87 y FoxM1-dNdK and 10 y siFoxM1 samples (Supplementary Fig. 8e, f). We found that both SASP and senescence core signature genes are largely regulated by FoxM1 levels, with 19 *SASP* genes and 14 senescence core signature genes consistently altered in all experimental conditions. Further validating our experimental conditions of FoxM1 expression modulation, we found extensive overlap between genes altered in 10 y siFoxM1 and 87 y FoxM1-dNdK (1915 genes; Supplementary Fig. 8d; Supplementary Data 17), with 158 out of the 249 DREAM/MMB-FOXM1 gene targets included in this overlap. Importantly, FoxM1-dNdK expression also partly rescued the mitotic defects (Supplementary Fig. 9a–d; Supplementary Movies 7, 8), aneuploidy levels (Supplementary Fig. 9e, f), and percentage of senescent cells (Supplementary Fig. 9g, h) in HGPS fibroblasts. Overall, these data demonstrate that modulation of mitotic efficiency through FoxM1 induction in elderly and HGPS cells prevents aneuploidy and delays cellular senescence.

**Increased aneuploidy in elderly senescent cells**. Experimental modulation of FoxM1 levels in the young (FoxM1 RNAi) and elderly fibroblasts (FoxM1-dNdK) demonstrated an inherent link between aneuploidy and senescence. Previous studies have found cellular/mouse models of constitutional aneuploidy or chromosomal instability (CIN) to prematurely senesce/age[11,47–51]. Although informative, these studies either represent conditions where aneuploidy is present from early development onward and/or conditions where CIN-inducing single gene editing/drug treatment leads to higher aneuploidy levels than those accumulated during normal aging. Whether the mild aneuploidy levels caused by global transcriptional shutdown of mitotic genes account for cellular senescence in natural aging remains unknown. To address this question, we fluorescence-activated cell (FACS)-sorted senescent cells from neonatal and elderly fibroblast cultures using SA-β-galactosidase (SA-β-gal) as a senescence marker and measured the aneusomy index of three chromosome pairs (Fig. 6a–f). As expected, the percentage of SA-β-gal-positive cells was considerably higher in elderly vs. neonatal early passage cell cultures (Fig. 7a, b, e), and consistent with our quantitative analysis using fluorescence microscopy (Supplementary Fig. 1a). Strikingly, even though the aneusomy indices of SA-β-gal-positive cells were significantly higher (two-tailed $\chi^2$ test) than those of unsorted controls in both neonatal and elderly samples, we found a cumulative 9.1% aneusomy index for three assessed chromosome pairs in elderly senescent cells in

---

**Fig. 3** Mitotic gene downregulation and senescence gene signature in elderly mitotic fibroblasts. **a** Experimental layout. Fibroblast cultures from different age donors were treated with kinesin-5 inhibitor (STLC) to enrich for mitotic index (MI). Mitotic (detached) cells were collected and inspected for MI. Protein extracts and total RNA were prepared for gene expression analyses. **b** Representative images of neonatal (HDF N) and elderly (HDF 87 y) mitotic cell shake-offs. Insets are ×2 magnifications. Scale bar, 50 μm. **c** MI quantification of neonatal and octogenarian mitotic cell shake-offs. **d** Number of altered genes in HDF 87 y RNA-seq comparing to HDF N (Supplementary Data 1). **e** RNA-seq multidimensional scaling (MDS) plot. Axes in MDS plot represent leading log-fold changes (logFCs) calculated by the root-mean-square of the largest absolute logFCs between each pair of libraries. **f** Top ten altered GO terms organized by *p*-values using the DAVID Functional Annotation tool. **g** Heatmap of genes within the "mitosis" GO term differentially expressed between HDF N and HDF 87 y (Supplementary Data 2). **h** Heatmap of SASP and senescence-associated (*CDKN1A, LMNB1*) genes differentially expressed between HDF N and HDF 87 y (Supplementary Data 4); 2logFCs in red indicate genes behaving differently from expected[36, 37]. **i** Heatmap of senescence core signature genes differentially expressed between HDF N and HDF 87 y (Supplementary Data 5); 2logFCs in red indicate genes behaving differently from expected[37]. In all heatmaps, 2logFC cutoff value < −0.5 and >0.5, *p*-value < 0.05; genes represented in columns and technical replicates represented in rows; *Z*-score column color intensities representing higher (red) to lower (blue) expression. **j** Transcript levels of senescence genes in total RNA from mitotic elderly samples normalized to *TBP* transcript levels and compared with neonatal. NS, *p* > 0.05, \**p* ≤ 0.05, and \*\**p* ≤ 0.01 by two-tailed Mann–Whitney statistical test (**c, j**)

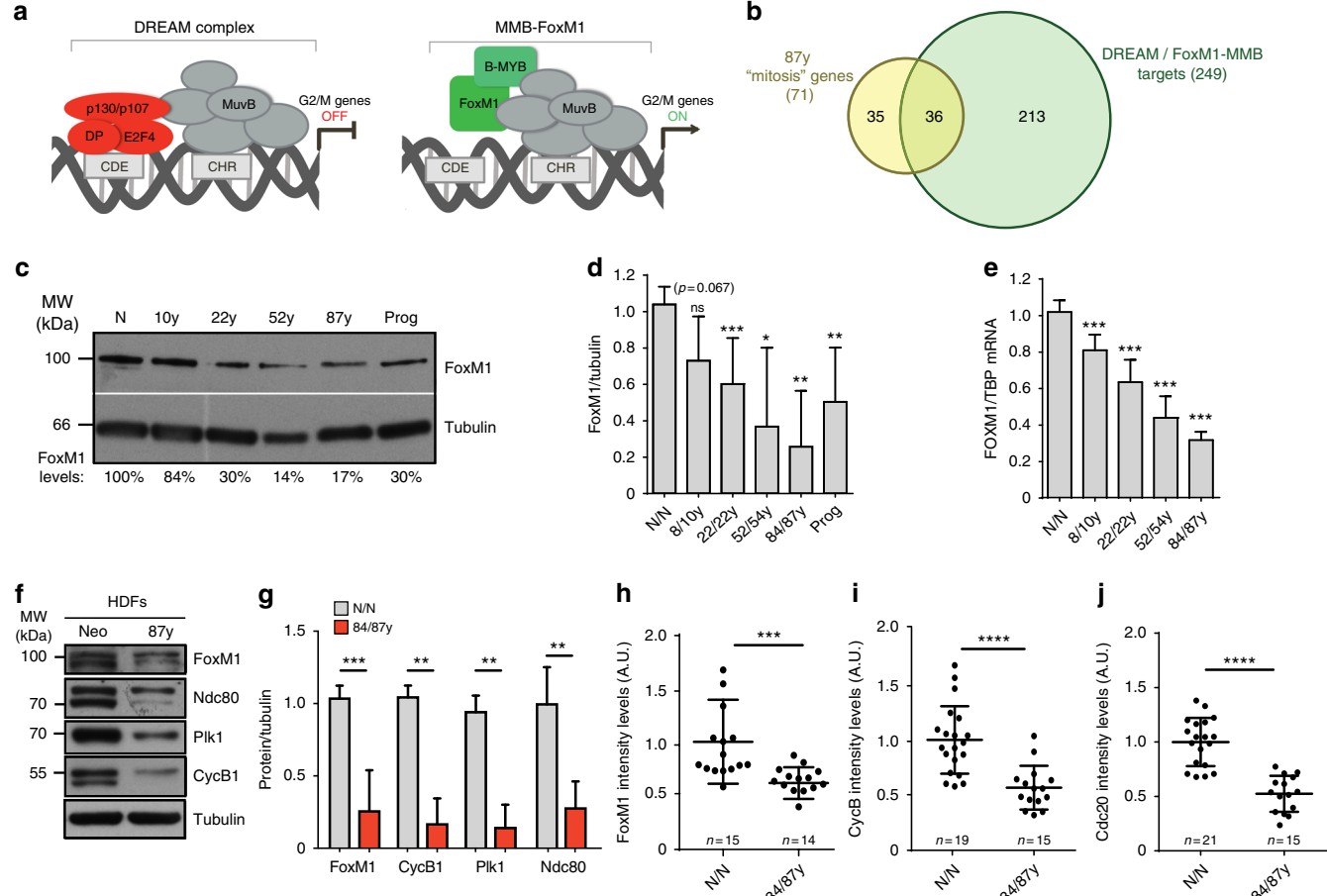

**Fig. 4** Aging cells express low levels of FoxM1 and mitotic transcripts during mitosis. **a** Regulation of G2/M cell cycle genes by competitive binding of DREAM and FoxM1-MMB transcriptional complexes to CDE/CHR promoter elements, as described in ref. [40]. **b** Venn diagram displaying the overlap between mitosis genes altered in elderly fibroblasts (87 y) and targets of DREAM/FoxM1-MMB complex[40] (Supplementary Data 6). **c, d** FoxM1 protein levels in mitotic extracts of different age samples. FoxM1 levels were normalized to α-tubulin levels and to the neonatal sample (N/N). **e** *FOXM1* transcript levels in mitotic total RNA from different age samples normalized to *TBP* transcript levels and compared to neonatal. **f, g** Protein levels of FoxM1 transcriptional targets in mitotic extracts of neonatal and elderly fibroblasts. Values are normalized to α-tubulin levels and to the neonatal sample (N/N). **h–j** Quantitative analysis of FoxM1 (**h**), Cyclin B (**i**), and Cdc20 (**j**) immunofluorescence intensity levels in $n =$ single mitotic cells (N/N and 84/87 y) (Supplementary Fig. 5). Relative intensity levels are indicated as arbitrary units (A.U.). Values are mean ± SD of two (**h–j**) or three (**d, e, g**) independent experiments. NS, $p > 0.05$, *$p \leq 0.05$, **$p \leq 0.01$, ***$p \leq 0.001$, and ****$p \leq 0.0001$ in comparison with N/N by two-tailed Mann–Whitney test

comparison with 3.2% in neonatal senescent cells (Fig. 7f). Thus, aneuploidy appears as an overt feature in elderly senescent cells, especially if we consider that only 3 out of 23 chromosome pairs were measured. To investigate whether FoxM1 repression is responsible for this difference, we next FACS-sorted SA-β-gal-positive cells from siFoxM1-depleted neonatal cell cultures and from octogenarian cell cultures expressing FoxM1-dNdK (Fig. 7c–e). We observed FoxM1 repression to induce an increase in the percentage of SA-β-gal-positive cells and their aneusomy index, and FoxM1-dNdK expression to decrease the percentage of SA-β-gal-positive cells and their aneusomy index. These results demonstrate that FoxM1 levels modulate the accumulation of aneuploid senescent cells.

**Aneuploidy triggers full senescence following old cell division**. To directly demonstrate that aneuploidy acts as ultimate trigger to elderly cell permanent cycle arrest, and thus the establishment of a full senescent state, we performed the experimental layout shown in Fig. 7g. Octogenarian fibroblasts expressing H2B–GFP were followed under long-term time-lapse microscopy. Daughter cell fate of mitotic cells with chromosome segregation defects (anaphase lagging chromosome and/or micronucleus generation), as well as of mitotic cells without apparent chromosome mis-segregation, was tracked individually during 2–3 days for cell death, cell cycling or senescence (SA-β-gal assay at the end of the movie and immunostaining for 53BP1/p21) (Methods). Using this live-cell fixed-cell correlative microscopy analysis, we found that daughter cells from apparently normal mitoses most often kept on cycling (71.7 ± 0.4%; Fig. 7h, i; Supplementary Movie 9) and cell death was never observed. Among the 28.3 ± 0.4% daughter cells from apparently normal mitoses that stopped cycling, < 25% exhibited SA biomarkers (β-gal + or 53BP1 + /p21 + ) (Fig. 7h, i; Supplementary Movie 10). In contrast, daughter cells from mitoses with mis-segregation events consistently stopped proliferating (100%; Fig. 7h, i). Moreover, when stained for SA-β-gal and 53BP1/p21 SA biomarkers, aneuploid daughter cells were significantly positive (46.4 ± 3.6%, $p < 0.05$ and 85.7 ± 7.1%, $p < 0.0001$, respectively; two-tailed $\chi^2$ test) in comparison with non-cycling daughter cells produced by apparently normal mitoses (Fig. 7h, i; Supplementary Movie 11). Furthermore, other senescence markers (epigenetic histone

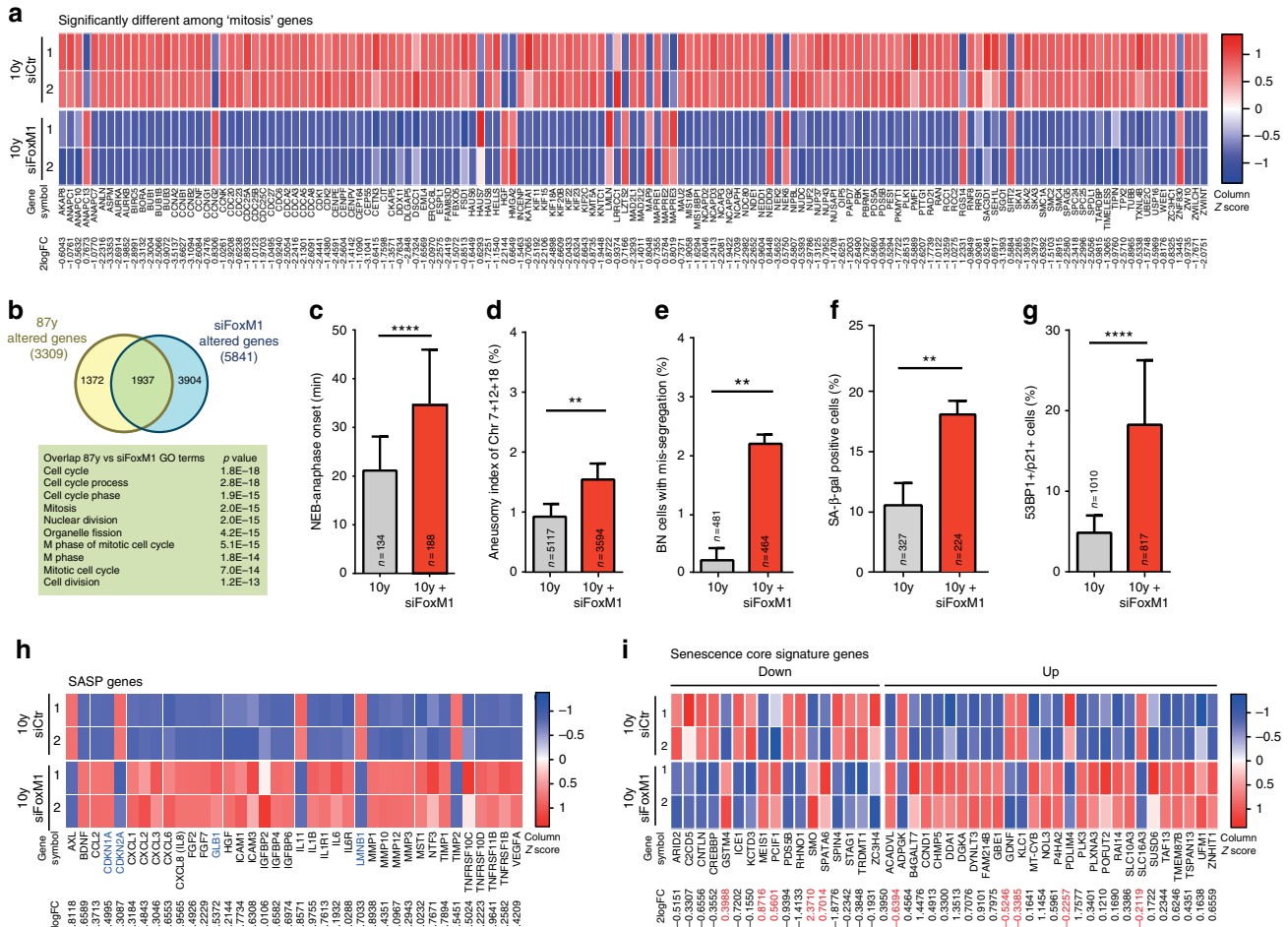

**Fig. 5** FoxM1 repression dictates cellular phenotypes associated with aging. **a** Heatmap of genes within the "mitotis" GO term differentially expressed between control and FoxM1 siRNA-depleted 10-year-old fibroblasts (2logFC cutoff value < −0.5 and > 0.5, *p*-value < 0.05, FDR < 5%, Supplementary Data 8). **b** Venn diagram illustrating the overlap between genes altered in 87 y and FoxM1 siRNA-depleted fibroblasts and the top ten altered GO terms (Supplementary Data 9). **c** Mitotic duration (NEB-anaphase onset) of control and FoxM1 siRNA-depleted fibroblasts. **d** Aneusomy index in interphase FISH. **e** Chromosome mis-segregation rate in CytoD-FISH. **f**, **g** Percentage of cells staining positive for the senescence markers β-galactosidase (**f**) and 53BP1/p21 (**g**). **h** Heatmap of SASP and senescence (*CDKN1A*, *CDKN2A*, *GLB1*, *LMNB1*) genes differentially expressed in HDF 10 y and 10 y siFoxM1 (Supplementary Data 10)[36, 37]. **i** Heatmap of senescence core signature genes differentially expressed in HDF 10 y and 10 y siFoxM1 (Supplementary Data 11); 2logFCs in red indicate genes behaving differently from expected[37]. In all heatmaps: *p*-value < 0.05; genes represented in columns and technical replicates represented in rows; *Z*-score column color intensities representing higher (red) to lower (blue) expression. Values are mean ± SD of three independent experiments. Sample size (*n*) is indicated in each graph. *$p \leq 0.05$, **$p \leq 0.01$, ***$p \leq 0.001$, and ****$p \leq 0.0001$ by two-tailed Mann–Whitney (**c**) and $\chi^2$ (**d**–**g**) statistical tests

modifications, H3K9me3 and H4K20me3[52], and lamin B1 levels[53]) were additionally found altered in elderly cells with segregation defects (presence of micronucleus) vs. elderly cells without segregation defects (Supplementary Fig. 10). Altogether, our data support a model in which proliferating naturally aged cells with SA gene expression signature (early senescence) lose mitotic fidelity, and generate aneuploid progeny that significantly accounts for the accumulation of full senescent phenotypes (permanent cell cycle arrest) (Fig. 8). Reinstating FoxM1 transcriptional activity in elderly cells ameliorates cell autonomous and non-autonomous feedback effects between mitotic fidelity and senescence, suggesting that this mechanism evolved as a positive feedback loop between cellular aging and aneuploidy.

## Discussion

Aging is the largest risk factor for most diseases[54]. In recent years, seminal studies have shown that the accumulation of senescent cells in tissues over time shortens healthy lifespan and that

selective clearance of senescent cells can greatly postpone aging[31,55]. Our results bring insight into how senescent cells arise and provide a molecular basis to delay senescence and thus, a potential strategy against aging and age-associated diseases.

Previous studies have suggested G2 and 4N G1 (mitosis skip) as the phase at which senescent cells exit the cell cycle[56,57]. However, these studies made use of the acute senescence stimuli, gamma irradiation, and oncogenic Ras overexpression, respectively. In contrast to acute senescence, chronic senescence results from long-term, gradual macromolecular damage caused by stresses during lifespan (e.g., loss of proteostasis, DNA damage, epigenetic changes). We demonstrate that in a natural aging cellular model, consisting of dermal fibroblasts retrieved from elderly donors and cultured under limited passage number to prevent replicative in vitro aging, senescent cells are often aneuploid as the result of defective chromosome segregation. Nonetheless, as shown for oncogene-induced senescence and replicative senescence[57,58], one round of cell division is likely required for a transition from early to fully senescent state.

Consistent with this idea of senescence phenotype evolution, our data disclosed the interesting finding that increased pro-inflammatory response is a characteristic of aged cells that are still proliferating, and that chromosome mis-segregation events in these cells are sufficient to trigger transition into permanent cell cycle arrest (full senescence). This is in line with recent findings supporting that micronuclei generated during defective mitoses are a key source of immunostimulatory cytosolic DNA that triggers a cGAS-STING-mediated pro-inflammatory response[59–61]. Whether the evolution of SASP during natural aging is dependent of cytosolic DNA signaling is an interesting question to address in the future.

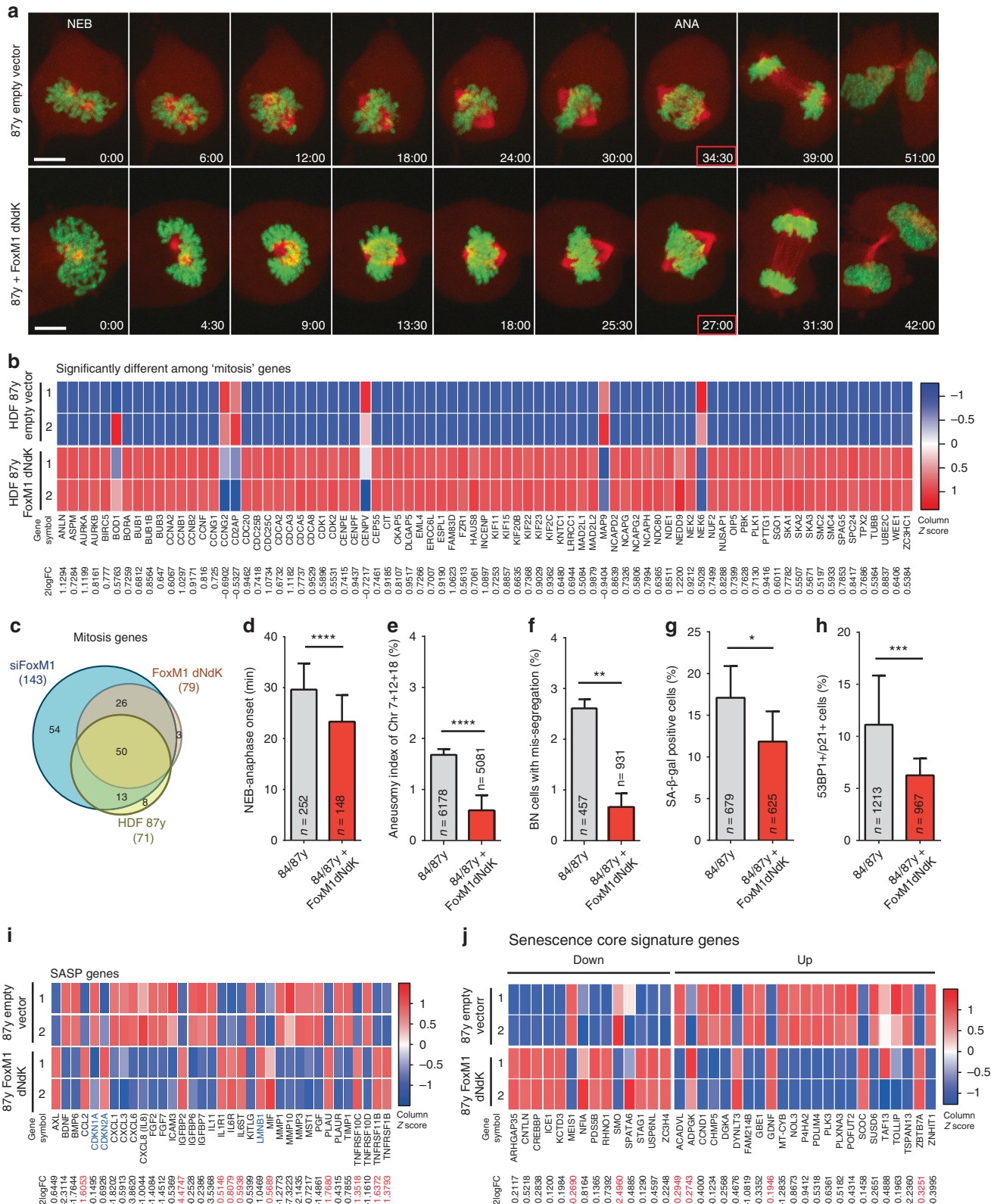

There is still little to no direct evidence demonstrating that aneuploidy per se is a driver or facilitator of aging. In our study we aimed to understand the exact causal network between age-associated aneuploidy and senescence. Using innovative experimental layouts, such as aneuploidy measurement in FACS-sorted senescent cells with high SA-β-gal activity and long-have demonstrated that aneuploidization in an aging cell context ultimately triggers permanent cell cycle arrest and full senescence.

We showed that the frequency of mitotic abnormalities increases in older cells, resulting in mild aneuploidy levels, and identified the FoxM1 transcription factor as the molecular determinant of this age-associated mitotic decline. FoxM1 repression is likely mediated by activation of stress pathways, presumably triggered by primary causes of cellular damage, such as genomic instability or loss of proteostasis. For instance, it has been shown that genotoxic stress can activate the p53-p21-DREAM pathway, which in turn prevents early cell cycle gene expression required for FoxM1 transcriptional activity[39,62]. Although transcriptional deregulation during aging has been widely reported in asynchronous cell populations[63,64], in this study we have used synchronized mitotic populations and found that many cell cycle genes were dysregulated in old cells as compared with young cells. Intriguingly, old mitotic cells have intrinsic low levels of mitotic transcripts. Although this supports the observed mitotic abnormalities and increased chromosome mis-segregation rate (e.g., low levels of major regulators of the error-correction machinery for chromosome attachments), it is still surprising how elderly mitotic cells can cope with such reduced levels of transcripts, e.g., CCNB1. One possibility might be the balanced/stoichiometric repression of most mitotic genes. Furthermore, parallel mechanisms might buffer aging-mediated repression of mitotic transcripts. For example, even though we found SAC genes to be downregulated in elderly cells, SAC functionality was similar in young and old cells, suggesting that concurrent downregulation of genes contributing to proteasome activity (e.g., Cdc20 and APC/C subunits) might buffer SAC gene repression.

We demonstrated that re-establishment of FoxM1 expression in elderly and HGPS fibroblasts can rescue mitotic decline and delay senescence. Fibroblasts are the main constituents of connective tissue and are important factors in extracellular matrix production and tissue homeostasis. Therefore, delayed accumulation of senescent fibroblasts might counteract a SASP-induced inflammatory microenvironment in these tissues and help to protect stem cell and parenchymal cell functions. Re-expression of FoxM1 could thus protect against aging of adult stem cell and post-mitotic tissues. Indeed, increased expression of one FoxM1 transcriptional target, BubR1[39], was previously shown to prevent aneuploidization and delay aging[13]. However, other aneuploidy mouse models have not been reported to exhibit premature aging.

Possible explanations are the premature killing of mice before they start developing aging phenotypes later in life and the cursory analysis for overt age-related degeneration missing tissue-specific phenotypes[14]. Alternatively, aneuploidy-associated genes that are strongly linked with premature aging might require induction of mild levels of aneuploidy or counteracting functions in additional cellular stresses that engage senescence response pathways. This appears to be the case of FoxM1. FoxM1 repression translates into mild aneuploidy levels and may also further act by counteracting age-associated cellular damage caused by genotoxic and oxidative stresses[43,65]. Interestingly, increased FoxM1 expression was shown to improve liver-regenerating capacity in older mice[42] and lung regeneration following injury[66]. However, further evidence shows that FoxM1 is elevated in human cancers, thus questioning its potential for anti-aging therapy. Thus far, FoxM1 has only been shown to be tumorigenic if tumor suppressor genes are lost or oncogenic mutations (such as K-RAsG12D) are present[67,68]. As we found that brief reactivation of FoxM1 activity back to the levels of young cells in elderly cells rescues the observed aging phenotypes, we propose that a cyclic FoxM1 induction scheme, starting in adulthood, could work safely in vivo. FoxM1 induction acts not only through modulation of mitotic fidelity and senescence pro-inflammatory phenotype, but also by improving the proliferative capacity of fitter (undamaged) cells in the elderly cell population that dilute, rather than totally clearing, the full-blown senescent cells, thereby coping with the detrimental but also beneficial (wound healing, tumor suppression) effects of senescence. Thus, our findings disclose a molecular mechanism with potential clinical benefit to healthy lifespan extension and HGPS treatment.

## Methods

**Cell culture**. A total of 11 human fibroblast cultures, established from skin samples of Caucasian males with ages ranging from neonatal to octogenarian (two biological samples per age), were acquired from cell biobanks as summarized (Supplementary Table 1). Several time points over the human lifespan were included to reinforce the validity of any correlation found. All donors were reported as "healthy", except the 8-year-old donor diagnosed with the Hutchison–Gilford progeria. HDFs were seeded at $1 \times 10^4$ cells per $cm^2$ of growth area in minimal essential medium Eagle–Earle (MEM) supplemented with 15% fetal bovine serum (FBS), 2.5 mM L-glutamine, and 1× antibiotic–antimycotic (all from Gibco, Thermo Fisher Scientific, CA, USA). Only early passage dividing fibroblasts (up to passage 3–5) with cumulative population doubling level PDL < 24 were used in all experiments. $PDL = 3.32 \ (\log UCY - \log l) + X$, where UCY = the cell yield at that point, $l$ = the cell number used to begin that subculture, and $X$ = the doubling level of the cells used to initiate the subculture. $PD = T*[\ln 2/\ln(Xe/Xi)]$, where $T$ = time interval (hours) between two consecutive cell passages, Xe = number of cells collected at culture passaging, and Xi = number of cells seeded to initiate subculture. Murine adult fibroblasts (MAFs) were cultured in Dulbecco's modified Eagle's medium (DMEM) supplemented with nutrient mixture F12, 10% FBS, L-glutamine, and antibiotic–antimycotic (all from Gibco). All cells were grown at 37 °C and humidified atmosphere with 5% $CO_2$.

**Fig. 6** Constitutively active FoxM1 ameliorates mitotic fitness and aging markers in elderly cells. **a** Movie frames of elderly dividing cells expressing H2B-GFP/α-Tubulin–mCherry (upper panel) and H2B-GFP/α-Tubulin–mCherry + FoxM1-dNdK (lower panel) (Supplementary Movies 5, 6). Time, min:s. Scale bar, 5 μm. **b** Heatmap of genes within the "mitosis" GO term differentially expressed in mitotic elderly cells transduced with lentiviral empty vector or vector expressing FoxM1-dNdK (2logFC cutoff value < −0.5 and > 0.5, p-value < 0.05, FDR < 5%, Supplementary Data 13). **c** Venn diagram illustrating the overlap between mitosis genes altered in 87 y, 10 y FoxM1 siRNA-depleted, and 87 y FoxM1-dNdK-expressing fibroblasts (Supplementary Data 14). **d** Mitotic duration (NEB to anaphase onset) in elderly fibroblasts infected with control and FoxM1-dNdK lentiviruses. **e** Aneusomy index in interphase FISH. **f** Chromosome mis-segregation rate in CytoD-FISH. **g**, **h** Percentage of cells staining positive for the senescence markers β-galactosidase **g** and 53BPI/p2l **h**. **i** Heatmap of SASP and senescence (CDKN1A, CDKN2A, LMNB1) genes differentially expressed in HDF 87 y and 87 y FoxM1-dNdK (Supplementary Data 15); 2logFCs in red indicate genes behaving differently from expected[36, 37]. **j** Heatmap of senescence core signature genes' expression in 87 y and 87 y FoxM1-dNdK (Supplementary Data 16); 2logFCs in red indicate genes behaving differently from expected[37]. In all heatmaps, p-value < 0.05; genes represented in columns and technical replicates represented in rows; Z-score column color intensities representing higher (red) to lower (blue) expression. Values are mean ± SD of three independent experiments. Sample size (n) is indicated in each graph. *p ≤ 0.05, **p ≤ 0.01, ***p ≤ 0.001, and ****p ≤ 0.0001 by two-tailed Mann–Whitney (**d**) and $\chi^2$ (**e**–**h**) statistical tests

**Isolation of mouse fibroblasts**. Sv/129 mice were housed and handled accordingly to European Union and MAFs were collected from ears of $n > 3$ sv/129 females of the same litter at their age of 8 weeks, 6 months, 1 year, 1.5 years, and 2 years. Ears were washed with phosphate-buffered saline (PBS), cut into small pieces, and incubated with 1 mg/ml collagenase D and 1 mg/ml collagenase/dispase (both from Roche Applied Science, Germany) in DMEM:F12 without FBS, for 45 min at 37 °C and 5% $CO_2$. Cells were then grown on a six-well dish containing DMEM:F12, supplemented with 10% FBS and antibiotic–antimycotic.

**Drug treatments**. Fibroblasts were incubated for 24 h in medium containing 2 μg/ml cytochalasin D (C8273, Sigma-Aldrich, MO, USA) to block cytokinesis. To inhibit the SAC, cells were treated with 5 μM Mps1 inhibitor (AZ3146, TOCRIS, USA). To induce chronic activation of the SAC, cells were treated with 100 nM Taxol (Sigma-Aldrich, MO, USA) and 5 μM and STLC (2799-07-7, Tocris). For proteasome activity enhancement, 10 μM of Usp14 inhibitor were used (I-300, BostonBiochem, Cambridge, MA). To inhibit kinesin-5, STLC (Tocris) was used at 5 μM during 16 h, to enrich the MI for mitotic cell shake-off.

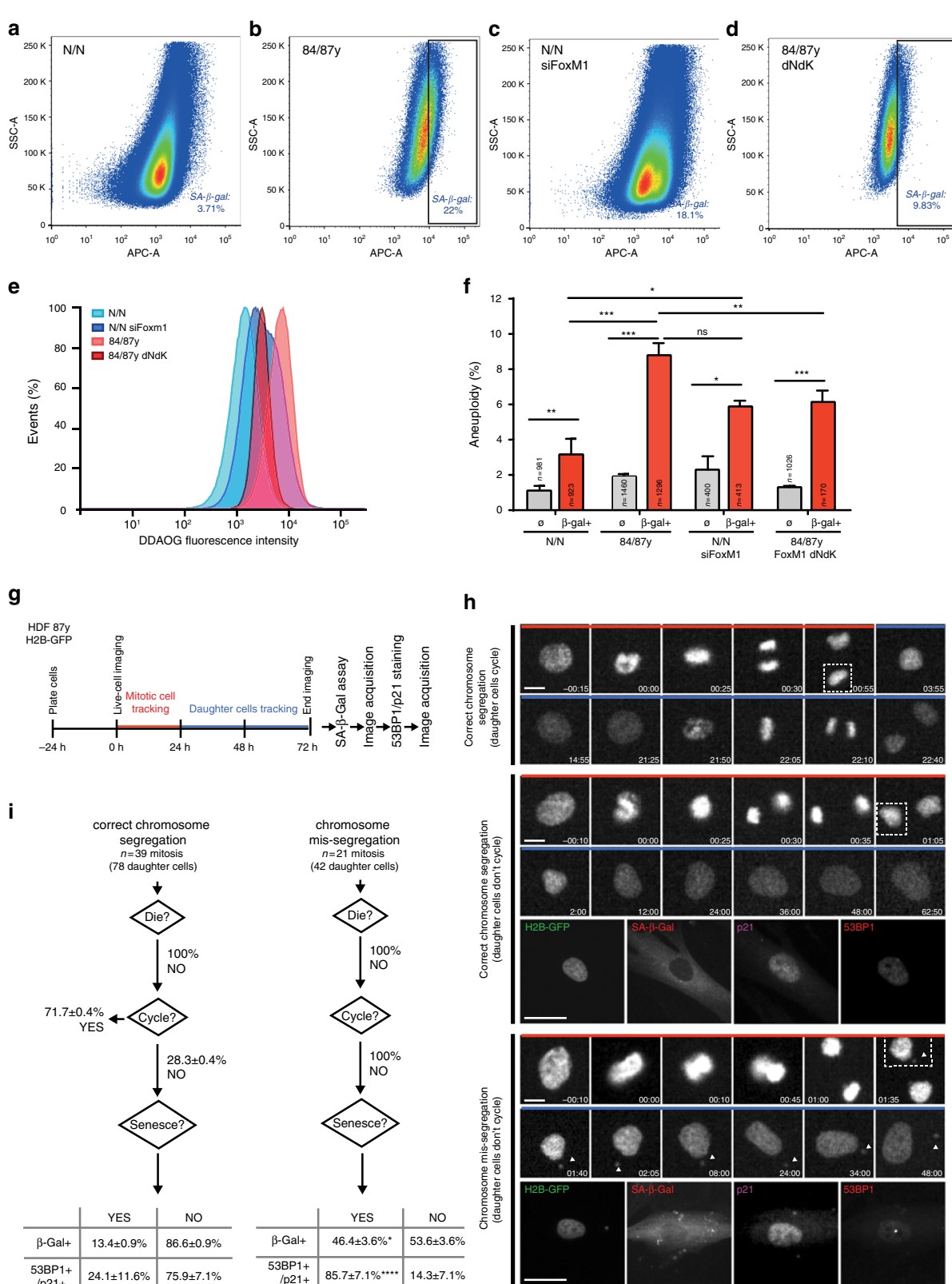

**Fluorescence in situ hybridization**. FISH analysis was used to measure aneusomy index (Interphase FISH; Figs. 1a, 5d, 6e, 7f and Supplementary Fig. 9e) and chromosome mis-segregation rate (CytoD FISH; Figs. 1b, 5e, 6f and Supplementary Fig. 9f). "Interphase FISH" measures the prevalence of somatic aneuploidy by the ratio of aneusomic cells for chromosomes 7, 12, and 18 to the total cell count for a sample (> 2500 nuclei) (aneusomy index). "CytoD FISH" measures the rate of chromosome mis-segregation (number of events in which two sister chromatids co-segregate to the same daughter cell) by combining a cytokinesis-block assay (cytochalasin D 24 h treatment) with FISH staining. "CytoD FISH" measures the percentage of BN telophases exhibiting chromosome 7, 12, and 18 mis-segregation over the total BN cell count for a sample (> 200 BN cells). For both Interphase and CytoD FISH, fibroblasts were grown on Superfrost™ Plus microscope slides (Menzel, Thermo Scientific, CA, USA) placed in a quadriperm dish (Sarsted, Nümbrecht, Germany). Cells were given a hypotonic shock during 30 min (0.03 M sodium citrate, Sigma-Aldrich, MO, USA), followed by fixation in ice-cold Carnoy fixative added drop-wise and incubated for 5 min. This step was repeated two more times. FISH was performed with the Vysis centromeric probes CEP7 Spectrum Aqua, CEP12 Spectrum Green, and CEP18 Spectrum Orange (Abbott Laboratories, Chicago, IL, USA) according to manufacturer's instructions. Slides were mounted with mounting medium containing DAPI (Vectashield, Vector Laboratories, CA, USA) and scored blindly.

**Apoptosis assay**. Programmed cell death (apoptosis) was evaluated by flow cytometry using fluorescein isothiocyanate (FITC)-conjugated Annexin V/Apoptosis detection kit (BioLegend, Inc. San Diego, CA). Briefly, cells were washed twice in cold cell staining buffer and resuspended in 100 μL of Annexin V-binding buffer. Subsequently, 5 μL of FITC-Annexin V were added to cell suspension. Cells were incubated for 15 min in the dark, washed with Annexin V-binding buffer, and analyzed immediately by flow cytometry.

**SA-β-gal assay**. In fixed-cell analysis, cells were incubated for 90 min in medium with 100 nM Bafilomycin A1 (B1793, Sigma-Aldrich, MO, USA) to induce lysosomal alkalinization. Fluorogenic substrate (33 μM) for β-galactosidase, fluorescein di-β-D-galactopyranoside (F2756, Sigma-Aldrich, MO, USA) were then added to the medium, and incubation carried out for 90 min. Cells were fixed in 4% paraformaldehyde for 15 min, rinsed with PBS, and permeabilized with 0.1% Triton-X100 in PBS for 15 min. Finally, cells were counterstained with 1 μg/ml DAPI (Sigma-Aldrich, MO, USA). For FACS, cells were incubated in Bafilomycin A1 as described above and then exposed to 10 μM of fluorogenic substrate for β-galactosidase, DDAOG (9H-(1,3-dichloro-9,9-dimethylacridin-2-one-7-yl) β-D-Galactopyranoside), for 90 min (Setareh Biotech LLC, USA).

**Fluorescence-activated cell sorting**. FACS sorting was used to isolate subpopulations of senescent (SA-β-gal positive) live fibroblasts. FACS sorting was performed in FACSAria™ I Cell Sorter (BD Biosciences, CA, USA), using the laser line of 633 nm. All cells within a single experiment were detected using the same voltage settings and sorted using an 85 μm nozzle. Cells were initially gated by forward scatter area (FSC-A) vs. side scatter area, which excludes dead cells and subcellular debris, with subsequent exclusion of cell doublets and clumps through FSC-A vs. FSC-width plot. The signal was detected using the APC-A channel. The relative β-galactosidase activity was inferred from the median fluorescence intensity of the population. The sorting gates were designed accordingly to the respective auto-fluorescent control. Cells were sorted directly into MEM with 15% FBS and seeded in Superfrost™ Plus microscope slides for subsequent FISH analysis. Analysis of FACS data was done using FlowJo v10 software (TreeStar, Inc., Ashland, OR).

**Immunostaining**. Fibroblasts were grown on sterilized glass coverslips coated with 50 μg/ml fibronectin (F1141, Sigma-Aldrich, MO, USA). Cells were fixed in freshly prepared 4% paraformaldehyde in PBS for 20 min. Following fixation, cells were rinsed in PBS and permeabilized in PBS + 0.3% Triton-X100 for 7 min. Cells were next blocked in 10% FBS in PBS-T (PBS + 0.05% Tween-20) for 1 h and then incubated overnight at 4 °C with primary antibodies diluted in PBS-T + 5% FBS as follows: rabbit anti-53BP1 (4937, Cell Signaling Technology, MA, USA), 1:100; mouse anti-p21 (SC-6246, Santa Cruz Biotechnology, CA, USA), 1:1000; rabbit anti-FoxM1 (13147, ProteinTech Group, Inc., IL, USA), 1:1500; mouse anti-Cdc20 (SC-5296, Santa Cruz Biotechnology), 1:100; mouse anti-Cyclin B (SC-245, Santa Cruz Biotechnology), 1:1000 secondary antibodies AlexaFluor-488 and -568 (Life Technologies, CA, USA) were diluted 1:1500 in PBS-T + 5% FBS. DNA was counterstained with 1 μg/ml DAPI (Sigma-Aldrich, MO, USA). Coverslips were mounted in slides with mounting solution (90% glycerol, 0.5% N-propyl-gallate, 20 nM Tris, pH 8).

**Telomere PNA FISH**. Fibroblasts were incubated in 0.05 μg/ml colcemid (15212012, Gibco, Thermo Fisher Scientific) for 4 h, to induce metaphase arrest. Following trypsinization, fibroblasts were incubated in 0.03 M sodium citrate for 30 min at 37 °C. Cells were then fixed in freshly made Carnoy fixative solution and stored at 4 °C. Metaphase spreads were fixed with 4% formaldehyde in PBS for 2 min, followed by pepsin digestion (1 mg/ml) for 10 min at 37 °C. After a dehydration step with ethanol, dried slides were hybridized with Telomere PNA probe (Applied Biosystems, CA, USA) (0.5 μg/ml) in 10 mM Tris pH 7.5, 70% formamide, 0.25% blocking reagent (Roche, Germany), 2 mM MgCl$_2$, 700 μM citric acid, 7 mM Na$_2$HPO$_4$, for 3 min in a hot plate at 80 °C and then for 2 h at 37 °C in humidified chamber. Slides were washed in 70% formamide, 10 mM Tris, 0.1% bovine serum albumin twice for 15 min, then washed three times in Tris-buffered saline (TBS) for 5 min, and finally mounted with mounting media containing DAPI (Vectashield, Vector Laboratories).

**Phase-contrast live-cell imaging**. Fibroblasts were grown in glass-bottom 35 mm μ-dishes (Ibidi GmbH, Germany), coated with 50 μg/ml fibronectin (F1141, Sigma-Aldrich, MO, USA). Images were acquired on a Zeiss Axiovert 200 M inverted microscope (Carl Zeiss, Oberkochen, Germany) equipped with a CoolSnap camera (Photometrics, Tucson, USA), XY motorized stage and NanoPiezo Z stage, under controlled temperature, atmosphere, and humidity. Neighbor fields (20–25) were imaged every 2.5 min for 2–3 days, using a ×20/0.3 numerical aperture (NA) A-Plan objective. Stitching of neighboring fields was done using the plugin "Stitch Grid" (Stephan Preibisch) from ImageJ/Fiji software.

**Live-cell and fixed-cell correlative microscopy**. Eighty-seven-year-old fibroblasts expressing H2B–GFP were grown in μ-Slide 2 Well ibiTreat (Ibidi GmbH) and analyzed accordingly to the experimental layout shown in Fig. 7i. Images were acquired on a Leica DMI6000b inverted microscope (Leica Microsystems, Germany), equipped with an ORCA-Flash4.0 camera (Hamamatsu, Japan), using the laser line 488 nm for the excitation of green fluorescent protein (GFP), and under controlled temperature, atmosphere and humidity. Neighbor fields (60–70) were imaged every 5 min for 3–4 days, using a ×20 LD/0.4 NA objective (Leica Microsystems). Following the long-term live-cell imaging, cells were processed for SA-β-gal assay and 53BP1/p21 double immunostaining as described elsewhere in this section, and the same neighbor fields acquired using a ×40 LD/0.6 NA objective. LAS X software (Leica Microsystems) was used for image acquisition and analysis.

**Spinning-disk confocal microscopy**. Four-dimensional data sets were collected with Andor Revolution XD spinning-disk confocal system (Andor Technology, Belfast, UK), equipped with an electron-multiplying charge-coupled device

**Fig. 7** FoxM1 governs aneuploidization-driven cellular senescence in elderly cells. **a–d** FACS sorting of senescent cells from neonatal **a**, elderly **b**, FoxM1 siRNA-depleted neonatal **c**, and 84 y/87 y with FoxM1-dNdK **d** cell populations with high β-galactosidase activity. The gates were defined accordingly to the respective auto-fluorescent control. **e** Relative intensity levels of the fluorogenic substrate DDAOG in the sorted cell populations. **f** Aneuploidy index in FACS-sorted β-gal-positive fibroblast subpopulations (β-gal +) vs. unsorted populations (∅) as determined by FISH analysis for three chromosome pairs. **g** Experimental layout of live-cell/fixed-cell correlative microscopy analysis shown in **h**, **i**. Mitotic elderly fibroblasts expressing H2B–GFP and respective daughter cells were imaged for 72 h. SA-β-Gal assay and immunostaining for 53BP1/p21 were performed at the end of imaging. **h** Movie frames of representative phenotypes observed for elderly cells expressing H2B–GFP. Top panel, correct chromosome segregation, with cycling daughter cells (Supplementary Movie 9). Middle panel, correct chromosome segregation with non-cycling daughter cells staining negative for SA-β-Gal and 53BP1/p21 (Supplementary Movie 10). Bottom panel, chromosome mis-segregation leading to micronuclei formation (arrowheads), with non-cycling daughter cells staining positive for senescence markers (SA-β-Gal and 53BP1/p21) (Supplementary Movie 11). Dashed line indicates the tracked daughter cell. Time, hour:minute. Scale bar, 30 μm (movie frames) and 15 μm (immunostaining). **i** Single-cell analysis of daughter cell fate (cell death, cell cycle arrest, and cell senescence) from mitoses with apparent correct chromosome segregation or with mis-segregation (leading to micronuclei formation). Non-cycling daughter cells were stained for senescence markers (β-gal and 53BP1/p21). Values are mean ± SD of two independent experiments. Sample size (n) is indicated for **f** and **i**. NS, $p > 0.05$, *$p \leq 0.05$, **$p \leq 0.01$, ***$p \leq 0.001$, and ****$p \leq 0.0001$ by two-tailed $\chi^2$ statistical tests

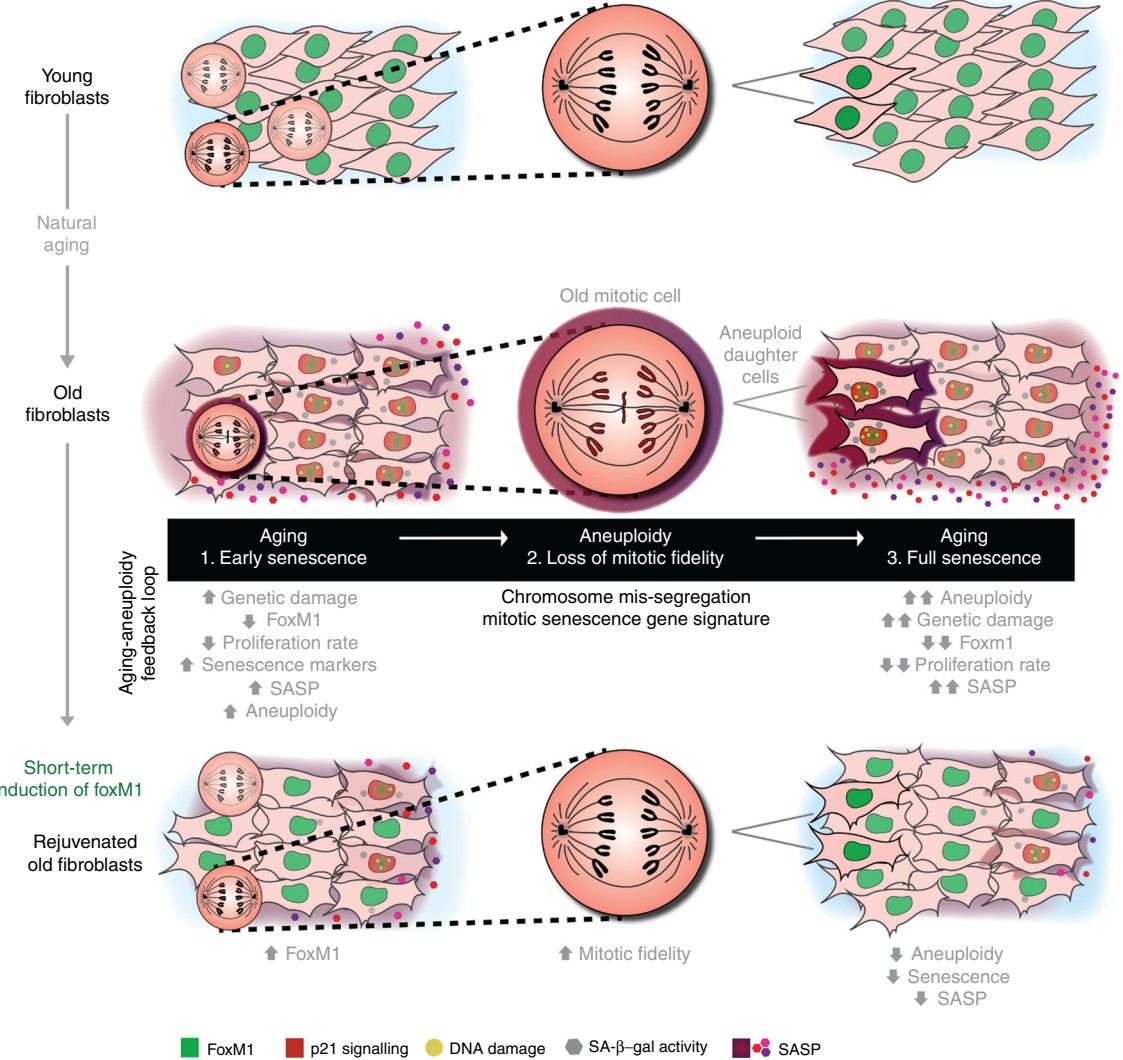

**Fig. 8** Positive feedback loop between aging and aneuploidy. Model summarizing the molecular basis of age-associated mitotic decline and accumulation of senescent cells induced by aneuploidy. FoxM1 repression in aged dividing cells with senescence gene signature (1. early senescence) leads to general dysfunction of the mitotic machinery and increased chromosome mis-segregation rate (2. loss of mitotic fidelity). The aneuploid progeny stops proliferating exhibiting aggravated senescence phenotypes (3. full senescence) with pro-inflammatory impact in neighboring pre-senescent cells. FoxM1 induction in old cells improves mitotic fitness and delays the accumulation of senescent cells

iXonEM Camera and a Yokogawa CSU 22 unit based on an Olympus IX81 inverted microscope (Olympus, Southend-on-Sea, UK). Two laser lines at 488 and 561 nm were used for the excitation of GFP and mCherry and the system was driven by Andor IQ software. Z-stacks (0.8–1.0 μm) covering the entire volume of the mitotic cells were collected every 1.5 min with a PlanApo ×60/1.4 NA objective. All images represent maximum-intensity projections of all z-planes. ImageJ/Fiji software was used to edit the movies.

**Fluorescence microscopy**. Analysis of the SA-β-gal fluorescence assay was carried out on a Zeiss AxioImager Z1 (Carl Zeiss) equipped with an Axiocam MR and using an EC-Plan-Neofluor ×40/1.3 NA objective. Cells displaying > 5 fluorescent granules were considered positive for SA-β-gal activity. Image acquisition of FoxM1, Cyclin B, and Cdc20 immunostaining was performed on a Zeiss AxioImager using a Plan-Apochromat ×63/1.4 NA objective. Prometaphase cells were acquired with 0.24 μM Z-stacks. Image acquisition of H4K20me3, H3K9me3, and Lamin B1 immunostaining was performed on Leica 6000B using a ×40 LD/0.6 NA objective. LAS X software (Leica Microsystems) was used for image acquisition and cells were acquired with 0.44 μM Z-stacks. User-defined fluorescence intensity thresholds were set and used consistently for samples within each experiment. AutoQuant X2 (Media Cybernetics, Rockville, USA) was used for image deconvolution.

**Automated microscopy**. Image fields of 53BP1/p21 double immunostaining and FISH staining were acquired on IN Cell Analyzer 2000 (GE Healthcare, UK),

equipped with a Photometrics CoolSNAP K4 camera and using a Nikon ×20/0.45 NA Plan Fluor objective. User-defined fluorescence intensity thresholds were set and used consistently for samples within each experiment.

**Image analysis**. Live-cell phenotypes (mitotic duration, spindle positioning, and cytokinesis failure) and fixed-cell experiments (FISH, SA biomarkers, and micronuclei) were blindly quantified using ImageJ/Fiji software.

**Western blotting**. Mitotic cell populations were collected by shaking-off cell culture flasks enriched for MI by a 16 h treatment with STLC. MI > 95% was determined by visual scoring of cells with condensed chromosomes after Carnoy fixation, followed with 1 μg/ml DAPI in PBS. Lysis buffer (150 nM NaCl, 10 nM Tris-HCl pH 7.4, 1 nM EDTA, 1 nM EGTA, 0.5% IGEPAL) with protease inhibitors was added to mitotic cell pellets, and lysates quantified for protein content by the Lowry Method (DC™ Protein Assay, Bio-Rad, CA, USA). Twenty micrograms of total extract were then loaded in SDS-polyacrylamide gel electrophoresis gels and transferred onto nitrocellulose membranes for western blot analysis. Membranes were blocked during 1 h with TBS containing 5% low-fat milk. Primary antibodies were diluted in TBS containing 2% low-fat milk as follows: rabbit anti-FoxM1 (13147, ProteinTech Group, Inc.), 1:1000; mouse anti-Cyclin B (SC-245, Santa Cruz Biotechnology), 1:1000, mouse anti-Ndc80 (clone 9G3, Abcam, Cambridge, UK), 1:1750; mouse anti-Plk1 (SC-17783, Santa Cruz Biotechnology), 1:100; mouse anti α-tubulin (T5168, Sigma-Aldrich, CA, USA), 1:50,000; and mouse anti-GAPDH (60004, ProteinTech Group, Inc.), 1:30,000. Goat anti-rabbit (SC-2004,

Santa Cruz Biotechnology) and goat anti-mouse (SC-2005, Santa Cruz Biotechnology) horseradish peroxidase-conjugated secondary antibodies were diluted at 1:3000 in TBS containing 2% low-fat milk. Signal was detected using Clarity Western ECL Substrate reagent (Bio-Rad Laboratories, CA, USA) according to manufacturer's instructions. A GS-800 calibrated densitometer with Quantity One 1-D Analysis Software 4.6 (Bio-Rad Laboratories) was used for quantitative analysis of protein levels. Uncropped scans of all blots in the main manuscript and supplementary figures are provided in Supplementary Fig. 11.

**Lentiviral plasmids**. H2B–GFP[69] was amplified as a BglII–H2B–GFP–T2A–BamHI–NotI fragment. This PCR fragment was digested with BglII + NotI and ligated into pRetroX-Tight-Puro (Clontech, CA, USA) digested with BamHI + NotI, thus destroying the 5′ BamHI/BglII site, while reintroducing a BamHI site 3′ of H2B–GFP–T2A. In parallel, α-Tubulin[69] was amplified as a BglII–α-Tubulin–BamHI–NotI fragment and ligated into pRetrox-Tight-Puro digested with BamHI + NotI, again destroying the 5′ BamHI/BglII site, while reintroducing a BamHI site 3′ of α-Tubulin. mCherry was amplified from pExchange-1-Cherry (Agilent Technologies, CA, USA) as a BglII–mCherry–NotI fragment and was ligated into pRetrox–α-Tubulin digested with BamHI–NotI, yielding pRetrox–α-Tubulin–mCherry. Finally, α-Tubulin–mCherry was PCR-amplified as a BglII–α-Tubulin–mCherry–NotI fragment and ligated into pRetrox–H2B–GFP–T2A digested with BamHI + NotI, yielding pRetrox–H2B–GFP–T2A–α-Tubulin–mCherry. To obtain pLVX–Tight-Puro–H2B–GFP–T2A–α-tubulin–mCherry, a BglII–H2B–GFP–T2A–α-Tubulin–mCherry–NotI fragment was amplified from pRetroX–H2B–T2A–GFP–α-Tubulin–mCherry and ligated into pLVX–Tight-Puro (Clontech) digested with BamHI + NotI. To generate pLVX–Tight-Puro–FoxM1-dNdK, a BglII–FOXM1–dNdK–NotI fragment was amplified from pcDNA3–Flag-ΔN-ΔKEN-FoxM1[45] and ligated into pLVX–Tight-Puro digested with BamHI + NotI. Primers used are described in Supplementary Table 4.

**Lentiviral production and infection**. Lentiviruses were produced according to the protocol described in Lenti-X Tet-ON Advanced Inducible Expression System (Clontech). Lentiviruses carrying empty pLVX–Tight-Puro, pLVX––Tight-Puro–H2B–GFP–α-Tubulin–mCherry or pLVX–Tight-Puro–FoxM1-dNdK, as well as lentiviruses carrying pLVX–Tet-On Advanced (which expresses rtTA), were generated in HEK293T helper cells transfected with packaging plasmids (pMd2.G and psPAX2) using Lipofectamine 2000 (Life Technologies, Thermo Scientific, CA, USA). Human fibroblasts were co-infected for 12–16 h with responsive and transactivator lentiviruses at 2:1 ratio, in the presence of 8 μg/ml polybrene (AL-118, Sigma-Aldrich, MO, USA). In the following day, 750 ng/ml doxycycline (D9891, Sigma-Aldrich, MO, USA) was added to the medium to induce co-transduction. Phenotypes were analyzed and quantified 48–72 h later, and transfection efficiency monitored by scoring the number of fluorescent cells or by western blotting.

**FoxM1 RNA interference**. Cells were transfected 1 h after plating, with 45 nM FoxM1 small interfering RNA (SASI_Hs01_00243977 from Sigma-Aldrich, MO, USA) using Lipofectamine RNAiMAX (Thermo Scientific) according to manufacturer's instructions. Phenotypes were analyzed and quantified 72 h post transfection, and depletion efficiency monitored by western blotting.

**Real-time PCR**. Total RNA was obtained from mitotic fibroblasts (collected as depicted in Fig. 3a and in the Western blotting section) using the RNeasy® Mini kit (Qiagen, Hilden, Germany). RNA purity and integrity was confirmed in the Experion™ system (Bio-Rad Laboratories). cDNA was synthesized from total RNA (1 μg) using iScript Advanced Select cDNA Synthesis kit (Bio-Rad Laboratories). The $2^{-\Delta\Delta Ct}$ method was used to quantify the transcript levels of FOXM1, CDKN1A, MMP1, CXCL8, TSPAN13, and FAM214B against the transcript levels of the housekeeping gene (TBP). Primers were designed to span at least one exon–intron junction (Supplementary Table 3). Amplification was performed in a C1000 Touch Thermal Cycler (CFX384 Real-Time System, Bio-Rad Laboratories) and analyzed using CFX Manager Software (Bio-Rad Laboratories).

**RNA-seq and bioinformatics**. RNA was isolated from mitotic human fibroblasts and validated as described above for Real-time PCR. RNA-sequencing libraries were prepared using TruSeq Stranded Total RNA with Ribo-Zero Human/Mouse/Rat (RS-122-2201; Illumina, CA, USA) according to manufacturer's protocol. Pooled libraries were sequenced on an Illumina HiSeq 2500 (single-end 50 bp). Raw unaligned sequencing reads (fastq-format) were deposited in the European Nucleotide Archive (ENA) under the accession number PRJEB27047. Reads were aligned to the human genome (hg19) using a splicing-aware aligner (StarAligner). Aligned reads were fragments per million (FPM) normalized, excluding low abundance genes (mean FPM > 1). Differential gene expression analysis was performed using a likelihood ratio test constructed under the R-package edgeR (v3.14.0 available from Bioconductor at http://www.bioconductor.org/packages/release/bioc/html/edgeR.html). Significant differential gene expression of aged fibroblasts was defined as p-value < 0.05 and 2logFC cutoff value < −0.5 or > 0.5; and in case of genetic manipulations as p-value < 0.05, 2logFC cutoff value < −0.5

or > 0.5, and FDR < 0.05. For generation of RNA-seq heatmaps, transcript counts were normalized to the sample library size. These values were subsequently scaled by gene using normalized scores or z-scores (i.e. a value of 0 corresponds to the mean gene expression of that gene across all libraries, and ± 1, ± 2, etc. represent 1, 2, etc. SDs from the gene mean). Hierarchical clustering was performed using the R library heatmap.2. RNA-seq data represent two independent experimental replicates of each biological sample. GO term enrichment analysis of differentially expressed genes was performed using Database for Annotation, Visualization and Integrated Discovery Functional Annotation Tool v6.7[70] (www.david-d.ncifcrf.gov). To calculate the significance of gene set enrichment, empirical p-values were generated using DAVID tool.

**Statistical analysis**. Sample sizes and statistical tests for each experiment are indicated in the fugure legends. p-values were obtained using GraphPad Prism version 6 (GraphPad, San Diego, CA, USA). Data were tested for parametric vs. non-parametric distribution using D'Agostino–Pearson omnibus normality test. Spearman's rank correlation, Mann–Whitney, two-tailed $\chi^2$, or one-way analysis of variance for multiple comparisons tests were then applied accordingly. NS, $p > 0.05$, $*p \leq 0.05$, $**p \leq 0.01$, $***p \leq 0.001$, and $****p \leq 0.0001$. Values are shown as mean ± SD or mean ± SEM.

**Data availability**. Raw unaligned sequencing reads (fastq-format) that support the findings of this study have been deposited in the ENA and are available under the accession number PRJEB27047. The authors declare that data supporting the findings of this study are available in the main manuscript and supplemental information, or otherwise available from the corresponding author upon request.

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

## Acknowledgements

We thank H. Maiato, J.M. Cabral, and J. Bessa at i3S for critical discussions and reading of the manuscript; P. Sampaio and A. Maia for technical help with Advanced Light Microscopy. E.L. holds an FCT Investigator Postdoctoral Grant (IF/00916/2014) from FCT/MCTES (Fundação para a Ciência e a Tecnologia/Ministério da Ciência, Tecnologia e Ensino Superior). FCT Fellowships (SFRH/BD/74002/2010; SFRH/BD/125017/2016; PD/BD/128000/2016) supported J.C.M., S.V., and R.R. The following project grants supported this work: National Funds through FCT under the project PTDC/BEX-BCM/2090/2014 ; NORTE-01-0145-FEDER-000029 funded by North Regional Operational Program (NORTE2020) under PORTUGAL 2020 Partnership Agreement through Regional Development Fund (FEDER); NORTE-07-0124-FEDER-000003 co-funded by North Regional Operational Program (ON.2) through FEDER and by FCT; and POCI-01-0145-FEDER-007274 i3S framework project co-funded by COMPETE 2020/PORTUGAL 2020 through FEDER and by FCT; Foundation Pediatric Oncology Groningen grant and Dutch Cancer Society grant 2012-RUG-5549 to F.F.

## Author contributions

J.C.M. initiated the project. J.C.M, S.V., and R.R. designed and performed the experiments, and analyzed the data. B.B. and P.B. performed RNA-sequencing and bioinformatics analysis. J.M.E. performed telomeric-FISH analysis. M.G.F., R.M., and F.F. contributed to the study design and edited the manuscript. E.L. conceived the idea,

supervised the work, and wrote the manuscript. All authors discussed results, prepared figures, and edited the manuscript.

## Additional information

**Competing interests:** The authors declare no competing interests.

