## [Peer Review File · Nature Communications]

Reviewers' comments:

Reviewer #1 (Remarks to the Author):

In humans, chronological aging is associated with the development of chronic diseases and declining overall health. Aging is not only a chronological process but also a biological process, and the consequences of aging are in part attributed to altered cell physiology. One cellular hallmark of aging is an increase in aneuploidy. Aneuploidy is a state of abnormal chromosome numbers that is caused by chromosome segregation errors occurring during cell division. Further, aneuploidy is also linked to aging with elevated rates of aneuploidy leading to the onset of age-associated phenotypes. Thus, understanding the relationship between aneuploidy and cellular aging is an important topic in aging-related research.

Although previous studies have demonstrated an inter-dependent relationship between aneuploidy and aging, the molecular basis of this relationship remains unknown. These authors investigate the relationship between aneuploidy and aging by comparing human dermal fibroblasts (HDF) cells collected from participants ranging in age from neonatal to octogenarian. Using a variety of approaches, including fluorescent in-situ hybridization (FISH), immunofluorescent microscopy, and live-cell imaging, the authors demonstrate that aged cells exhibit an increase in mitotic errors and aneuploidy. In addition, using RNA-sequencing to compare the transcriptomes of young and old cells, the authors show that aged cells exhibit a down-regulation of mitotic cell cycle genes and an increase in senescence-associated secretory phenotype (SASP) genes. The down-regulation of mitotic cell cycle genes is shown to be a consequence of decreased mRNA and protein levels of the transcription factor FoxM1 that drives the transcription of many G2/M genes. Importantly, the authors demonstrate that siRNA-mediated repression of FoxM1 in young cells increases mitotic errors and aneuploidy. Moreover, they show that over-expression of a constitutively active FoxM1 decreases mitotic errors and aneuploidy in aged cells. This is a very powerful experiment. Lastly, the authors show that the population of senescent aged cells has an increase in the percent of aneuploid cells compared to the population of young senescent cells.

In conclusion, the authors propose a positive feedback loop between aging and aneuploidy with aged cells displaying a SASP phenotype and increased mitotic errors that generates aneuploid progeny and in turn aneuploid progeny enter full senescence causing age-related pathologies. Overall these studies provide evidence that a loss of mitotic fidelity and an increase in aneuploidy due to down-regulation of FoxM1 in aged cells is the molecular cause of increased senescent cells during natural aging and that restoring mitotic fidelity is an anti-aging therapeutic strategy. This manuscript provides novel data on the molecular basis of aging, and I am strongly supportive of publication provided that the authors address some specific concerns.

Major Comments:

1. The authors should provide a quantification of mitotic index for the young vs. old-aged cells in order to compare differences in proliferation rate between the two cell types.

2. In the manuscript, could the authors please provide a further explanation for their aneusomy index calculation (Figures 1B, 1C, and 6F)? Does this index include tetraploid cells? In Figure 2, the authors demonstrate that there is a significant population of aged cells that fail cytokinesis. Are these cells included in the aneusomy index calculation? Also, rather than combining the data for all three chromosome probes, it would be informative to provide the data for each probe

separately (plotted as a deviation from the mode). In particular, the authors extrapolate the data to all 23 chromosome pairs to determine overall levels of aneuploidy. This calculation relies on the assumption that all chromosomes have an equal probability of contributing to aneuploid progeny.

3. In the text, the authors state that they observe higher levels of aneuploidy using a cytokinesis block assay. A cytokinesis block assay does not measure aneuploidy levels, but is a measure of the rate of chromosome mis-segregation. This is an important distinction as cells can mis-segregate chromosomes but fail to proliferate further with an aneuploid genome. A higher rate of chromosome mis-segregation is not strictly equivalent to higher levels of aneuploidy. The text needs to be revised to reflect what the assay is measuring.

4. Could the authors provide an explanation for why there is an observed decrease in aneuploidy for the young cells in Figure 1C (vs Figure 1B)?

5. Could the authors provide quantifications for chromosome alignment defects and lagging chromosome phenotypes depicted in Figure 2B and 2C?

6. In figure 2B, the authors show images from a live-cell movie of an aged cell with unaligned chromosomes. It is unclear from the text description if aged cells are taking longer to achieve chromosome congression but eventually do or if aged cells are entering anaphase with unaligned chromosomes. The distinction is important because in the former scenario the SAC is functioning properly, but in the latter the SAC is not.

7. It is unclear what the authors are trying to show in the image for figure 3B. What cell type is being imaged? Are we looking at mitotic chromosome spreads? If the point of this figure is to show that the mitotic indices are the same in both the young and aged cell populations used in the RNA-seq experiment, then both a neonatal and 87yo representative image should be displayed. Some indication of magnification should be provided in the figure legend for the insets.

8. The RNA-seq data is described as showing two biological replicates, but in the methods, it states that the two biological replicates came from a single experiment. If so, then these are not biological replicates, but technical replicates. The gene expression data should be averaged across the two technical replicates and not reported as separate biological replicates. This is necessary to identify the significant changes in gene expression between young and aged cells.

9. The authors provide evidence suggesting that decreased mRNA expression levels of mitotic genes cause a loss of mitotic fidelity in aged cells. To strengthen this conclusion, it is necessary to demonstrate that decreased mRNA levels are leading to decreased protein levels in aged vs. young cells. Western blot data for some of the candidate mitotic genes is needed to validate the results from the RNA-seq experiment in figure 3.

10. In Figures 4D and 5A, could the authors provide a biological explanation for why a specific subset of mitotic transcripts is upregulated in the FOXM1 KD condition and downregulated in the FOXM1 over-expression condition, respectively?

11. In supplementary figure 3A, the authors observe a significant increase in multitelomeric signal with age. Could the authors define what a multitelomeric signal is and better explain how this increase is not contributing to the mitotic defects observed in the aged cells?

12. Could the authors provide a table or figure illustrating overlap between a) the transcripts that were downregulated in the aged cells (relative to the young cells-Figure 3) and b) the transcripts that were downregulated in the FOXM1 KD cells (Figure 4)?

13. In the text, the authors discuss a feedback loop between cellular aging and aneuploidy. A diagram depicting the feedback loop would be a helpful addition at the end of this manuscript to

clarify the steps in this loop. Additionally, it may be helpful to further define the difference between an "early senescent phenotype" and a "full senescent phenotype" (lines 238-239).

Minor Comments:

1. In the introduction, it might be beneficial for the authors to define the difference between biological age and pathological age (referenced in line 87).

2. The authors measure spindle mis-positioning as a read-out for general mitotic dysfunction, but it is unclear how spindle positioning with regards to the substrate surface contributes to mitotic errors. Could the authors please add in-text references for the contribution of spindle mis-positioning to mitotic errors?

3. In supp figure 1B, could the authors explain why it is important to quantify cell area? Is this a readout for senescence?

4. Please reference all supplemental videos in the text.

5. Overall, additional details are required in the figure legends. In particular:

a. The gray box in supplemental Figure 6E is unlabeled.

b. Line 764: Please reference Supp Fig 4 in Figure legend 2E

c. Please provide magnification for the insets in figures 3B and sup figure 1C.

d. Time on representative images in supp figure 4A should be displayed as "Time hr:min" (Line 902). Additionally, the length of scale bar for supp fig 4A should be defined.

e. What cell type is used for the representative images in supp figures 1C and 1E?

f. Supp Fig 5G: Does the FOXM1 dNdK protein run as a doublet? Is the non-specific band in both lanes? The figure legend also says the control cells are "untransduced" when the figure itself implies that they were transduced with the vector.

g. In Supp Fig 5H-J, were these quantifications done in both 84 and 87 year old donor cells?

6. The mitotic indices for supp figures 5A and 5F should be quantified.

7. In Figure 3D, the Y-axis of the PCA plot says dimension "3", we believe it should say "2".

8. Could the authors provide an explanation for how the axes were defined for the PCA plots in figure 3D and supp figure 6B? This could be addressed in either the figure legend or methods section.

9. For the RNA-seq heatmaps, the scale is a Row Z-score. Could the authors please provide a further explanation for the Z-score calculation in the methods?

10. Could the authors highlight which of the downregulated transcripts in Figure 3G are proposed targets of FOXM1?

11. In the text (line 160), the authors state that "66 out of the 105 'mitotic cell cycle genes'" identified by RNA-seq are targets of FOXM1. Could the authors speculate on what might be regulating the other 39 mitotic genes?

12. Figure 4A: In the methods section it says that 20ug of protein were loaded for this blot. Does this mean that tubulin levels are actually decreased in aged cells? Is the white line on the blot between lanes 2 and 3 an artifact of scanning?

13. In the text (line 226), the authors state that they estimate ~25% and ~70% aneuploidy in the

young and old cell populations, respectively. Could the authors explain which data they used to reach this conclusion?

14. Typographical errors and suggested in-text edits:

- a. Line 21: Change "Supported by" to "Through"
- b. Line 22: add comma after "middle-aged,"
- c. Line 34: consider changing the text to "Numerous studies over the past several decades have demonstrated a link between aging and aneuploidy."
- d. Line 47: Repetitive with first sentence of previous paragraph (line 34)
- e. Line 53: add comma after "mouse,"
- f. Line 58: change wording to "hallmarks of cellular aging"
- g. Line 60: add comma after "aneuploidy,"
- h. Line 64: change wording to "we show that mitotic duration increases with advancing age, concurrent with a..."
- i. Line 66: change "shutdown" to "repression"
- j. Lines 69 and 70: consider changing the text to "Finally, we identify aneuploidy as a major hallmark of full senescence in naturally aged cells, which suggests that mitotic fitness enhancement may be a potential anti-aging strategy."
- k. Line 74: add comma after "study,"
- l. Line 82: add comma after "lifespan,"
- m. Line 94: add comma after "old-aged,"
- n. Line 122: add comma after "duration,"
- o. Line 152: add comma after "(MMPs),"
- p. Line 212: consider changing the text to "(FOXM1-dNdK) demonstrated an inherent link between aneuploidy and senescence."
- q. Line 216, change "accumulating" to "accumulated"
- r. Line 237: consider changing the text to "Altogether, our data provide support for the presence of a positive feedback loop between cellular aging and aneuploidy..."
- s. Line 238: Change "loose" to "lose"
- t. Line 239: Remove "an"
- u. New paragraphs should be generated after lines 250, 272, and 290
- v. Line 257: remove the word "mostly"
- w. Line 258: change "mis-segregation" to "segregation".
- x. Line 261: change text to "we found that elderly mitotic cells express SASP genes."
- y. Line 268: change "prominently" to "more"
- z. Line 305: consider changing the text to "FoxM1 repression translates into mild aneuploidy levels and may also further act..."
- aa. Line 310: change "it would" to "would it"
- bb. Line 781: add comma after "(MMPs),"
- cc. Line 849: change to "8-year-old donor"
- dd. Line 877: add comma after "stress,"
- ee. Line 902: change to "Time hour:min."
- ff. Line 914: change "α-tubulin" to "GAPDH"

Reviewer #2 (Remarks to the Author):

Macedo et al. propose that FoxM1-mediated gene expression declines with aging and leads to aneuploidy, which in turn promotes senescence and aging. The authors imply that this information could be used as a therapeutic strategy for senescence and aging.

Overall, the manuscript has several intriguing findings, the data on which the central conclusions

are based are rather weak at the present time. For instance, the proposed model of aging is solely based on studies of dermal fibroblasts and lacks in vivo validation. The authors did not critically test the central conclusion that aneuploidy drives senescence. Also, the literature suggests that FoxM1 is elevated in human cancers, which poses a potential problem for the anti-aging strategy proposed, which is not addressed by the authors.

The overall structure of the manuscript can be improved by changing the order of the figures and by making the supplemental figures relevant to the main figures presented.

By addressing the concerns below the authors could improve the study and obtain more compelling evidence for their central conclusions.

Major concerns:

1. As mentioned above, many conclusions throughout the manuscript are speculative and not supported by strong experimental evidence. The authors should strengthen their data or significantly tone down their statements. These include:

a. Line 238 - aging triggers abnormalities at several mitotic stages

From the provided data it is not clear, if mitotic mis-segregation is a cause or consequence of aging.

b. Line 208-209 - Overall, these data demonstrate that modulation of mitotic efficiency through FoxM1 induction in elderly and HGPS cells prevents both aneuploidy and cellular senescence.

The rather mild effect on % SAbGal+ cells after FoxM1 re-expression, does not warrant this conclusion. If all cells re-express FoxM1, why do still ~12% of cells show SAbGal+ compared to ~17% in control cells? The authors may want to rephrase this and suggest that FoxM1 delays senescence induction. Same applies to the effect of knocking down FoxM1 and SAbGal positivity in Figure 4 which is also rather modest.

c. Line 210, 225-229 – Aneuploidy is a hallmark of senescence.

The authors find a mere 8% of their SA-bgal+ positive to be aneuploid, which does not justify the use of the term hallmark of senescence. Besides, extrapolation of aneuploidy indices to all 23 chromosome pairs is not a recognized or recommended parameter thereby discrediting their statement.

d. Line 235-236 - These results demonstrate that FoxM1 levels modulate aneuploidy-driven senescence.

It has not been experimentally demonstrated in the manuscript that aneuploidy drives senescence. The authors merely suggest a correlation between aneuploidy and senescence but the causative events and consequence are unclear. The authors should follow the fate of cells with mitotic segregation defects to see if they get cell cycle arrested or continue into the next cell cycle to determine the effect of aneuploidy on the subsequent cell cycle. (However a recent paper by Santaguida S et al, Dev Cell 2017, shows that aneuploid cells are cleared by the immune system in the subsequent cell cycle). I recommend toning down this conclusion or not making it at all as the evidence is missing to support it.

e. Line 261-263 - The authors should be clearer, more complete and up-to-date with their discussion. The SASP is a complex program that is initiated and sustained by multiple pathways and stimuli, one of which is p38-MAPK. Please rephrase.

2. The authors use a parameter called aneusomy index throughout the paper but have not mentioned anywhere what it is or how is it measured, making it difficult to interpret any data associated with it. Please mention in the methods section as well as the main text how it is defined

and measured.

a. Nonetheless, it appears in e.g. Figure 1b that 1-2% of cells in either age group are aneuploid, however, this number seems to be surprising low. Similarly, an increase of % of cells with aneuploidy in from 1% to 1.5% in FoxM1-deficient cells, as well as a reduction from ~1.8% to 0.5% after FoxM1-dNdK expression is extremely mild. It is questionable if these changes – although statistically significant – are indeed biologically significant.

b. As for Figure 1c, I'm wondering how the authors addressed the issue of non-dividing, senescent cells in their assay. Growing, post-mitotic cells inhibited to undergo cytokinesis would appear as 4n cells, whereas non-proliferating, senescent cells would appear as mostly 2n population – I would assume these cells would be excluded from the analysis. Please, explain and set into context with the conclusion later in the manuscript stating that senescent, SAbGal+ cells are the main source of aneuploidy in aged cultures.

3. In order to warrant interpretation of the results, the authors need to exclude the possibility of delayed/enhanced proliferation rates and different apoptosis kinetics that may explain a difference in % SA-beta Gal+ cells or cells with aneuploidy, and other phenotypes. Do cultures with and without FoxM1, as well as young v/s old fibroblasts have the same cell cycle time, population doubling time and apoptosis rates?

4. The authors aim to show that mitotic timing increases with age and that this change is SAC dependent and partly responsible for the mis-segregation phenotype observed with aging. While the authors show that over-riding the SAC by inhibition of Mps1 reduces mitotic timing in aged fibroblasts, they do not demonstrate that this reduces the segregation defects observed. In fact the data in Figure 2e showing that treatment with the Mps1 inhibitor induces segregation defects in the 87y old fibroblasts is in stark contrast to their conclusion. Further, their conclusion of the delay being SAC dependent is weak as one would expect to see SAC hyperactivity, measured by the time arrested in mitosis after STLC treatment. I recommend checking SAC activity using spindle poisons (nocodazole or taxol) rather than using a kinesin-5 inhibitor. Also, if the increased time is indeed responsible for the mitotic delay, can the segregation defects observed in aged fibroblasts be recapitulated in young fibroblasts by increasing mitotic timing using APCIN or PROTAME?

5. It is very puzzling why the authors have made use of an Mps1 inhibitor in Figure 2 to investigate the main segregation defects seen in young v/s old fibroblasts. This investigation should instead be performed on unperturbed mitoses. The classification of segregation defects as mild and severe is also confusing. Why are cells with bipolar anaphase without cytokinesis failure being called mild mis-segregation and cells with cytokinesis failure being called severe? Some mis-segregation events or their consequences (such as cytokinesis failure, micronuclei formation and spindle mis-positioning) were quantified on a cell to cell basis, important mis-segregation events like chromosomal misalignment, lagging chromosomes and others were not quantified. Please, add quantification of these with appropriate statistics.

Also, the authors mention spindle mis-positioning to be a mitotic defect. They need to demonstrate how this affects their phenotype? For quantifying % of cells with spindle mis-positioning in Figure 2g, please add the cut-off criteria (angle) for scoring a normal vs. mis-positioned spindle.

6. Concerning the discussion: The authors fail to see that senescence is established as tumor suppressive mechanism and that induction of senescence in the context of aging and diseases is first and foremost a mechanism induced after cellular damage. This program restricts damaged cells from propagating and prevents neoplastic transformation. The authors should be more clear in this and should not mix up the difference between removal of established senescent cells, a strategy that has been shown to ameliorate diseases and age-related phenotypes, with the prevention of senescence and its deleterious risks. The authors should also discuss the possible side effects of FoxM1 overexpression as their therapeutic strategy to prevent senescence and aging associated disorders. The authors try to substantiate their statements by citing papers with

transgenic overexpression of FoxM1 not having any tumorigenic side effects in a model of lung tissue damage and liver regeneration. However, in those papers the mice were not aged long enough to be able to comment on the tumorigenic potential of FoxM1 overexpression, thus making it difficult to believe that FoxM1 overexpression might have no unpleasant consequences.

Again, based on concerns raised in point 1, the overall thought of the discussion having found a new mechanism for senescence induction and novel therapy complementary to senolytics should be toned down.

7. Supplementary tables are not helpful and lack completeness for the reader. Please add gene names to all lists, and their fold changes and p-value (FDR) etc.

Minor concerns:

- It is appreciated that the authors perform several unbiased transcriptomic RNA-seq experiments to assess the transcriptional pattern of aged, FoxM1-deficient and FoxM1-re-expressing systems. One trade-off is the very low number of replicates (2) which makes interpretation of the data challenging.

As heat maps that display a min/max Row Z-score based on normalized counts can be both helpful and misleading, please also indicate the fold change expression level between control and sample in the same heat maps. For the heat map in Figure 4j, 5h, gene names are missing.

- How do FoxM1 protein levels in 10y fibroblast after siFoxM1 compare to aged fibroblast? To which age group does the knock-down correspond to? A direct comparison of the samples on one Western Blot will address this. This may also give indications if the phenotype of FoxM1 deficiency is solely responsible for the age-related mitotic phenotype or if FoxM1 is one of multiple contributing factors,

- The SASP consists of a plethora of factors and is cell type- and senescence-stimulus specific. Please expand the presentation of SASP factors in the RNA-seq datasets and display in the heat maps. Also, the variability between the two replicates seems high and some SASP factors seem to be down-regulated, which makes the data less convincing. Employing additional methods to assess the SASP would help convince the reader. Perhaps Western Blotting, Immunofluorescence analysis or antibody arrays could be used.

- Please be consistent in the use of the qPCR reference gene. Figure 4c uses TBP whereas in Suppl. Figure 2 TBP and GAPDH was utilized.

- In the text from line 95-100, the authors talk about measuring aneuploidy in post mitotic cells. However the data corresponding to the text (Fig1c) shows BN cells with mis-segregation. Can the authors clarify if they are measuring aneuploidy by FISH or looking at segregation defects by live cell imaging?

- In Supp. Figure 2, the authors make use of murine adult fibroblasts to corroborate the findings found with human fibroblasts. However the qPCR data showing FoxM1 decline is not significant and should be mentioned and clarified in the text (line 166). There is also no Western Blot showing FoxM1 decline with age in MAFs. Besides, murine ear fibroblasts have a tendency to become polyploid early in culture. Could the authors please provide ploidy status of the MAFs used in their experiments by e.g. Propidium Iodide staining?

Reviewer #3 (Remarks to the Author):

The manuscript by Macedo et al. describes an investigation into the mitotic phenotype of aging fibroblasts. By surveying a panel of cells from donors with different ages, as well as fibroblasts from an individual with progeria, they report that the accuracy of chromosome segregation decreases with age, and they attribute this phenotype to a suppression of FOXM1-dependent transcription during mitosis. By over-expressing FOXM1, they report that they can rescue mitotic accuracy in aging cells, suggesting an interesting strategy to counteract age-related cellular decline.

Overall, the paper makes a strong case for its central thesis, and the experiments performed are relevant and well-described. The parallel experiments in human and mouse cells in particular are helpful in ruling out genetic confounders. Additionally, as a number of questions have been raised in recent years regarding the accuracy of FISH, it is reassuring to see that the fixed cell experiments were scored blindly (perhaps this could be mentioned in the main text, in addition to the methods section?).

My major question concerning this manuscript is the reported downregulation of mitotic genes in aging mitotic cells. This result is interpreted as implying that aging mitotic cells express lower levels of these genes. An alternate explanation would be that cell synchrony in aging cells is worse, and the population of aging mitotic cells had slightly more G1 or S-phase cells that were collected for RNA-Seq. I wonder about this possibility, given what the authors' interpretation would imply. Various gene expression thresholds exist that control cell cycle transitions (e.g., cyclin B/CDK levels for mitotic entry or CDC20 levels for anaphase). The authors' model suggests that these thresholds are lower for aging cells, perhaps because there is a stoichiometric decrease in all factors involved. This would raise a number of interesting questions about the biology of aging, which I have no doubt that the authors are following up. For this current manuscript, I think that it would be informative to present further evidence that aging cells express low levels of mitotic transcripts during mitosis. In particular, I think that the authors could perform quantitative IF experiments for one or two mitotic markers (cyclin B, FOXM1, CDC20, etc.) and show that in single aging cells these genes are in fact expressed at low levels during mitosis.

Besides that, a few minor comments:

1) Concerning premature aging, the BubR1 mouse is really an exception, and most other aneuploid mouse models (CDC20, MAD2, cohesin, etc.) do not show a progeroid phenotype. This fact is mentioned in the discussion, but I believe that it is mildly misleading to present the BubR1 mouse in the introduction as representative of CIN models.

2) It has previously been shown that single chromosome aneuploidy is sufficient to promote senescence, particularly in the case of Down syndrome (e.g., Contestabile 2009 and Sheltzer 2017). Thus, the authors' claim that while CIN has been shown to promote senescence, it is unknown whether low levels of aneuploidy do the same is not strictly accurate.

3) Though not necessary for this paper, it would be very interesting if BubR1 siRNA blocked the rejuvenating effects of FOXM1 over-expression. Have the authors tested this?

Overall, the authors present a solid manuscript that will significantly contribute to the literature on mitosis and aging. If the authors can present an independent method confirming the low expression of mitotic genes in aging cells, I would fully support publication.

Rebuttal letter

Reviewer #1 (Remarks to the Author):

In humans, chronological aging is associated with the development of chronic diseases and declining overall health. Aging is not only a chronological process but also a biological process, and the consequences of aging are in part attributed to altered cell physiology. One cellular hallmark of aging is an increase in aneuploidy. Aneuploidy is a state of abnormal chromosome numbers that is caused by chromosome segregation errors occurring during cell division. Further, aneuploidy is also linked to aging with elevated rates of aneuploidy leading to the onset of age-associated phenotypes. Thus, understanding the relationship between aneuploidy and cellular aging is an important topic in aging-related research.

Although previous studies have demonstrated an inter-dependent relationship between aneuploidy and aging, the molecular basis of this relationship remains unknown. These authors investigate the relationship between aneuploidy and aging by comparing human dermal fibroblasts (HDF) cells collected from participants ranging in age from neonatal to octogenarian. Using a variety of approaches, including fluorescent in-situ hybridization (FISH), immunofluorescent microscopy, and live-cell imaging, the authors demonstrate that aged cells exhibit an increase in mitotic errors and aneuploidy. In addition, using RNA-sequencing to compare the transcriptomes of young and old cells, the authors show that aged cells exhibit a down-regulation of mitotic cell cycle genes and an increase in senescence-associated secretory phenotype (SASP) genes. The down-regulation of mitotic cell cycle genes is shown to be a consequence of decreased mRNA and protein levels of the transcription factor FoxM1 that drives the transcription of many G2/M genes. Importantly, the authors demonstrate that siRNA-mediated repression of FoxM1 in young cells increases mitotic errors and aneuploidy. Moreover, they show that over-expression of a constitutively active FoxM1 decreases mitotic errors and aneuploidy in aged cells. This is a very powerful experiment. Lastly, the authors show that the population of senescent aged cells has an increase in the percent of aneuploid cells compared to the population of young senescent cells.

In conclusion, the authors propose a positive feedback loop between aging and aneuploidy with aged cells displaying a SASP phenotype and increased mitotic errors that generates aneuploid progeny and in turn aneuploid progeny enter full senescence causing age-related pathologies. Overall these studies provide evidence that a loss of mitotic fidelity and an increase in aneuploidy due to down-regulation of FoxM1 in aged cells is the molecular cause of increased senescent cells during natural aging and that restoring mitotic fidelity is an anti-aging therapeutic strategy. This manuscript provides novel data on the molecular basis of aging, and I am strongly supportive of publication provided that the authors address some specific concerns.

Major Comments:

1. The authors should provide a quantification of mitotic index for the young vs. old-aged cells in order compare differences in proliferation rate between the two cell types.

We have now included in the revised version of the manuscript a quantification of mitotic index for the young vs. old-aged cells (Supplementary Fig.4a). Although normally low even for young donor fibroblast cultures, the mitotic index is significantly reduced in old-aged vs. young cells ($0.90 \pm 0.14\%$ vs. $1.80 \pm 0.42\%$, respectively; $p < 0.01$) consistently to decreased proliferation rate.

Moreover, we evaluated the population doubling (PD) over time, which provides an estimate of overall proliferation capacity of a cell population. Young and old-aged fibroblasts were

cultured up to fifth passage and numbers of cells at the end of each passage were counted to determine PD (Supplementary Table 1; Materials and Methods). Average population doubling time calculated for the cumulative passages in our experiments (≤ 5) shows significantly different proliferation rates between young and elderly cultures (49.1 ± 9.2 vs. 119.1 ± 32.0 hrs, $p < 0.05$). This was not accompanied by an increase in apoptotic cell death, as examination of Annexin V-positive cells by flow cytometry at passages 2-3 indicated a similar percentage in neonatal and octogenarian cell cultures (Supplementary Fig.4b,c).

2. In the manuscript, could the authors please provide a further explanation for their aneusomy index calculation (Figures 1B, 1C, and 6F)?

In the revised version of the manuscript please note that Fig. 1b, 1c and 6f are now Fig. 1a, 1b and 7f, respectively.

In Fig. 1a, we measured the prevalence of somatic aneuploidy by the ratio of aneusomic cells for chromosomes 7, 12 and 18 to the total cell count for a sample (> 2500 nuclei) (aneusomy index). The same measurement applies to Fig. 5d, 6e, 7f and Supplementary Fig.9e. This methodology is described in Materials and Methods as ‘Interphase FISH’.

In Fig. 1b, we measured the rate of chromosome mis-segregation (number of events in which two sister chromatids co-segregate to the same daughter cell) by combining a cytokinesis-block assay (using cytochalasin D for 24h) with FISH staining. This methodology (referred as Cyto D-FISH; Materials and Methods) allows the analysis of the reciprocal distribution of chromosomes between the daughter nuclei of a single mitotic division. Fig. 1b shows the percentage of binucleated telophases exhibiting chromosome mis-segregation over the total binucleated cell count for a sample (> 200 BN cells). The same measurement applies to Fig. 5e, 6f and Supplementary Fig 9f.

We have now clarified this issue in the revised version of the manuscript, both in the main text (lines 90-94; 98-104) and in Material & Methods section.

2.1. Does this index include tetraploid cells? In Figure 2, the authors demonstrate that there is a significant population of aged cells that fail cytokinesis. Are these cells included in the aneusomy index calculation?

Mononucleated tetraploid cells were not included in the aneusomy index calculation. The percentage of these cells in all samples varied, but with few significant differences found between distinct samples (Reviewer 1 figure 1). However, binucleated tetraploid cells (arising from cytokinesis failure) were likely included in the aneusomy index calculation in Fig. 1a, 5d and 6e, as the total cell count is based on nuclei staining.

Reviewer 1 figure 1. Percentage of mononucleated tetraploid cells. Sample size > 2500 nuclei per condition. n.s. $p > 0.05$, $**p < 0.01$ and $****p < 0.0001$ by two tailed χ^2 statistical tests.

2.2. Also, rather than combining the data for all three chromosome probes, it would be informative to provide the data for each probe separately (plotted as a deviation from the mode). In particular, the authors extrapolate the data to all 23 chromosome pairs to determine overall levels of aneuploidy. This calculation relies on the assumption that all chromosomes have an equal probability of contributing to aneuploid progeny.

We now show the aneusomy index of each chromosome separately (plotted as deviation from the mode) (Supplementary Table 2). In most cases, the aneusomy index of each chromosome is not statistically significant, suggesting that a combination of aneusomy indices of 3 chromosomes is a stronger predictor in case of mild aneuploidy levels (lines 94-98). A few exceptions were found in which one chromosome separately contributes significantly to increased aneuploidy (chromosome 18 in 52/54y and progeria samples; chromosome 7 in 84/87y and Neo siFoxM1 samples). Thus, from these current data for 3 chromosome pairs is difficult to conclude for a bias in the aneuploid progeny. The extrapolation to 23 chromosome pairs indeed assumed that all chromosomes have an equal probability of contributing to aneuploid progeny, as we averaged the aneusomy indices of chromosomes 7, 12 and 18. We would certainly like to address the existence of a bias vs. random chromosome mis-segregation during aging, but for the context of this work, we will just exclude any extrapolation from 3 to 23 chromosome pairs.

3. In the text, the authors state that they observe higher levels of aneuploidy using a cytokinesis block assay. A cytokinesis block assay does not measure aneuploidy levels, but is a measure of the rate of chromosome mis-segregation. This is an important distinction as cells can mis-segregate chromosomes but fail to proliferate further with an aneuploid genome. A higher rate of chromosome mis-segregation is not strictly equivalent to higher levels of aneuploidy. The text needs to be revised to reflect what the assay is measuring.

We thank the Reviewer for this comment and we have now revised the main text accordingly (lines 90-94; 98-104) (see also answer to point 2. above). Indeed, whereas the CytoD-FISH measures chromosome mis-segregation rate in daughter nuclei of mitotic divisions that took place during the 24h period of cytochalasin D treatment, the interphase FISH measures the prevalence of somatic aneuploidy in a population under selective pressure. Comparison between Fig. 1a and 1b, highlights that whereas chromosome mis-segregation rate increases 2.5x in 87y vs neonatal (Fig. 1b), the aneuploidy levels are 1.3x higher in 87y vs neonatal (Fig. 1a), supporting that mis-segregation rate and aneuploidy are not equivalent. Cells that mis-segregate chromosomes might be outcompeted by diploid cells, deflating the levels of aneuploidy found by interphase FISH in Fig. 1a in comparison to the higher rate of chromosome mis-segregation found by cytoD-FISH in Fig. 1b.

4. Could the authors provide an explanation for why there is an observed decrease in aneuploidy for the young cells in Figure 1C (vs Figure 1B)?

As clarified in points 2 and 3 above, the assays in Fig. 1a and 1b measure aneuploidy levels and chromosome mis-segregation rates, respectively. Therefore, for young cells we can only state that aneuploidy levels are lower than in aged cells (Fig. 1a), and that chromosome mis-segregation rate is lower than in aged cells (Fig. 1b). Because aneuploidy levels and mis-segregation rate are not equivalent, the values for young cells in the two graphs are not comparable. Furthermore, whereas interphase FISH (Fig. 1a) measures aneuploidy levels accumulated for undetermined time in an asynchronous population under selective pressure, cytoD-FISH (Fig. 1b) measures the mis-segregation events taking place during the 24h treatment with cytochalasin D. However, as suggested in comment 3 above, if aged cells that mis-segregate chromosomes might be outcompeted by diploid cells, deflating the levels of aneuploidy found by interphase FISH, then one possible explanation for the fact that young cells exhibit higher somatic aneuploidy than chromosome mis-segregation rate, might be that

young aneuploid cells are fitter (and thereby more slowly outcompeted by diploid cells) than elderly aneuploid cells.

5. Could the authors provide quantifications for chromosome alignment defects and lagging chromosome phenotypes depicted in Figure 2B and 2C?

In the revised version of the manuscript we now provide quantifications for chromosome alignment (congression) delay and lagging chromosome phenotypes based on spinning-disk confocal live-cell imaging analysis (Fig. 2e,f).

In the previous version of the manuscript, we quantified these phenotypes indirectly based on the following rationales:

1) If older cells have chromosome alignment delay, these cells will exhibit extended mis-segregation defects when rushed to exit mitosis through inhibition of SAC activity with the Mps1 small molecule inhibitor. Considering that even the young cells most often mis-segregate chromosomes when treated with Mps1i, the mis-segregation defects were classified into mild or severe depending on the mitotic exit phenotype, i.e. bipolar anaphase generating two daughter cells or slippage generating one 4N daughter cell.

2) If older cells exhibit lagging chromosome phenotypes, these cells will more often generate micronuclei (Thompson and Compton, 2011).

However, we concede these rationales are likely unacquainted outside mitosis research, and therefore, the phenotypes have been now quantified directly from live-cell imaging analysis. In the revised version of the manuscript, data from Fig. 2e and Supplementary Fig.4a have been excluded.

6. In figure 2B, the authors show images from a live-cell movie of an aged cell with unaligned chromosomes. It is unclear from the text description if aged cells are taking longer to achieve chromosome congression but eventually do or if aged cells are entering anaphase with unaligned chromosomes. The distinction is important because in the former scenario the SAC is functioning properly, but in the latter the SAC is not.

Fig. 2b shows still images from Supplementary Movie 2 of an aged cell with chromosome congression delay. Aged cells entering anaphase with unaligned chromosomes were never observed under spinning-disk confocal fluorescence microscopy. This is in agreement with the fact that SAC is functioning properly as shown in Supplementary Fig.3g,h. The text has now been revised to clarify this issue (lines 143-144).

7. It is unclear what the authors are trying to show in the image for figure 3B. What cell type is being imaged? Are we looking at mitotic chromosome spreads? If the point of this figure is to show that the mitotic indices are the same in both the young and aged cell populations used in the RNA-seq experiment, then both a neonatal and 87yo representative image should be displayed. Some indication of magnification should be provided in the figure legend for the insets.

Fig. 3b shows chromosome spreads from the mitotic cell populations retrieved by culture flask shake-off, and then used for RNA isolation and sequencing. The purpose is to show how pure the mitotic cell fractions are, with mitotic indices >95% for both young and aged cell populations. We used a similar number of neonatal and 87y mitotic cells to isolate RNA (of course more culture flasks were needed to shake-off enough number of 87y mitotic cells). For clarity, we now include representative images for both neonatal and 87y mitotic fractions (Fig. 3b), as well as the quantification of their mitotic indexes (Fig. 3c). Indication of magnification for the insets is provided in figure legend.

8. The RNA-seq data is described as showing two biological replicates, but in the methods, it states that the two biological replicates came from a single experiment. If so, than these are not biological replicates, but technical replicates. The gene expression data should be averaged across the two technical replicates and not reported as separate biological replicates.

This is necessary to identify the significant changes in gene expression between young and aged cells.

In the revised version of the manuscript, we clarify this issue. The RNA-seq data are from two independent RNA isolation experiments for each biological sample and/or condition, neonatal vs. 87y fibroblasts (Fig. 3 g,h,i), 10y vs. 10y siFoxM1 (Fig. 5 a,h,i), and 87y vs. 87y dNdK (Fig. 6 b,i,j). We averaged the gene expression data across the technical replicates to identify significant changes in gene expression (Materials and Methods).

9. The authors provide evidence suggesting that decreased mRNA expression levels of mitotic genes cause a loss of mitotic fidelity in aged cells. To strengthen this conclusion, it is necessary to demonstrate that decreased mRNA levels are leading to decreased protein levels in aged vs. young cells. Western blot data for some of the candidate mitotic genes is needed to validate the results from the RNA-seq experiment in figure 3.

We demonstrate now that decreased mRNA levels are leading to decreased protein levels in aged vs. young mitotic cells by providing Western blot data for some of the candidate mitotic genes, namely *CCNB1*, *PLK1* and *NDC80* (Fig. 4f,g). Additionally, we include single-cell immunofluorescence analysis for the relative intensity levels of FoxM1, Cyclin B and Cdc20 in young vs. old aged mitotic fibroblasts (Supplementary Fig.5; Fig. 4h-j).

10. In Figures 4D and 5A, could the authors provide a biological explanation for why a specific subset of mitotic transcripts is upregulated in the FOXM1 KD condition and downregulated in the FOXM1 over-expression condition, respectively?

Using the “mitotic cell cycle” GO term from the DAVID Gene Ontology tool, we indeed got a subset of genes upregulated in the FOXM1 RNAi (*CCND1*, *CCNE1*, *CDH13*, *CDKN1A*, *GOLGA2*, *HIST1H4A*, *HIST1H4B*, *HIST1H4D*, *HIST1H4F*, *HIST1H4H*, *LPIN1*, *NINL*, *OPTN*, *PSMB10*, *SDCCAG8*, *TUBA4A*, *TUBG2*, *UBC*) and downregulated upon FOXM1 over-expression (*CCNE1*, *CCNE2*, *HIST1H4B*, *HIST1H4H* and *PRKAR2B*). However, we realized that most of these genes are non-canonical mitotic genes, and that the ‘mitotic cell cycle’ GO term actually includes genes with annotated roles in other phases of the cell cycle. Therefore, for simplicity, in the revised version of the manuscript we generated heatmaps for the ‘mitosis’ GO term, which specifically includes genes with annotated mitotic function. Still, some genes are upregulated in FOXM1 RNAi (*ANAPC13*, *CCNG2*, *HAUS7*, *HGF*, *HMGA2*, *LMLN*, *LZTS2*, *MAP9*, *MAPRE2*, *MAPRE3*, *NEDD9*, *NEK6*, *RGS14*, *SIRT2* and *ZNF830*), or downregulated in FOXM1 dNdK (*BOD1*, *CCNG2*, *CD2AP*, *CENPV*, *MAP9* and *NEK6*). *CCNG2*, *MAP9* and *NEK6* are genes whose expression is altered in both conditions, KD and over-expression, suggesting that are indeed FoxM1-responsive even though indirect targets. For the other genes, is difficult to come out with an explanation. The differences can be induced by specific experimental conditions (RNAi or lentiviral transduction), or might reflect genes whose transcriptional regulation changes in response to alterations in FoxM1 targets. Moreover, the subset of mitotic genes upregulated in FoxM1 RNAi and downregulated in FoxM1 dNdK includes mostly genes with quite elusive role in mitosis. For instance, *HGF*, which is upregulated in FoxM1 RNAi, is actually a SASP gene (Fig. 5h).

11. In supplementary figure 3A, the authors observe a significant increase in multitelomeric signal with age. Could the authors define what a multitelomeric signal is and better explain how this increase is not contributing to the mitotic defects observed in the aged cells?

We thank the referee for the comments and helping us to clarify this point.

First described by the deLange laboratory (Sfeir et al., Cell 2009), Multi-Telomeric Signals (or MTS) is an indirect measurement of telomere replication stress (now mentioned in Supplementary Fig.3 legend). The authors showed by combining Immunofluorescence and SMARD (single-molecule analysis of replicated DNA) techniques that MTS are formed upon replication fork stalling at telomeres. Since then, this measurement is commonly used as a hallmark of telomere dysfunction. Importantly, the formation MTS occurs in S phase. Thus,

MTS damage produces S and G2 cell cycle checkpoints. However, because MTS are scored in metaphase spreads, one may be misguided to think that defects occur primarily in mitosis. A recent publication in Nature (Hayashi et al 2015) described that mitotic delay can occur as a direct consequence of telomere deprotection in mitosis. However, Hayashi and colleagues showed that both telomere crisis and mitotic delay are specifically dependent on telomere-end fusions (seen on Extended Figure 3 of the article). Our experiments were specifically designed to rule out that the mitotic delay was a direct consequence of telomere fusions. We were unable to detect any fusion event in our data. Thus, although we cannot totally rule out a possible mitotic checkpoint due to replication defects at telomeres (MTS), it is unlikely to be its primary source.

12. Could the authors provide a table or figure illustrating overlap between a) the transcripts that were downregulated in the aged cells (relative to the young cells-Figure 3) and b) the transcripts that were downregulated in the FOXM1 KD cells (Figure 4)?

We now provide a figure illustrating the overlap between the transcripts altered in aged cells and the transcripts altered in FoxM1 KD cells (Fig. 5b). 58% of the age-associated transcriptional differences are found in FoxM1 RNAi. The list of overlapping genes is included in Supplementary dataset 6. The top 10 GO terms for the overlapping genes are related to cell cycle and mitosis.

13. In the text, the authors discuss a feedback loop between cellular aging and aneuploidy. A diagram depicting the feedback loop would be a helpful addition at the end of this manuscript to clarify the steps in this loop. Additionally, it may be helpful to further define the difference between an “early senescent phenotype” and a “full senescent phenotype” (lines 238-239).

A diagram depicting the feedback loop is shown at the end of the manuscript (Fig. 8) and accordingly to the referee’s suggestions. The diagram depicts the steps in the feedback loop between cellular aging and aneuploidy, and refers to ‘early’ and ‘full’ senescence. The difference between ‘early’ and ‘full’ senescence is further defined in the main text (lines 314-318).

Minor Comments:

1. In the introduction, it might be beneficial for the authors to define the difference between biological age and pathological age (referenced in line 87).

We revised the main text for simplicity. In line 84 we refer now “... accumulation of senescent cells has been widely reported for chronological aging and age-related disorders”. We believe these terms are better adjusted for general readers and skip definition. Indeed, ‘biological age’ is used just recently to refer to the body’s real age compared to chronological age (for instance based on the analysis of methylation biomarkers).

2. The authors measure spindle mis-positioning as a read-out for general mitotic dysfunction, but it is unclear how spindle positioning with regards to the substrate surface contributes to mitotic errors. Could the authors please add in-text references for the contribution of spindle mis-positioning to mitotic errors?

We agree with the referee that it is largely unknown how spindle mis-positioning in relation to the substrate surface contributes to mitotic errors (chromosome mis-segregation). We found only a few references supporting this correlation. *Ertych N et al. (Nat Cell Biol 2014)* have shown increased microtubule assembly rates to be associated with transient abnormalities in mitotic spindle geometry in relation to substrate surface that promote generation of lagging chromosomes. Also, *Vitiello E et al. (Nat Commun 2016)* have shown DLC2 or Kif1B depletion to promote microtubule stabilization, defective spindle positioning, chromosome misalignment and aneuploidy.

Since the contribution of spindle mis-positioning to mitotic errors is elusive, we removed previous data shown in Supplementary Fig.4b-d (detailed quantitative analysis of the spindle mis-positioning phenotype) to avoid this phenotype of being overemphasized.

3. In supp figure 1B, could the authors explain why it is important to quantify cell area? Is this a readout for senescence?

Enlarged cell morphology is a readout for senescence (*Rodier and Campisi, J Cell Biol 2011; Hayflick, 1965*). Thus, cell area was quantified as readout of enlarged cell morphology, together with other senescence biomarkers (β -gal activity, p21 and 53BP1).

4. Please reference all supplemental videos in the text.

We now refer to all supplemental videos in the revised version of the main text.

5. Overall, additional details are required in the figure legends. In particular:

a. The gray box in supplemental Figure 6E is unlabeled. The gray box is now labeled in revised Supplementary Fig.8d.

b. Line 764: Please reference Supp Fig 4 in Figure legend 2E. Previous data from Supplementary Fig 4 is no longer included in the revised version of the manuscript (see major point 5 above).

c. Please provide magnification for the insets in figures 3B and sup figure 1C. Magnification is now provided in the figures' legends.

d. Time on representative images in supp figure 4A should be displayed as "Time hr:min" (Line 902). Additionally, the length of scale bar for supp fig 4A should be defined. Supplementary Fig.4a is no longer included in the revised version of the manuscript.

e. What cell type is used for the representative images in supp figures 1C and 1E? The cell type is now indicated in these figures.

f. Supp Fig 5G: Does the FOXM1 dNdK protein run as a doublet? Is the non-specific band in both lanes? The figure legend also says the control cells are "untransduced" when the figure itself implies that they were transduced with the vector.

FoxM1 dNdK protein runs as a doublet. The non-specific band is in both lanes (an asterisk indicating the unspecific band was missing in lane FoxM1dNdK). The referee can also find FoxM1 dNdK protein doublet in Supplementary Fig.7d. Actually, depending on the WB experiment, sometimes the unspecific band was absent (as in Supplementary Fig.7d).

Control cells were transduced with empty vector; the legend has been corrected.

g. In Supp Fig 5H-J, were these quantifications done in both 84 and 87 year old donor cells? In Supplementary Fig.5h-j the quantifications were done in both 84 and 87 year old donor cells.

6. The mitotic indices for supp figures 5A and 5F should be quantified.

The mitotic indices of 10y vs. 10y siFoxM1 and of 87y vs. 87y FoxM1 dNdK (asynchronous populations) are now shown in revised Supplementary Figs. 6b and 7b.

However, note that Supplementary Fig. 5c,f (now Supplementary Figs. 6c and 7c) represent cell cultures treated with STLC for mitotic enrichment, and used to prepare mitotic cell protein extracts for western blotting analysis (WB in Supplementary Figs. 6d and 7d) accordingly to the experimental layout of Fig. 3a. These figures highlight the different

proliferation rate achieved following modulation of FoxM1 expression, as judged by the poor vs. high enrichment of mitotic cells under STLC treatment in 10y siFoxM1 vs. 87y FoxM1 dNdK.

7. In Figure 3D, the Y-axis of the PCA plot says dimension “3”, we believe it should say “2”. A new MDS plot has been provided in the revised manuscript (Fig. 3e). The axis was indeed mislabeled.

8. Could the authors provide an explanation for how the axes were defined for the PCA plots in figure 3D and supp figure 6B? This could be addressed in either the figure legend or methods section.

Additional text has been added to the legends: “The axes in the MDS plot represent leading logFCs, calculated by the root-mean-square of the largest absolute log-fold changes between each pair of libraries”.

9. For the RNA-seq heatmaps, the scale is a Row Z-score. Could the authors please provide a further explanation for the Z-score calculation in the methods?

Additional information has been added to the methods section: “Briefly, transcript counts were normalized to the sample library size. These values were subsequently scaled by gene using normalized scores or z-scores (i.e. a value of 0 corresponds to the mean expression of that gene across all libraries, and ± 1 , ± 2 , etc. represent 1, 2, etc. standard deviations from the gene mean).”

10. Could the authors highlight which of the downregulated transcripts in Figure 3G are proposed targets of FOXM1?

As mentioned in major point 10, the GO term ‘mitotic cell cycle’ from DAVID Gene Ontology tool includes many genes with functional roles outside mitosis. Therefore, in the revised version of the manuscript, we now show heatmaps of genes of the GO term ‘mitosis’, which is refined for mitotic genes. For this GO term, we found 36 out of the 71 mitotic genes significantly altered in 87y vs. neonatal HDFs (Fig. 3g) to be DREAM/FoxM1-MMB target genes containing CHR or CDE/CHR elements in their promoters (*Muller GA et al, Oncotarget 2016 8(58):97736-97748; Fischer M et al., Nucleic Acids Res 2016 44(13):6070-86*) (Fig. 4b). However, when comparing with ‘mitosis’ genes downregulated in FoxM1 RNAi, actually we found 63 out of the 71 mitotic genes significantly altered in 87y to be FoxM1-dependent (Fig. 6c). This suggests that either there are several mitotic FoxM1 targets that might lack CHR elements in their promoters, or that were not detected under the experimental conditions and cell lines used in the previous studies cited above. Alternatively, those genes are indirectly regulated by FoxM1 levels rather than being direct FoxM1 transcriptional targets. Fig. 4b and Fig. 6c have been included in the revised version of the manuscript to clarify this question raised by the referee.

11. In the text (line 160), the authors state that “66 out of the 105 ‘mitotic cell cycle genes’” identified by RNA-seq are targets of FOXM1. Could the authors speculate on what might be regulating the other 39 mitotic genes?

As explained in major point 10 and minor point 10 above, the GO term ‘mitotic cell cycle’ has been replaced by the GO term ‘mitosis’ to generate the heatmaps in Fig. 3g, Fig. 5a and Fig. 6b. For this GO term, we found that 63 out of the 71 mitotic genes significantly altered in aged cells (Fig. 3g) are also significantly altered in FoxM1 RNAi (Fig. 6c). The 8 outlying genes, include 4 upregulated genes (*SUGT1, VCPIPI1, PAPD5, STAG2*) and 4 downregulated genes (*NEK3, CHMP1A, RUVBL1, HAUS4*). With the exception of *PAPD5* (poly(A) RNA polymerase D5), the other 3 upregulated genes exhibited inconsistent changes in the RNA-seq replicates, so we question whether these are truly upregulated. The 4 downregulated genes that were not found in FoxM1 siRNA, are likely to be regulated by an alternative FoxM1-

independent pathway. However, the mitotic role of *NEK3*, *CHMP1A* and *RUVBL1* is unclear. *HAUS4* is a subunit of the Augmin complex with established role in mitotic spindle assembly that, contrarily to *HAUS8*, seems not to be a FoxM1 target.

12. Figure 4A: In the methods section it says that 20ug of protein were loaded for this blot. Does this mean that tubulin levels are actually decreased in aged cells? Is the white line on the blot between lanes 2 and 3 an artifact of scanning?

Accordingly to densitometry analysis, the levels of tubulin were decreased in 52y and 87y samples for this blot and there was a reason for this. As a higher number of cell culture flasks are required to isolate enough mitotic cells from the older cell cultures, increased level of albumin (from serum) is detected, underrating total protein quantification by Lowry. However, this discrepancy in the loading between lanes is adjusted by normalizing FoxM1 levels to tubulin levels.

The white line on the blot is an artifact introduced during scanning on the densitometer as shown below (Reviewer 1 figure 2). However, we have another scanning of the FoxM1 blot where the white line was not crossing any of the bands (see below Fig. 2 b1, b2); thus, this other scanning was used to revise Fig. 4a (which is now Fig. 4c). For the tubulin blot, unfortunately all scanings had the white line in the same place (c1, c2), the reason why we could not replace this image.

Reviewer 1 figure 2. Western blot in Fig. 4c. a) Scanning image of FoxM1 immunoblot used previously in Fig. 4a. During scanning on the densitometer, a white line artifact was introduced, which crosses a band of interest. b1) Scanning image of the same blot used to revise Fig. 4a. Here the white line artifact is outside the region of interest. Boxes U1-U6 indicate adjusted volume density analysis. B1 is the background subtracted. b2) Image crop from b1 used in revised Fig. 4a (which is now Fig. 4c). c1) Scanning image of tubulin immunoblot used to revise Fig. 4a. Here the white line artifact crosses the region of interest. Boxes U1-U6 indicate the adjusted volume density analysis. B1 is the background subtracted. c2) Image crop from c1 in revised Fig. 4a (which is now Fig. 4c).

13. In the text (line 226), the authors state that they estimate ~25% and ~70% aneuploidy in the young and old cell populations, respectively. Could the authors explain which data they used to reach this conclusion?

Note that these estimations are for young and old cell populations sorted for β -galactosidase positivity and stained for interphase FISH analysis of 3 chromosome pairs (7, 12 and 18). This conclusion is based in a simple rule extrapolation of the aneuploidy levels found for 3

chromosome pairs (AI) (Fig. 7f) to 23 pairs (AI/3 x 23). Of course this calculation relies on the assumption that all chromosomes have an equal probability of contributing to aneuploid progeny (as raised in major point 2 above).

14. Typographical errors and suggested in-text edits:

All revised accordingly to the Reviewer's suggestions.

- a. Line 21: Change "Supported by" to "Through"
- b. Line 22: add comma after "middle-aged,"
- c. Line 34: consider changing the text to "Numerous studies over the past several decades have demonstrated a link between aging and aneuploidy."
- d. Line 47: Repetitive with first sentence of previous paragraph (line 34)
- e. Line 53: add comma after "mouse,"
- f. Line 58: change wording to "hallmarks of cellular aging"
- g. Line 60: add comma after "aneuploidy,"
- h. Line 64: change wording to "we show that mitotic duration increases with advancing age, concurrent with a..."
- i. Line 66: change "shutdown" to "repression"
- j. Lines 69 and 70: consider changing the text to "Finally, we identify aneuploidy as a major hallmark of full senescence in naturally aged cells, which suggests that mitotic fitness enhancement may be a potential anti-aging strategy."
- k. Line 74: add comma after "study,"
- l. Line 82: add comma after "lifespan,"
- m. Line 94: add comma after "old-aged,"
- n. Line 122: add comma after "duration,"
- o. Line 152: add comma after "(MMPs),"
- p. Line 212: consider changing the text to "(FOXM1-dNdK) demonstrated an inherent link between aneuploidy and senescence."
- q. Line 216, change "accumulating" to "accumulated"
- r. Line 237: consider changing the text to "Altogether, our data provide support for the presence of a positive feedback loop between cellular aging and aneuploidy..."
- s. Line 238: Change "loose" to "lose"
- t. Line 239: Remove "an"
- u. New paragraphs should be generated after lines 250, 272, and 290
- v. Line 257: remove the word "mostly"
- w. Line 258: change "mis-segregation" to "segregation".
- x. Line 261: change text to "we found that elderly mitotic cells express SASP genes."
- y. Line 268: change "prominently" to "more"
- z. Line 305: consider changing the text to "FoxM1 repression translates into mild aneuploidy levels and may also further act..."
- aa. Line 310: change "it would" to "would it"
- bb. Line 781: add comma after "(MMPs),"
- cc. Line 849: change to "8-year-old donor"
- dd. Line 877: add comma after "stress,"
- ee. Line 902: change to "Time hour:min."
- ff. Line 914: change " α -tubulin" to "GAPDH"

Reviewer #2 (Remarks to the Author):

Macedo et al. propose that FoxM1-mediated gene expression declines with aging and leads to aneuploidy, which in turn promotes senescence and aging. The authors imply that this information could be used as a therapeutic strategy for senescence and aging.

Overall, the manuscript has several intriguing findings, the data on which the central conclusions are based are rather weak at the present time. For instance, the proposed model of aging is solely based on studies of dermal fibroblasts and lacks *in vivo* validation.

We have now included additional data that strongly support the central conclusions and address the Reviewer's major concerns (see answers below).

We are pursuing with an *in vivo* validation of our proposed model, but an extended period of time will be needed for the evaluation of lifespan and aging phenotypes.

The authors did not critically test the central conclusion that aneuploidy drives senescence.

In the revised version of this manuscript, we have included an experiment that critically tests the central conclusion that aneuploidy drives senescence. We performed long-term time-lapse microscopy of octogenarian fibroblasts expressing H2B-GFP. Mitotic cells with chromosome segregation defects (anaphase lagging chromosomes and/or micronuclei), as well as mitotic cells without apparent chromosome mis-segregation, were imaged for >48 hrs. Daughter cell fate was tracked individually for cell death, cell cycling or senescence (β -galactosidase assay at the end of the movie). Moreover, cells were fixed and stained for 53BP1 and p21 senescence markers, and fixed cell analysis correlated with the live cell imaging records (Supplementary Movies 9-11). We show that all cells with mis-segregation events stop proliferating and largely evolve into the acquisition of senescence biomarkers, supporting that aneuploidy triggers full senescence (permanent cell cycle arrest) in aged cells.

Also, the literature suggests that FoxM1 is elevated in human cancers, which poses a potential problem for the anti-aging strategy proposed, which is not addressed by the authors.

Even though FoxM1 is typically elevated in human cancers, FoxM1 is only tumorigenic if tumor suppressor genes are lost or oncogenic mutations (such as K-RASG12D) are present. Two references are included in the revised manuscript that directly prove this (Wang IC et al. *Developmental Biology* 2010 347(2):301-304; Cai Y et al. 2013 *J Biol Chem* 288(31):22527-41). The first reference is for lung cancer. The second reference is for prostate cancer where FoxM1 overexpression was not tumorigenic even in the absence of p19 tumor suppressor.

Moreover, as mentioned previously in the discussion section, FoxM1 overexpression was shown to improve liver-regenerating capacity in older mice (Wang X et al. 2001 *Proc Natl Acad Sci USA* 98:11468-73) and lung regeneration after injury (Kalinichenko VV et al. *J Biol Chem* 2003 278:37888-94). Thus, similarly to cellular reprogramming by transient expression of Yamanaka factors, recently shown to ameliorate age-associated symptoms without tumorigenesis induction (Ocampo A et al. 2016 *Cell* 167:1719-33), cyclic induction of FoxM1 transcriptional activity could potentially work safely.

[Redacted]

The overall structure of the manuscript can be improved by changing the order of the figures and by making the supplemental figures relevant to the main figures presented.

Even though we could not deduce the order of the figures the Reviewer had in mind, the figures and supplementary figures have been now significantly revised. We expect the revised format meets the Reviewer's concerns.

By addressing the concerns below the authors could improve the study and obtain more compelling evidence for their central conclusions.

Major concerns:

1. As mentioned above, many conclusions throughout the manuscript are speculative and not supported by strong experimental evidence. The authors should strengthen their data or significantly tone down their statements. These include:

a. Line 238 - aging triggers abnormalities at several mitotic stages

From the provided data it is not clear, if mitotic mis-segregation is a cause or consequence of aging.

We revised our model scheme (see Fig. 8) to better depict the steps in the feedback loop between aging and aneuploidy. These steps are the following, all supported by the experimental data:

1. Aged cells known for their accumulated genetic damage, can still divide even though exhibiting an early senescence phenotype (Fig. 3h,i) and repression of FoxM1 transcription factor (Fig. 4a-c);
2. This leads to general dysfunction of the mitotic machinery (Fig. 2) and increased chromosome mis-segregation rate (Fig. 1b) (“chromosome mis-segregation as consequence of aging”);
3. The aneuploid progeny stops proliferating (full senescence state) (Fig. 7g-i) (“chromosome mis-segregation as cause of aging”);
4. Induction of FoxM1 transcriptional activity in elderly cells ameliorates cell autonomous and non-autonomous (SASP) aging phenotypes (Fig. 6), thereby slowing down the feedback loop.

b. Line 208-209 - Overall, these data demonstrate that modulation of mitotic efficiency through FoxM1 induction in elderly and HGPS cells prevents both aneuploidy and cellular senescence.

The rather mild effect on % SAbGal⁺ cells after FoxM1 re-expression, does not warrant this conclusion. If all cells re-express FoxM1, why do still ~12% of cells show SAbGal⁺ compared to ~17% in control cells? The authors may want to rephrase this and suggest that FoxM1 delays senescence induction. Same applies to the effect of knocking down FoxM1 and SAbGal positivity in Figure 4 which is also rather modest.

We agree with the Reviewer’s suggestion to rephrase that FoxM1 delays senescence induction. Indeed, our experimental data support that FoxM1 delays senescence rather than clearing senescent cells. While improving mitotic fitness of early (pre-)senescent cells, FoxM1 prevents the generation of aneuploid progeny that we show to largely contribute to the SA-bGal⁺ cell population. The mild effect on SA-bGal⁺ cells after FoxM1 re-expression represents therefore the out-competition of permanently cell cycle arrested senescent cells in the population by the diploid progeny generated from short-term FoxM1 re-expression. The modest effect of knocking-down FoxM1 in SA-bGal positivity is in agreement with the mild levels of induced aneuploidy. Still, is a 2-fold change under a short-term experiment window (72h RNAi).

c. Line 210, 225-229 – Aneuploidy is a hallmark of senescence.

The authors find a mere 8% of their SA-bgal⁺ positive to be aneuploid, which does not justify the use of the term hallmark of senescence. Besides, extrapolation of aneuploidy indices to all 23 chromosome pairs is not a recognized or recommended parameter thereby discrediting their statement.

We have revised this statement (line 271; lines 285-290). The reduced number of chromosome pairs analyzed by FISH (3 pairs) limits the sensitivity of aneuploidy detection in the SA-bGal⁺ 84/87y population, even though a high number of cells can be quantified.

However, we agree with the Reviewer concern that extrapolation of aneuploidy indices of 3 chromosome pairs to 23 chromosome pairs is not a recommended parameter, which limits our data interpretation. Still, based on the definition of a ‘hallmark’, we provide evidence that aneuploidy meets the following criteria: 1) it is overtly present in elderly senescent cells; 2) its experimental aggravation (in our study through FoxM1 repression) accelerates senescence; and 3) its experimental amelioration (in our study through FoxM1-induced mitotic fidelity) delays senescence.

d. Line 235-236 - These results demonstrate that FoxM1 levels modulate aneuploidy-driven senescence.

It has not been experimentally demonstrated in the manuscript that aneuploidy drives senescence. The authors merely suggest a correlation between aneuploidy and senescence but the causative events and consequence are unclear. **The authors should follow the fate of cells with mitotic segregation defects to see if they get cell cycle arrested or continue into the next cell cycle to determine the effect of aneuploidy on the subsequent cell cycle.** (However a recent paper by Santaguida S et al, Dev Cell 2017, shows that aneuploid cells are cleared by the immune system in the subsequent cell cycle). I recommend toning down this conclusion or not making it at all as the evidence is missing to support it.

We accept the reviewer concern regarding the experimental demonstration that aneuploidy drives senescence. Even though previous studies have shown evidence supporting that constitutional aneuploidy and/or chromosomal instability trigger senescence (*Contestabile et al. 2009 Cell Prolif* 42(2):171-81; *Sheltzer et al. 2017 Cancer Cell* 31(2):240-55; *Baker et al. 2004 Nat Genet* 36:744-9; *Andriani et al., 2016 Sci Rep* 6:35218), we have now addressed the fate of elderly cells following mitotic segregation defects (anaphase lagging chromosomes or micronuclei generation), and we include new data in the revised manuscript showing that those cells become senescent (Fig. 7g-i). We used long-term time-lapse microscopy to follow individual elderly dividing cells expressing H2B-GFP, and we monitored the fate (cell death, cell cycling and senescence) of cells that mis-segregated chromosomes vs. cells without apparent mis-segregation after >48h. We found that daughter cells of defective mitoses consistently stopped cycling and often exhibited senescence biomarkers, such as SA- β -galactosidase activity and p21/53BP1 positivity. Importantly, cells without apparent chromosome mis-segregation mostly kept on cycling (72%), or if arresting (28%), only rarely exhibited senescence biomarkers (<25%). Thus, this experiment demonstrates that aneuploidization in an aging cell phenotype ultimately triggers permanent cell cycle arrest and full senescence.

We did not address whether the aneuploidy-driven full senescent cells are cleared by the immune system, even though it is an interesting question to pursue. Intriguingly, Santaguida et al., Dev Cell 2017 have shown that only aneuploid cells with complex karyotypes acquire a pro-inflammatory phenotype and are cleared by NK cells. However, we hypothesize that for elderly cells with accumulated genetic damage and an early pro-inflammatory phenotype, single chromosome aneuploidy might be sufficient to trigger a full-blown pro-inflammatory phenotype. Still, even if aneuploid elderly cells are cleared by the immune system *in vitro*, would this mean the same happens *in vivo*? If in one hand it would be in agreement with the mild aneuploidy levels found during natural aging, on the other hand we should consider the possibility that the immune system also ages (inflammaging).

e. Line 261-263 - The authors should be clearer, more complete and up-to-date with their discussion. The SASP is a complex program that is initiated and sustained by multiple pathways and stimuli, one of which is p38-MAPK. Please rephrase.

The paragraph has been revised in the new version of the manuscript (lines 340-347).

2. The authors use a parameter called aneusomy index throughout the paper but have not mentioned anywhere what it is or how is it measured, making it difficult to interpret any data

associated with it. Please mention in the methods section as well as the main text how it is defined and measured.

In the revised version of the manuscript please note that Fig. 1b, 1c and 6f are now Fig. 1a, 1b and 7f, respectively.

In Fig. 1a, we measured the prevalence of somatic aneuploidy by the ratio of aneusomic cells for chromosomes 7, 12 and 18 to the total cell count for a sample (>2500 nuclei) (aneusomy index). The same measurement applies to Fig. 5d, 6e, 7f and Supplementary Fig.9e. This methodology is described in Materials and Methods as ‘Interphase FISH’.

In Fig.1b, we measured the rate of chromosome mis-segregation (number of events in which two sister chromatids co-segregate to the same daughter cell) by combining a cytokinesis-block assay (using cytochalasin D for 24h) with FISH staining. This methodology (referred as Cyto D-FISH; Materials and Methods) allows the analysis of the reciprocal distribution of chromosomes between the daughter nuclei of a single mitotic division. Fig. 1b shows the percentage of binucleated telophases exhibiting chromosome mis-segregation over the total binucleated cell count for a sample (>200 BN cells). The same measurement applies to Fig. 5e, 6f and Supplementary Fig.9f.

We have now clarified this issue in the revised version of the manuscript, both in the main text (lines 90-94; 98-104) and in Material & Methods section.

a. Nonetheless, it appears in e.g. Figure 1b that 1-2% of cells in either age group are aneuploid, however, this number seems to be surprising low. Similarly, an increase of % of cells with aneuploidy in from 1% to 1.5% in FoxM1-deficient cells, as well as a reduction from ~1.8% to 0.5% after FoxM1-dNdK expression is extremely mild. It is questionable if these changes – although statistically significant – are indeed biologically significant.

We understand that the differences might seem surprisingly low, however it should be kept in mind that interphase FISH measurements only covered 3 chromosome pairs. Also, only mild levels of aneuploidy are associated with natural aging, which limits the use of single-cell sequencing as an option (still too expensive to cover the cell sample size that would be required to measure mild levels of aneuploidy). If extrapolating the aneuploidy levels into 23 chromosome pairs, the changes would be more perceptible, but we admit this is not a recommended procedure. Nevertheless, our experimental data, and in particular the data included in Fig. 7, show: 1) that SA-βGal+ cells purified from an elderly population are 3-fold more aneuploid than those purified from a young population; and 2) chromosome mis-segregation, which rate is 3-fold higher in elderly cells (Fig. 1b), triggers permanent cell cycle arrest and full senescence in the daughter elderly cells; we consider both findings 1) and 2) to give aneuploidy a biological significance. Moreover, cell-division errors or DNA breakage can result in chromosomes being partitioned into abnormal nuclear structures called micronuclei. Recently, it has been proposed that chromosomes released from micronuclei activate the DNA-sensing branch of the innate immune system (*Cai X et al. 2014 Mol Cell 54:289-96; Harding et al., 2017 Nature 548:466-70; McKenzie et al. 2017 Nature 461-65*) (lines 343-347). Thus, chromosome mis-segregation in elderly cells, even if mild, is likely sufficient to trigger the DNA-sensing pro-inflammatory response, and considering the self-reinforcing nature of the SASP, 2-fold changes in aneuploidy levels might actually translate into an highly significant immune response.

b. As for Figure 1c, I’m wondering how the authors addressed the issue of non-dividing, senescent cells in their assay. Growing, post-mitotic cells inhibited to undergo cytokinesis would appear as 4n cells, whereas non-proliferating, senescent cells would appear as mostly 2n population – I would assume these cells would be excluded from the analysis. Please, explain and set into context with the conclusion later in the manuscript stating that senescent, SA-βGal+ cells are the main source of aneuploidy in aged cultures.

We believe that the issue raised here by the Reviewer is being generated by the use of the term ‘post-mitotic’, which we accept was confusing. As explained above (point 2.), in Fig. 1b we measured the rate of chromosome mis-segregation (number of events in which two sister chromatids co-segregate to the same daughter cell) by combining a cytokinesis-block assay with FISH staining (CytoD-FISH). This method allows the analysis of the reciprocal distribution of chromosomes between the daughter nuclei of a single mitotic division. Fig. 1b shows the percentage of binucleated telophases (that we mentioned as post-mitotic cells) exhibiting chromosome mis-segregation over the total binucleated cell count for a sample (>200 BN cells). These BN cells arise from cells that divided during the 24h treatment with cytochalasin D (cytokinesis inhibitor). Note that senescence was never measured in this experimental setup using cytochalasin D, but instead in untreated cell cultures (2N populations). Later in the manuscript (Fig. 7a-f), the SA-βGal+ cells were sorted from untreated cell cultures (no treatment with cyto D), and aneuploidy measured by ‘interphase FISH’ analysis.

We have now clarified this issue in the revised version of the manuscript, both in the main text (lines 90-94; 98-104) and in Material & Methods section, and we replaced ‘Post-mitotic FISH’ by ‘CytoD-FISH’ in Fig. 1b.

3. In order to warrant interpretation of the results, the authors need to exclude the possibility of delayed/enhance proliferation rates and different apoptosis kinetics that may explain a difference in % SA-βGal+ cells or cells with aneuploidy, and other phenotypes. Do cultures with and without FoxM1, as well young v/s old fibroblasts have the same cell cycle time, population doubling time and apoptosis rates?

We don’t see why different proliferation rates should be excluded to warrant interpretation of the results. Decreased proliferation capacity has been widely reported for both senescent and aneuploid cell populations. Also, FoxM1 is known to regulate proliferation and senescence (*Smirnov et al., Aging 2016*). As shown in Supplementary Table 1 of the revised manuscript, the average population doubling time calculated for the cumulative passages in our experiments (≤ 5) shows significantly different proliferation rates between young and elderly cultures (49.1 ± 9.2 vs. 119.1 ± 32.0 hrs, $p < 0.05$). This was not accompanied by an increase in apoptotic cell death, as evaluated by cytometry of annexin V-positive cells in neonatal and octogenarian cell cultures at passages 2-3 (Supplementary Fig.4b,c). We have also quantified the mitotic indices for the young vs. old-aged cells (Supplementary Fig.4a), as well as for siFoxM1 and FoxM1dNdK cells (Supplementary Figs.6b and 7b), which further supports differences in proliferation rates. However, our experimental layout (Fig. 3a) was designed so that gene expression changes in aging cells were uncoupled from the effect of a lower mitotic index/decreased cell proliferation by analyzing the transcriptomes of equivalent numbers of young and elderly mitotic cells (lines 154-165). Also, the daughter cell fate of elderly cell mitoses was tracked individually to correlate the mitotic phenotype with the establishment of a full senescence phenotype (Figure 7g-i).

4. The authors aim to show that mitotic timing increases with age and that this change is SAC dependent and partly responsible for the mis-segregation phenotype observed with aging. While the authors show that over-riding the SAC by inhibition of Mps1 reduces mitotic timing in aged fibroblasts, they do not demonstrate that this reduces the segregation defects observed.

The authors would like to clarify this issue. We do show that mitotic timing increases with age and that the mitotic delay is due to SAC activity. However, we do not state the mitotic delay to be responsible for the mis-segregation phenotype observed with aging. Actually, the mitotic delay means that SAC is being activated to give cells time to correct defective chromosome-spindle attachments. The experiment using the Mps1 inhibitor was performed to exclusively demonstrate that the mitotic delay is due to SAC activation, as inhibition of the Mps1 SAC kinase reverts the mitotic delay in aged cells. Mps1 inhibition was never expected to reduce the segregation defects observed. In the contrary, Mps1 inhibition aggravates the

segregation defects, as aged cells with mis-attached chromosomes are being rushed to exit mitosis in the absence of checkpoint control. This was demonstrated in Supplementary Fig.4a, where we show that Mps1 inhibition in aged cells leads to more severe mis-segregation defects in comparison to young cells treated with Mps1 inhibitor. Nevertheless, because the assay is unacquainted outside mitosis research, and since it seems to have misled the Reviewer to the idea that Mps1 inhibition should rescue chromosome mis-segregation, we decided to exclude this data from the revised version of the manuscript.

a. In fact the data in Figure 2e showing that treatment with the Mps1 inhibitor induces segregation defects in the 87y old fibroblasts is in stark contrast to their conclusion.

We never expected neither concluded Mps1 inhibition to reduce segregation defects. As mentioned above, Mps1 inhibition was used as an experimental approach to demonstrate that aged mitotic cells are activating the SAC due to chromosome-spindle attachment defects.

b. Further, their conclusion of the delay being SAC dependent is weak as one would expect to see SAC hyperactivity, measured by the time arrested in mitosis after STLC treatment. I recommend checking SAC activity using spindle poisons (nocodazole or taxol) rather than using a kinesin-5 inhibitor.

We have checked SAC activity using both taxol and STLC (kinesin-5 inhibitor). Both treatments induce chronic activation of the SAC and prolonged mitotic arrest, in the absence (taxol 100nM) and presence (STLC) of spindle microtubules. Duration of the mitotic arrest under high concentration of spindle poisons reflects the “strength” of the SAC (*Weaver and Cleveland, 2005*). Interestingly, we found no significant difference in the time aged cells take to slip out of mitosis in comparison to younger cells, upon prolonged arrest induced by taxol or STLC treatments (Supplementary Fig.3g,h). This suggests that SAC activity in old cells is as “strong” as in younger cells (see also discussion section, lines 370-373).

c. Also, if the increased time is indeed responsible for the mitotic delay, can the segregation defects observed in aged fibroblasts be recapitulated in young fibroblasts by increasing mitotic timing using APCIN or PROTAME?

We performed the experiment suggested by the Reviewer. We treated neonatal cells with 10 μ M PROTAME (APC/C inhibitor) for 2 hours. Untreated cells were used as control. Next, medium was washed out and replaced by medium containing cytochalasin D (cytokinesis inhibitor) (Reviewer 2 Figure 1). After 2h, cells were fixed and stained with FISH centromeric probes against 3 chromosome pairs. As explained in major point 2 above, CytoD-FISH allows the measurement of chromosome mis-segregation events between daughter nuclei of cells that divided during the cytochalasin D treatment. The Reviewer can appreciate that chromosome mis-segregation is not induced by the mitotic delay.

Reviewer 2 figure 1. a) Experimental layout. Neonatal fibroblasts were treated with proTAME for 2h. Cells entering mitosis in the presence of proTAME will arrest due to inhibition of the APC/C. Untreated cells were used as control. Medium was replaced by medium containing cytochalasin D to inhibit cytokinesis. Cells were fixed for FISH staining with centromeric probes against 3 chromosome pairs. b) Binucleated cells with chromosome mis-

segregation were quantified from a total n number of binucleated cells as indicated. Two tailed χ^2 statistical test. n.s., not significant.

5. It is very puzzling why the authors have made use of an Mps1 inhibitor in Figure 2 to investigate the main segregation defects seen in young v/s old fibroblasts. This investigation should instead be performed on unperturbed mitoses. The classification of segregation defects as mild and severe is also confusing. Why are cells with bipolar anaphase without cytokinesis failure being called mild mis-segregation and cells with cytokinesis failure being called severe? Some mis-segregation events or their consequences (such as cytokinesis failure, micronuclei formation and spindle mis-positioning) were quantified on a cell to cell basis, important mis-segregation events like chromosomal misalignment, lagging chromosomes and others were not quantified. Please, add quantification of these with appropriate statistics.

In the revised version of the manuscript we now provide quantifications for chromosome alignment (congression) delay and lagging chromosome phenotypes based on spinning-disk confocal live-cell imaging analysis (Fig. 2e,f).

In the previous version of the manuscript, we quantified these phenotypes indirectly based on the following rationales:

1) If older cells have chromosome alignment delay, these cells will exhibit extended mis-segregation defects when rushed to exit mitosis through inhibition of SAC activity with the Mps1 small molecule inhibitor. Considering that even the young cells most often mis-segregate chromosomes when treated with Mps1i, the mis-segregation defects were classified into mild or severe depending on the mitotic exit phenotype, i.e. bipolar anaphase generating two daughter cells or slippage generating one 4N daughter cell. However, as mentioned in major point 4, we decided to exclude this data from the revised version of the manuscript, as the rationale seems confusing. Direct quantification of congression delay has been performed on spinning-disk confocal records.

2) If older cells exhibit lagging chromosome phenotypes, these cells will more often generate micronuclei (*Thompson and Compton, 2011*). Micronuclei were quantified from fixed cell samples (Fig. 2g).

Also, the authors mention spindle mis-positioning to be a mitotic defect. They need to demonstrate how this affects their phenotype? For quantifying % of cells with spindle mis-positioning in Figure 2g, please add the cut-off criteria (angle) for scoring a normal vs. mis-positioned spindle.

We agree that is largely unknown how spindle mis-positioning in relation to the substrate surface contributes to mitotic errors (chromosome mis-segregation). We found only few references supporting this correlation. Ertych N et al. (*Nat Cell Biol 2014*) have shown increased microtubule assembly rates to be associated with transient abnormalities in mitotic spindle geometry in relation to substrate surface that promote generation of lagging chromosomes. Also, Vitiello E et al. (*Nat Commun 2016*) have shown DLC2 or Kif1B depletion to promote microtubule stabilization, defective spindle positioning, chromosome misalignment and aneuploidy. These references are now included in the text.

In line with the poor evidence for correlation between spindle mis-positioning and chromosome mis-segregation, we decided to exclude the detailed analysis of spindle mis-positioning from Supplementary Fig.4, as we do not wish to overstress this phenotype.

In Fig. 2g, the spindle positioning was evaluated from spinning-disk confocal live-cell movies, whereas the angle parameter (Supplementary Fig. 4b-d, which is now data not shown) was measured in fixed cell samples. The criteria for scoring normal vs. mis-positioned spindle in Fig. 2g was based on the unparallel positioning of spindle poles to the growth surface under live cell imaging, followed by asymmetric and asynchronous adherence of the daughter cells. Angle calculation from live cell movie analysis would be challenging as one of the poles was often outside the z-stacks, and again we don't feel the phenotype is sufficiently relevant for such detailed analysis.

6. Concerning the discussion: The authors fail to see that senescence is established as tumor suppressive mechanism and that induction of senescence in the context of aging and diseases is first and foremost a mechanism induced after cellular damage. This program restricts damaged cells from propagating and prevents neoplastic transformation.

The authors should be more clear in this and should not mix up the difference between removal of established senescent cells, a strategy that has been shown to ameliorate diseases and age-related phenotypes, with the prevention of senescence and its deleterious risks.

We understand the Reviewer concern, and we consider we have better clarified this issue in the revised version of the manuscript. As the Reviewer mentions above, senescence is important to restrict damaged cells from propagating. However, the rationale behind our model is that FoxM1 expression acts to improve cellular fitness (decreased genetic damage, decreased pro-inflammatory phenotype), thereby allowing elderly cell populations to be replenished by fitted cells rather than proliferating damaged cells (lines 388-390; 398-402).

The authors should also discuss the possible side effects of FoxM1 overexpression as their therapeutic strategy to prevent senescence and aging associated disorders. The authors try to substantiate their statements by citing papers with transgenic overexpression of FoxM1 not having any tumorigenic side effects in a model of lung tissue damage and liver regeneration. However, in those papers the mice were not aged long enough to be able to comment on the tumorigenic potential of FoxM1 overexpression, thus making it difficult to believe that FoxM1 overexpression might have no unpleasant consequences.

We accept the Reviewer concern, and we have revised the discussion section to include the possible side effects of FoxM1 overexpression (lines 393-399). Nevertheless, we should not exclude the potential of this strategy as so far FoxM1 has only been shown to be tumorigenic if tumor suppressor genes are lost or oncogenic mutations (such as K-RASG12D) are present (Wang IC *et al.*, *Dev Biology* 2010; Cai Y *et al.*, *J Biol Chem* 2013).

[Redacted]

Again, based on concerns raised in point 1, the overall thought of the discussion having found a new mechanism for senescence induction and novel therapy complementary to senolytics should be toned down.

We believe we followed the Reviewer suggestions.

7. Supplementary tables are not helpful and lack completeness for the reader. Please add gene names to all lists, and their fold changes and p-value (FDR) etc.

Supplementary tables are now revised accordingly to the Reviewer's suggestion.

Minor concerns:

1. It is appreciated that the authors perform several unbiased transcriptomic RNA-seq experiments to assess the transcriptional pattern of aged, FoxM1-deficient and FoxM1-re-expressing systems. One trade-off is the very low number of replicates (2) which makes interpretation of the data challenging. As heat maps that display a min/max Row Z-score based on normalized counts can be both helpful and misleading, please also indicate the fold change expression level between control and sample in the same heat maps.

Fold changes (2logFC) for each gene between control and sample have been added to all heat maps to more faithfully show changes in gene expression.

For the heat map in Figure 4j, 5h, gene names are missing.

Gene names have been added to all heat maps.

2. How do FoxM1 protein levels in 10y fibroblast after siFoxM1 compare to aged fibroblast? To which age group does the knock-down correspond to? A direct comparison of the samples

on one Western Blot will address this. This may also give indications if the phenotype of FoxM1 deficiency is solely responsible for the age-related mitotic phenotype or if FoxM1 is one of multiple contributing factors.

In Fig. 4c, we show that FoxM1 protein levels in 87y fibroblasts are 17% in comparison to neonatal and 20% in comparison to 10y. However, as shown in Fig. 4d, when averaging 3 independent western blot experiments, FoxM1 levels in 87y fibroblasts are $25\% \pm \text{sd}$ in comparison to neonatal and $34\% \pm \text{sd}$ in comparison to 10y (western blot analyses typically exhibit variability). In Supplementary Fig.6a, we show that FoxM1 levels in 10y fibroblasts depleted with siFoxM1 are 18%, and average of three independent experiments is now shown in Supplementary Fig.6e ($34\% \pm \text{sd}$). Thus, the knock-down closely corresponds to the octogenarian age.

Nevertheless, we followed the Reviewer's suggestion and included a direct comparison of the samples on one western blot (Supplementary Fig.7j,k). The Reviewer can appreciate that FoxM1 levels in 87y fibroblasts and 10y siFoxM1 fibroblasts are 45.1% and 44.9% in comparison to levels in 10y fibroblasts. FoxM1 appears therefore to contribute primarily for the age-related mitotic phenotype, further supported by the RNA-seq analysis showing significant overlap between altered mitotic genes in aged, FoxM1-deficient and FoxM1-re-expressing systems (Fig. 6c).

3. The SASP consists of a plethora of factors and is cell type- and senescence-stimulus specific. Please expand the presentation of SASP factors in the RNA-seq datasets and display in the heat maps.

In the revised version of the manuscript we expanded the analysis of SASP factors in the RNA-seq datasets and the display in the heat maps (Fig. 3h, 5h and 6h). An expanded list of 51 genes was interrogated in the RNA-seq datasets. This list is the overlap between the 41 SASP genes reported in *Coppé et al., Plos Biol 2008* as significantly altered in senescent vs. pre-senescent fibroblasts (Dataset S4 in this reference), and the 42 genes (SASP genes and a few senescence-associated genes) assayed in *Hernandez-Segura et al., Curr Biol 2017* for senescent vs. proliferating fibroblasts (Supp. Figure S1B in this reference). RNA-seq datasets for the analysis of SASP factors are provided in Supplementary Dataset 2. Also, a Venn diagram displaying the overlaps between SASP genes significantly altered in HDF 87y, HDF10y siFoxM1 and HDF87y FoxM1-dNdK, is shown in Supplementary Fig.8e.

Also, the variability between the two replicates seems high and some SASP factors seem to be down-regulated, which makes the data less convincing. Employing additional methods to assess the SASP would help convince the reader. Perhaps Western Blotting, Immunofluorescence analysis or antibody arrays could be used.

Aware that the transcriptome of senescent cells is highly heterogeneous, depending of the cell type and senescence-stimulus, we additionally extended our analysis into 55 genes recently identified as comprising a 'senescence core signature' (*Hernandez-Segura et al., Curr Biol 2017*). Interestingly, we found 19 out of the 55 genes to be differentially regulated in the RNA-seq dataset from mitotic dermal fibroblasts of octogenarian donors (Supplementary Dataset 3). This is in agreement with the findings reported by Hernandez-Segura et al. (*Curr Biol 2017*) for lung tissues of patients with idiopathic pulmonary fibrosis where the number of senescent cells is small, and still 10 genes of the core signature were identified as differentially regulated. In the revised version of this manuscript, we now include real time-PCR analysis to further demonstrate alterations in the expression levels of SASP genes and genes of the 'senescence core signature' (Fig. 3j; Supplementary Figs.6i and 7i).

4. Please be consistent in the use of the qPCR reference gene. Figure 4c uses TBP whereas in Suppl. Figure 2 TBP and GAPDH was utilized.

For sake of consistency we now show qPCR analysis in both human and mouse fibroblasts using TBP as reference gene. The data in Supplementary Fig.2 did not change significantly

after removing GAPDH as reference gene. TBP is a robust reference gene that exhibits an efficiency of 100% in human cells qPCR and an efficiency of 95% in mouse cells qPCR.

5. In the text from line 95-100, the authors talk about measuring aneuploidy in post mitotic cells. However the data corresponding to the text (Fig1c) shows BN cells with mis-segregation. Can the authors clarify if they are measuring aneuploidy by FISH or looking at segregation defects by live cell imaging?

Figure 1b shows FISH in binucleated cells. In contrast to FISH analysis in asynchronous populations (Fig. 1a) where we measured the prevalence of somatic aneuploidy by the ratio of aneusomic cells for chromosomes 7, 12 and 18 to the total cell count per sample (>2500 nuclei), in Fig. 1b we measure the rate of chromosome mis-segregation (number of events in which two sister chromatids co-segregate to the same daughter cell) by combining a cytokinesis-block assay with FISH staining. This method allows the analysis of the reciprocal distribution of chromosomes between the daughter nuclei of a single mitotic division. Fig. 1b shows the ratio of mis-segregation events to a total of $n > 200$ binucleated cells. We mentioned to 'post-mitotic cells' in Fig. 1b since in this assay we are only measuring binucleated cells that arise from mitotic events taking place during the 24h treatment with the cytokinesis inhibitor (cytochalasin D).

We have now clarified this issue in the revised manuscript (lines 90-94; 98-104).

6. In Supp. Figure 2, the authors make use of murine adult fibroblasts to corroborate the findings found with human fibroblasts. However the qPCR data showing FoxM1 decline is not significant and should be mentioned and clarified in the text (line 166).

The statistical analysis of qPCR data showing FoxM1 decline was previously missing in Supplementary Figure 2f (formatting lapse) and has been included in the revised version of the manuscript. The Reviewer can now appreciate that significant differences were actually found (Supplementary Fig.2h).

6b. There is also no Western Blot showing FoxM1 decline with age in MAFs.

To further support the qPCR data, we have now included western blot evaluation of FoxM1 protein levels in MAFs (Supplementary Fig.2f,g). FoxM1 protein levels decline with age, with significant differences found.

6c. Besides, murine ear fibroblasts have a tendency to become polyploid early in culture. Could the authors please provide ploidy status of the MAFs used in their experiments by e.g. Propidium Iodide staining?

We accept the reviewer concern and we analyzed the ploidy status of the MAFs used in our experiments by measuring the nuclear area on fixed cell preparations previously used for quantitative analysis of senescence biomarkers (Supplementary Fig.2d,e). Polyploid cells can be both mononucleated or binucleated. The criterium used for nuclear area quantification under Image J/Fiji was defined so that binucleated cells were included (the water-shed function splitting the two adjacent nuclei was removed so that they appear as single nucleus). We found the nuclear area to distribute within similar values in all MAF samples, with no significant differences found in comparison to 4w MAFs (Reviewer 2 figure 1a). The mean nuclear area of binucleated cells (as determined in Reviewer 2 figure 1b) was established as threshold value above which nuclei were considered polyploid. Indeed, we found highly proliferative young MAFs to have a tendency to become polyploid early in culture (~25% in 4w MAFs), whereas low proliferative elderly MAFs exhibited lower polyploidy levels (<10% in 1.5y MAFs), suggesting that polyploidy is unlikely to contribute significantly to the increased percentage of senescent cells found in elderly MAF cultures.

Reviewer 2 figure 1. Nuclear area analysis of mouse adult fibroblasts. a) Histogram of nuclear area distribution in MAF cultures from different age mice. Criteria used for quantitative analysis using ImageJ/Fiji software: nucleus size (125-1000 μm^2) and circularity (0.5-1). Nuclear mask was applied so that binucleated cells were also included as one single nucleus (no water-shed). 4w MAFs used as reference for statistical analysis with Kolmogorov-Smirnov test. ns, not significant. b) To establish a threshold value of nuclear area above which cells are likely polyploid, 200 cells of each sample were randomly selected and the mean nuclear area of mononucleated vs. binucleated cells was determined. A nuclear area higher than 600 μm^2 was established for polyploidy. c) Percentage of cells with nuclear area >600 μm^2 , considered to be polyploid (either mono- or binucleated). Statistical analysis was performed by two-tailed χ^2 statistical test against 4w MAFs.

Reviewer #3 (Remarks to the Author):

The manuscript by Macedo et al. describes an investigation into the mitotic phenotype of aging fibroblasts. By surveying a panel of cells from donors with different ages, as well as fibroblasts from an individual with progeria, they report that the accuracy of chromosome segregation decreases with age, and they attribute this phenotype to a suppression of FOXM1-dependent transcription during mitosis. By over-expressing FOXM1, they report that they can rescue mitotic accuracy in aging cells, suggesting an interesting strategy to counteract age-related cellular decline.

Overall, the paper makes a strong case for its central thesis, and the experiments performed are relevant and well-described. The parallel experiments in human and mouse cells in particular are helpful in ruling out genetic confounders. Additionally, as a number of questions have been raised in recent years regarding the accuracy of FISH, it is reassuring to see that the fixed cell experiments were scored blindly (perhaps this could be mentioned in the main text, in addition to the methods section?).

We now mention in the main text (line 92) and in the Materials & Methods (line 645) that samples were scored blindly for FISH staining.

My major question concerning this manuscript is the reported downregulation of mitotic genes in aging mitotic cells. This result is interpreted as implying that aging mitotic cells express lower levels of these genes. An alternate explanation would be that cell synchrony in aging cells is worse, and the population of aging mitotic cells had slightly more G1 or S-phase cells that were collected for RNA-Seq. I wonder about this possibility, given what the authors' interpretation would imply.

We appreciate the reviewer's concern and we now include data to clarify this issue (Fig. 3b,c). We quantified the percentage of mitotic cells isolated following shake off (mitotic cell detachment) of neonatal and octogenarian cell cultures treated with STLC (Eg5 inhibitor) (Fig. 3a). Our isolation procedure yielded >95% mitotic cells from both neonatal and octogenarian cell cultures, as confirmed by microscopic scoring of metaphase spreads (Fig. 3b,c). It is indeed true that cell synchronization (mitotic enrichment) is limited in elderly cultures, in agreement with their lower proliferative capacity. However, in order to collect as much mitotic elderly cells as neonatal ones for RNA sequencing, we used a higher number of cell culture flasks of octogenarian fibroblasts treated with STLC. Thus, our procedure uncoupled mitotic gene downregulation in aging cells from the effect of a lower mitotic index by analyzing the transcriptomes of equivalent young and elderly mitotic cell populations (lines 154-162).

Various gene expression thresholds exist that control cell cycle transitions (e.g., cyclin B/CDK levels for mitotic entry or CDC20 levels for anaphase). The authors' model suggests that these thresholds are lower for aging cells, perhaps because there is a stoichiometric decrease in all factors involved. This would raise a number of interesting questions about the biology of aging, which I have no doubt that the authors are following up. For this current manuscript, I think that it would be informative to present further evidence that aging cells express low levels of mitotic transcripts during mitosis. In particular, I think that the authors could perform quantitative IF experiments for one or two mitotic markers (cyclin B, FOXM1, CDC20, etc.) and show that in single aging cells these genes are in fact expressed at low levels during mitosis.

We followed the reviewer's suggestion and performed quantitative IF analysis for FoxM1, cyclin B and Cdc20. The data show that single elderly mitotic cells exhibit significantly lower levels of these proteins in comparison to young mitotic cells (Fig. 4h-j; Supplementary Fig.5).

Besides that, a few minor comments:

1) Concerning premature aging, the BubR1 mouse is really an exception, and most other

aneuploid mouse models (CDC20, MAD2, cohesin, etc.) do not show a progeroid phenotype. This fact is mentioned in the discussion, but I believe that it is mildly misleading to present the BubR1 mouse in the introduction as representative of CIN models.

We revised the introduction paragraph so that the BubR1 mouse shows up as single example of a CIN model exhibiting a progeroid phenotype (we deleted the previous sentence “Studies of aneuploidy-prone mouse models exhibiting increased rates of chromosome mis-segregation uncovered a unanticipated nexus with the rate of aging and the development of age-related pathologies^{11-14”}).

2) It has previously been shown that single chromosome aneuploidy is sufficient to promote senescence, particularly in the case of Down syndrome (e.g., Contestabile 2009 and Sheltzer 2017). Thus, the authors’ claim that while CIN has been shown to promote senescence, it is unknown whether low levels of aneuploidy do the same is not strictly accurate.

We have rephrased this paragraph and included the references suggested by the Reviewer (lines 273-280). Still, our data disclosed the interesting finding that increased pro-inflammatory response is a characteristic of aged cells that are still proliferating, and that single chromosome mis-segregation events in these cells are sufficient to trigger transition into permanent cell cycle arrest (full senescence).

3) Though not necessary for this paper, it would be very interesting if BubR1 siRNA blocked the rejuvenating effects of FOXM1 over-expression. Have the authors tested this?

We have not tested this. We acknowledge the Reviewer for this suggestion that we would like to ascertain in the near future. Being BubR1 a downstream target of FoxM1 transcriptional activity, and thus far the single mitotic gene for which an hypomorphic mouse model has disclosed a link between aneuploidy and premature aging, it is reasonable to address whether the rejuvenating effect of FoxM1 over-expression is being through BubR1. We can only mention to the Reviewer that when over-expressing BubR1 to similar levels than those found following FoxM1 over-expression (2-2.5 fold increase in BubR1 levels), the aneuploidy and senescence levels are not significantly rescued by BubR1 OX whereas by FoxM1 OX are (our unpublished data). This suggests that FoxM1 likely acts on several mitotic targets to ameliorate those phenotypes and that higher BubR1 levels are needed to produce similar effect (indeed the case of the BubR1 OX mouse model; *Baker et al. 2013 Nat Cell Biol 15:96-102*).

Overall, the authors present a solid manuscript that will significantly contribute to the literature on mitosis and aging. If the authors can present an independent method confirming the low expression of mitotic genes in aging cells, I would fully support publication. We acknowledge the Reviewer for supporting the publication and, as mentioned above, we present the IF analysis of single cells, as well as western blot analysis of mitotic cell subpopulations, confirming the low expression of mitotic genes in aging cells.

REVIEWERS' COMMENTS:

Reviewer #2 (Remarks to the Author):

The authors should be commended in that they have satisfactorily addressed several of the concerns that were raised. However, the authors' claim that aneuploidy drives senescence remains unsupported by compelling evidence and needs further experimentation. The experiments on which the claim is based were all done in the context of FoxM1 insufficiency. This does not exclude the possibility that FoxM1 transcriptionally controls the senescence program independent of aneuploidy or chromosome segregation errors. And these experiments do not demonstrate that aneuploidy per se engages the senescence program.

It is therefore recommended that the authors perform experiments with different primary human cell lines with normal FoxM1 levels (for instance RPE-1, IMR-90, foreskin fibroblasts) to test if segregation errors caused by pharmacological or genetic manipulation of known mitotic regulators such as Mps1, BubR1, Mad2, CENP-E, AurB is sufficient to drive cells into senescence. The authors could also use multiple established MEF models of aneuploidy to further corroborate their claim that aneuploidy itself is driving senescence rather than a aneuploidy-independent role of FoxM1 in the senescence program. If aneuploidy were the senescence-inducing stressor, what might be its mechanistic basis?

Other remaining points:

1. The authors show in their RNA seq data set that all genes involved in forming the mitotic checkpoint complex and thereby regulating the SAC are low when FoxM1 levels are low (Bub3, BubR1, Mad1, Mad2 and Cdc20). It is therefore puzzling as to how these cells still have no change in SAC strength or arrest. CyclinB1/2 are also found to be decreased in their FoxM1 insufficient cells. Together, several genes which control mitotic timing are affected but the authors still observe an increase rather than a decrease in mitotic timing in aged fibroblasts. These discrepancies should be explained.

2. In the previous round of review, I inquired about spindle mispositioning as a type of segregation error. The authors have not been able to convincingly argue for its role in missegregation, yet still include that data in figure 2 and say it is a potential error. Just because the division of cells occurs in different Z planes in the imaging dish, this is not sufficient to call it a segregation error and we recommend not including or emphasizing this in their text. Also, the chromosome congression delay/chromosome misalignments assessed by the authors seem to be in early metaphase (and not just before anaphase onset, as is traditionally done by others studying mitotic errors) and they eventually seem to be corrected as depicted by their representative images. This does not suffice to be included as a segregation error as it would not cause the daughter cells to be aneuploid. The authors should therefore clarify if the misalignments seen persist till anaphase onset or if they get corrected prior to anaphase. If they do get corrected, it does not warrant inclusion in a type of chromosome segregation error. Together with these 2 points, the authors should therefore accordingly correct the data shown in fig 2f.

3. I mentioned that the aneuploidy rates are extremely low in FoxM1 insufficient cells and questioned its biological significance. Meaningful information about aneuploidy rates is a prerequisite given the claims of the paper. If single-cell DNA sequencing is too expensive, one could conduct chromosome counts on metaphase spreads instead.

4. The staining of beta-gal, p21 and 53BP1 in figure 7h (for the cell that has undergone a segregation error) seem very unconvincing for a cell that is apparently senescent and this casts doubts on the data shown in that figure. Could the authors use other markers for senescence such as p16 or LaminB to justify their claim?

Also, according to the data in figure 7, only half of the cells that have undergone a segregation

defect are positive for all 3 senescence markers tested (b-gal, p21 and 53BP1). This thereby suggests that aneuploidy does not drive senescence but there might be other unknown events that cause the cell to arrest (p53 status in the cells). The authors should therefore show more convincing data in order to make this conclusion, or not make it all.

Reviewer #3 (Remarks to the Author):

I am satisfied with the revisions and support publication of the manuscript.

Rebuttal to Reviewer 2

Reviewer #2's Comments

The authors should be commended in that they have satisfactorily addressed several of the concerns that were raised.

We appreciate the Reviewer's comment that we have satisfactorily addressed several of his/her concerns, and below we answer to the issues still raised by the Reviewer.

However, the authors' claim that aneuploidy drives senescence remains unsupported by compelling evidence and needs further experimentation.

We would like to stress that we claim that aneuploidy drives transition into full-blown senescence (line 30, ...*aneuploidy to be a key player in the progression into full senescence phenotypes*), which is different from claiming that aneuploidy drives senescence (as if senescence would only be due to aneuploidy). In agreement with data reported during our previous round of revision (*Hernandez-Segura A et al., Curr Biol 2017*), we found that senescence is an evolving phenotype before the cell reaches an irreversible cell cycle arrest with overt stereotyped phenotypic changes (full-blown senescence). We provide compelling evidence that elderly cells that still divide, already exhibit a senescence core gene expression signature (RNA-seq analysis of elderly mitotic cells) that ends up in a permanent cell cycle arrest full senescent phenotype if chromosome mis-segregation takes place (as shown in Fig. 7). For clarity, we have therefore rephrased the paragraph in lines 351-356 of the Discussion section: *“Using innovative experimental layouts, such as aneuploidy measurement in FACS-sorted senescent cells with high SA- β -gal activity and long-term live-cell imaging followed by correlative fixed-cell analysis of mitotic daughter cell fate, we have demonstrated that aneuploidization ultimately triggers permanent cell cycle arrest and full senescence in aged cells already transcriptionally ‘primed’ for senescence, thus acting as a facilitator of aging.”*

The experiments on which the claim is based were all done in the context of FoxM1 insufficiency. This does not exclude the possibility that FoxM1 transcriptionally controls the senescence program independent of aneuploidy or chromosome segregation errors.

We do not exclude that FoxM1 transcriptionally activates the senescence programme. Actually, we show that cells with FoxM1 insufficiency exhibit a senescence core gene expression signature in mitosis before the chromosome mis-segregation events have occurred (this has now been added to the discussion section, lines 400-402). However, through our challenging experiment shown in Fig. 7, long-term live-cell imaging of elderly dividing cells and correlative fixed-cell analysis of the daughter cell fate, we found that whereas in the absence of chromosome mis-segregation only 25% out of the 30% of elderly daughter cells that stop cycling exhibit senescence biomarkers (i.e., 7.5% in total), in the presence of chromosome mis-segregation all cells stop cycling and 50% exhibit senescence biomarkers. Thus, chromosome segregation errors in cells transcriptionally ‘primed’ for senescence (FoxM1 repression) aggravate the senescence phenotype. This is why we show an ‘aging-aneuploidy-aging’ cycle in Fig. 8 model.

And these experiments do not demonstrate that aneuploidy *per se* engages the senescence program. It is therefore recommended that the authors perform experiments with different primary human cell lines with normal FoxM1 levels (for instance RPE-1, IMR-90, foreskin fibroblasts) to test if segregation errors caused by pharmacological or genetic manipulation of known mitotic regulators such as Mps1, BubR1, Mad2, CENP-E, AurB is sufficient to drive cells into senescence. The authors could also use multiple established MEF models of aneuploidy to further corroborate

their claim that aneuploidy itself is driving senescence rather than a aneuploidy-independent role of FoxM1 in the senescence program.

As abovementioned, our experiments are not meant to demonstrate that aneuploidy *per se* engages the senescence program, but rather that chromosome mis-segregation in an aged cell with early senescent phenotype induces the transition into a late senescent phenotype. Based in previous reports, it is unlikely that aneuploidy itself drives senescence (trisomic MEFs do not activate p53, *Tang et al. Cell 2011*), even though constitutional aneuploidies are associated with premature senescence/aging (*Contestabile et al., Cell Prolif 2009; Sheltzer et al, Cancer Cell 2017*) apparently by contributing to the evolution of more complex karyotypes (*Santaguida et al., Dev Cell 2017; Nicholson et al., Elife 2015*). We believe that the experiments suggested by the Reviewer, pharmacological or genetic manipulation of chromosome segregation in primary cells with normal FoxM1 levels, are available from previous reports (Baker DJ et al., *Nat Genet 2004*; Andriani GA et al., *Sci Rep 2016*) (as mentioned in lines 274-276, *Previous studies have found cellular/mouse models of constitutional aneuploidy or chromosomal instability (CIN) to prematurely senesce/age*^{11,49-53}). However, CIN-inducing single gene editing/drug treatments lead to highly variable aneuploidy levels that might determine whether cells become senescent or not. We think our experimental setup shown in Fig. 7, long-term live-cell imaging of elderly cells followed by fixed-cell analysis of senescence outcome, provides a nice internal control, which are the elderly cells that did not mis-segregate chromosomes, and continued cycling, or if not, barely developed senescence biomarkers. So, from our experiments we claim that in the case of elderly dividing cells, and as proposed in Fig. 8 model, an aneuploidy-independent senescence program (that we show as FoxM1-dependent) is already scheduled before chromosome mis-segregation, and that chromosome mis-

segregation only contributes as an ultimate trigger to a full-blown senescent phenotype (revised paragraph in lines 316-319). We actually stand for an aneuploidy-independent role of FoxM1 in the senescent program (lines 397-399, *This appears to be the case of FoxM1. FoxM1 repression translates into mild aneuploidy levels and may also further act by counteracting age-associated cellular damage caused by genotoxic and oxidative stresses*^{45,65}), and we expect to disclose additional data in another opportunity. There is no sentence claiming that aneuploidy *per se* engages the senescence program (please check lines 30, 67, 272, 290, 298-299, 315-319, 330 and 354-356, where we have purposely referred to aneuploidy as contributing to transition in full senescence in elderly early senescent cells).

If aneuploidy were the senescence-inducing stressor, what might be its mechanistic basis?

Referring to aneuploidy caused by a chromosome mis-segregation event in an early senescent aged cell, we discuss a possible mechanistic basis even though this is beyond the scope of the current manuscript. *This is in line with recent findings supporting that micronuclei generated during defective mitoses are a key source of immunostimulatory cytosolic DNA that triggers a cGAS-STING-mediated pro-inflammatory response*⁵⁹⁻⁶¹. *Whether the evolution of SASP during natural aging is dependent of cytosolic DNA signaling is an interesting question to address in the future.* (lines 344-348, Discussion section)

Other remaining points:

1. The authors show in their RNA seq data set that all genes involved in forming the mitotic checkpoint complex and thereby regulating the SAC are low when FoxM1 levels are low (Bub3, BubR1, Mad1, Mad2 and Cdc20). It is therefore puzzling as to how these cells still have no change in SAC strength or arrest. CyclinB1/2 are also

found to be decreased in their FoxM1 insufficient cells. Together, several genes which control mitotic timing are affected but the authors still observe an increase rather than a decrease in mitotic timing in aged fibroblasts. These discrepancies should be explained.

In lines 369-374 of the Discussion section we provide explanation for these discrepancies: *One possibility might be the balanced/stoichiometric repression of most mitotic genes. Furthermore, parallel mechanisms might buffer aging-mediated repression of mitotic transcripts. For example, even though we found SAC genes to be downregulated in elderly cells, SAC functionality was similar in young and old cells, suggesting that concurrent downregulation of genes contributing to proteasome activity (e.g. Cdc20 and APC/C subunits) might buffer SAC gene repression.*

We understand that it seems counterintuitive that aged fibroblasts exhibit a mitotic delay if SAC proteins, as well as cyclin B1, are downregulated. However, the Reviewer should also appreciate that, on the other hand, Cdc20 and APC/C subunits are also downregulated, thereby counteracting the weakened checkpoint signalling. Moreover, several reports have previously shown that partial depletion of certain checkpoint proteins compromises the establishment of correct chromosome-microtubule attachments and chromosome congression (*Lampson and Kapoor, NCB 2005; Meraldi and Sorger, EMBO J 2005; Logarinho et al., Mol Biol Cell 2008; Logarinho et al., Cell Cycle 2008*), which translates into a mild delay before anaphase onset similar to that observed in aged fibroblasts. The dual role of these SAC proteins in chromosome-MT attachment and spindle checkpoint, ends up generating a situation in which their partial depletion leads to increased activation of the checkpoint due to attachment defects despite their lower levels to generate the checkpoint signalling. This explanation has now been added in the discussion section (lines 379-382).

2. In the previous round of review, I inquired about spindle mispositioning as a type of segregation error. The authors have not been able to convincingly argue for its role in missegregation, yet still include that data in figure 2 and say it is a potential error. Just because the division of cells occurs in different Z planes in the imaging dish, this is not sufficient to call it a segregation error and we recommend not including or emphasizing this in their text.

After the first round of revision, we followed the Reviewer's suggestion not to overemphasize this phenotype, and the Reviewer can appreciate that there is no mention to this phenotype as a type of segregation error. The only misleading sentence is perhaps "*Overall, the data indicated that aging triggers abnormalities at several mitotic stages, in agreement with the increased levels of aneuploidy found*", lines 149-151, that we would like to replace by "*Overall, the data indicated that aging triggers abnormalities at several mitotic stages, suggesting for an age-associated loss of mitotic proficiency*".

Also, the chromosome congression delay/chromosome misalignments assessed by the authors seem to be in early metaphase (and not just before anaphase onset, as is traditionally done by others studying mitotic errors) and they eventually seem to be corrected as depicted by their representative images. This does not suffice to be included as a segregation error as it would not cause the daughter cells to be aneuploid. The authors should therefore clarify if the misalignments seen persist till anaphase onset or if they get corrected prior to anaphase. If they do get corrected, it does not warrant inclusion in a type of chromosome segregation error.

After the first round of revision we clarified this, thereby excluding chromosome congression delay as a segregation error: *In agreement with proper SAC functioning*

(Supplementary Fig.3g,h), aged cells entering anaphase with unaligned chromosomes were never observed, lines 144-145.

Together with these 2 points, the authors should therefore accordingly correct the data shown in fig 2f.

We believe data in Fig. 2f should be included as we only refer to mitotic defects (not to segregation errors). Overall, Figure 2 aims to describe the thus far elusive mitotic phenotype of elderly dividing cells, rather than describing the mitotic defects that specifically contribute to aneuploidy. We agree with the Reviewer that only anaphase laggards likely contribute to aneuploidy, but the chromosome congression and spindle positioning defects are described together with the anaphase lagging chromosomes to highlight the pleiotropic mitotic phenotype of aged/FoxM1-insufficient cells, which nicely fits with the FoxM1-dependent global transcriptional repression of most mitotic genes shown subsequently. Even though we show relevance for the segregation errors as facilitators of transition into full senescence later in the manuscript, at this earlier phase in the manuscript, we believe we should not present a reductionist description of the mitotic phenotype, as other defects might be acknowledged by future studies as potentially relevant for the evolution of the senescent phenotype (see revised paragraph 360-363).

3. I mentioned that the aneuploidy rates are extremely low in FoxM1 insufficient cells and questioned its biological significance. Meaningful information about aneuploidy rates is a prerequisite given the claims of the paper. If single-cell DNA sequencing is too expensive, one could conduct chromosome counts on metaphase spreads instead.

We believe we provide meaningful information about aneuploidy rates using measurements that are more precise than chromosome counts on metaphase spreads.

We used FISH analysis to quantify the aneuploidy index in asynchronous cell populations, and indeed it suggests that the aneuploidy levels are mild in elderly and FoxM1-depleted cells. However, because aneuploidy is known to compromise proliferative capacity, euploid cells will outcompete aneuploid cells and thus dilute the fraction of aneuploid cells in the population. Therefore, we used the Cyto D-FISH methodology, which allows measurement of chromosome mis-segregation events in the cell subpopulation that divided during the Cyto-D treatment (binucleated cells). Furthermore, we directly quantified chromosome mis-segregation events (anaphase laggards) by live-cell imaging and determined their fate after the mis-segregation event took place. The caveat of using metaphase spreads is that they would provide information about the karyotype of mitotic elderly cells before mis-segregation events would have occurred, and based on our live-cell imaging analysis in Fig. 7, we therefore expect that metaphase spreads would disclose a totally euploid cell subpopulation, as we show that the aneuploid elderly cells stop cycling.

4. The staining of beta-gal, p21 and 53BP1 in figure 7h (for the cell that has undergone a segregation error) seem very unconvincing for a cell that is apparently senescent and this casts doubts on the data shown in that figure. Could the authors use other markers for senescence such as p16 or LaminB to justify their claim?

We could use other markers for senescence such as p16 and lamin B1, even though we are technically limited in this microscopy analysis to a few channels (green, red and far red); thus it would never be possible to cover all these markers simultaneously in the same cell. We opt by the SA-bGal assay and the double immunostaining with p21/53BP1 as these biomarkers are more stringent to senescent vs. pre-senescent proliferating cells, have been optimized and identified for their advantages compared to p16 and lamin B1 in the lab. We prefer p21/Cdkn1A over p16/Cdkn2A, as in our

hands this staining is more robust in human dermal fibroblasts, in particular in combination with 53BP1 as a marker for DNA damage, which allows circumventing the interference of 53BP1-positive S-phase cells (that are p21-negative) and p21-positive cells which are still proliferative (but are 53BP1-negative). SA-bGal positive cells were always double-positive for p21/53BP1, confirming the robustness of this staining, but not all double-positive p21/53BP1 cells were SA-bGal positive, suggesting that SA-bGal staining is more stringent, perhaps because we defined a threshold of >5 foci for positive staining. Indeed, as recently shown in Figure S1B (Hernandez-Segura et al., Curr Biol 2017), CDKN1A is consistently upregulated in RNA-seq datasets of different types of fibroblast senescence, whereas CDKN2A is not. Regarding lamin B1, we found that early senescent aged cells already exhibit low levels (also evident from the RNA-seq dataset), nevertheless not excluding that it could still work as suitable biomarker for full senescent cells.

Also, according to the data in figure 7, only half of the cells that have undergone a segregation defect are positive for all 3 senescence markers tested (b-gal, p21 and 53BP1).

We believe we have addressed this concern above.

This thereby suggests that aneuploidy does not drive senescence but there might be other unknown events that cause the cell to arrest (p53 status in the cells). The authors should therefore show more convincing data in order to make this conclusion, or not make it all.

We do not exclude the possibility that other unknown events might cause the cell to arrest, which we can explain in the discussion section of the manuscript, nevertheless the Reviewer will certainly acknowledge the striking difference between cell cycle arrest rates observed in daughter cells that encountered segregation errors vs. daughter

cells arising from apparently normal mitoses (30% vs 100%). We feel that we do provide convincing data that shows that aneuploidy can be an ultimate stressor of full-blown senescence. Importantly, we do not only show this through this live-cell/fixed-cell correlative microscopy analysis, but also by showing that aneuploidy is prevalent in elderly senescent cells FACS-sorted based on high SA-bGal activity.